# Parallel single-cell metabolic analysis and extracellular vesicle profiling reveal vulnerabilities with prognostic significance in acute myeloid leukemia

Dorian Forte [1], Roberto Maria Pellegrino [2], Paolo Falvo [3,4], Paulina Garcia-Gonzalez[5], Husam B. R. Alabed[2], Filippo Maltoni[1,11], Davide Lombardi[3,4], Samantha Bruno [1], Martina Barone [6], Federico Pasini [1], Francesco Fabbri[7], Ivan Vannini[7], Benedetta Donati[8], Gianluca Cristiano[1], Chiara Sartor[1,11], Simona Ronzoni[9], Alessia Ciarrocchi [8], Sandra Buratta[2], Lorena Urbanelli [2], Carla Emiliani[2,10], Simona Soverini[1], Lucia Catani[1,11], Francesco Bertolini [3,4], Rafael José Argüello [5], Michele Cavo[1,11] & Antonio Curti [6] ✉

Acute myeloid leukemia (AML) is an aggressive disease with a high relapse rate. In this study, we map the metabolic profile of CD34+(CD38low/-) AML cells and the extracellular vesicle signatures in circulation from AML patients at diagnosis. CD34+ AML cells display high antioxidant glutathione levels and enhanced mitochondrial functionality, both associated with poor clinical outcomes. Although CD34+ AML cells are highly dependent on glucose oxidation and glycolysis for energy, those from intermediate- and adverse-risk patients reveal increased mitochondrial dependence. Extracellular vesicles from AML are mainly enriched in stem cell markers and express antioxidant GPX3, with their profiles showing potential prognostic value. Extracellular vesicles enhance mitochondrial functionality and dependence on CD34+ AML cells via the glutathione/GPX4 axis. Notably, extracellular vesicles from adverse-risk patients enhance leukemia cell engraftment in vivo. Here, we show a potential noninvasive approach based on liquid 'cell-extracellular vesicle' biopsy toward a redefined metabolic stratification in AML.

Acute myeloid leukemia (AML) is a clonal disorder that originates from a rare population of bone marrow (BM)-derived leukemic stem cells (LSCs)[1]. It is an aggressive disease with a high relapse rate due to cell-intrinsic chemoresistance mechanisms[2,3]. Changes in cell metabolism and metabolic adaptation are hallmark features of many cancers, including AML, thus emerging as promising therapeutic targets[4]. Normal hematopoietic stem cells (HSCs) differ from LSCs in their metabolic profile. Generally, HSCs are in a dormant state and utilize

glycolysis, whereas LSCs show metabolic inflexibility and uniquely rely on mitochondrial oxidative phosphorylation (OXPHOS)[5]. Moreover, LSCs produce leukemic blasts that can utilize either mitochondrial respiration or glycolysis[6,7]. Thus, within the niche, the high metabolic demand imposed by AML cells leads to metabolic adaptation that favors LSC survival and chemoresistance[8]. We previously demonstrated that nestin-positive BM stem/stromal cells (BMSCs) provide antioxidant molecules (mainly glutathione, GSH), energy sources

---

(mitochondria), and metabolites, mostly from the tricarboxylic acid (TCA) cycle, driving LSC metabolic adaptation and chemoresistance. In particular, GSH-dependent antioxidant pathways emerge as key players in BMSC-LSC crosstalk, balancing reactive oxygen species (ROS) levels during leukemogenesis and chemotherapy[4]. However, to date, only a few works[9–11] have correlated the metabolic reprogramming that occurs in AML with the influence on oxidation-reduction (redox) homeostasis. Indeed, redox metabolism could be considered an intriguing and unexplored therapeutic target for treating leukemia. Moreover, although cell-autonomous pathways, driven by genetic alterations, have been associated with specific metabolic profiles of leukemic cells[12], it has not yet been defined whether and which extrinsic signals may regulate leukemic metabolic reprogramming. Notably, the development of an accurate, affordable, and less invasive tool to explore disease metabolic dynamics is highly warranted in AML. Liquid biopsy is a revolutionary approach to probing the genomic profile of the tumor using peripheral blood (PB) components[13]. An increasing amount of evidence supports the straightforward correlations between blood and BM biopsies in hematological malignancies[14,15]. In this regard, extracellular vesicles (EVs), small vesicles released by cells, have recently attracted attention as carriers of complex intercellular information within the microenvironment in several cancers[16], including AML[17]. Since EVs contain tumor-derived material (DNA, RNA, proteins, and lipids), they can act as a reservoir of clinically relevant biomarkers[18], suggesting their pivotal role in 'cell biopsy'[19]. In AML patients, EV levels are elevated in plasma at the time of diagnosis and remain elevated in complete remission after chemotherapy[20,21]. Of note, the miRNA content of EVs can predict both the risks of relapse and overall survival in AML patients, suggesting a critical role and function in AML[22]. However, although growing evidence has reported a key role for EVs in regulating energy metabolism[23,24], the functional role of circulating EVs in regulating leukemic cell metabolism is still far from being defined. Additionally, systematic exploitation of EV-based liquid biopsy in AML has not yet been established.

Here, we developed a strategy for parallel exploration of the metabolic landscape of AML cells and the signature of EVs directly in PB. Functional studies with circulating EVs were used to support their capacity to modulate LSC metabolic reprogramming. The final aim of our study was to unravel metabolic vulnerabilities and biomarkers that may be used in the clinical setting.

## Results

### Circulating CD34+CD38low/- stem/progenitor AML cells show a distinct metabolic profile with increased levels of GSH and mitochondrial potential (MITO)

To minimize the impact of intratumoral heterogeneity, our study focused on the CD34+(CD38low/-) fractions, representing the most common immunophenotype in AML, for a more consistent and standardized approach. These fractions are known to exhibit chemotherapy resistance and possess the highest leukemogenic ability[25], making them crucial targets for understanding disease progression and developing effective treatments. In an attempt to monitor in real-time the metabolic phenotype in AML using liquid biopsy, we first screened fresh (within 1 h) PB samples (whole blood) from AML patients at diagnosis. We investigated redox metabolism, a critical subgroup of cellular metabolism[26], combining staining for ROS, GSH, and mitochondrial potential/functionality (MITO) (Fig. 1a–c). We first observed that the percentage of CD34+ cells with low ROS levels and high mitochondrial membrane potential (ROSlo MITOhi) were the most highly expressed fraction (mean percentage: 38%) compared to the other combinations including ROShi MITOlo ($p < 0.0001$), ROShi MITOhi ($p = 0.03$) and ROSlo MITOlo ($p = 0.02$) (Fig. 1a). In contrast, the CD34+ fraction with high ROS levels and low GSH content (ROShi GSHlo) appeared to be less expressed (mean percentage: 13%) in AML patients

than ROSlo GSHhi ($p = 0.007$) and ROSlo GSHlo ($p = 0.0002$) fractions (Fig. 1b). Interestingly, the simultaneous analysis of GSH and MITO content revealed a significant increase in the subset of GSHhi MITOhi within the CD34+ fraction (mean percentage: 38%) (Fig. 1c). Overall, AML patients showed a significant increase in CD34+ fractions with low levels of ROS and high levels of both GSH and MITO. To confirm the redox metabolic profile within the most immature hematopoietic stem and progenitor cell (HSPC) compartment, CD34+CD38low/- cells were also assessed. A similar redox metabolic profile was reported comparing overall CD34+ AML cells and more immature CD34+ CD38low/- stem/progenitor cells (Supplementary Fig. 1a–d). Then, we further investigated more specific leukemic subpopulations capturing progenitors (CD34+CD38+, CD34+, and/or CD117+), myeloid cells (CD33+, HLA-DR+), and primitive LSC (CD123+) (Supplementary Fig. 1e–h). Within the leukemic blasts, we observed a higher frequency of CD34-CD38- with low ROS levels and low mitochondrial potential (ROSlo MITOlo) compared to CD34+CD117+ blasts ($p = 0.03$) (Supplementary Fig. 1f). Also, the percentage of CD34-CD38- cells with low GSH content and low mitochondrial potential (GSHlo MITOlo) were increased compared to more mature CD34+CD38+ subset ($p = 0.04$; Supplementary Fig. 1h). No other differences were reported in the redox metabolic fraction for the leukemic subsets explored (Supplementary Fig. 1f–h).

To further demonstrate the accuracy of the PB source for the analysis reported above, we also measured the redox metabolic markers in CD34+ cells from paired PB and BM samples. Of note, no significant differences were found in paired samples between the two sources (Fig. 1d–f). However, in circulation, we found sex-dependent differences in redox metabolic phenotypes. Indeed, GSHlo MITOhi CD34+ cells were highly increased in female versus male AML patients, with a mean percentage of 33% versus 19%, respectively ($p = 0.04$; Fig. 1g). In contrast, female AML patients showed a significant reduction in the frequency of both ROShi MITOlo and GSHhi MITOlo CD34+ cells ($p = 0.02$ and $p = 0.05$, respectively; Fig. 1g).

Next, we aimed to identify metabolic features using fresh whole blood associated with European LeukemiaNet (ELN) 2022 categories[27]. The overall distribution of redox metabolic subsets was reported in AML patients stratified by ELN (Fig. 1h–k). In particular, an increase in the fraction ROSlo with MITOhi was observed in both intermediate- and adverse-risk patients compared to favorable-risk AML patients ($p < 0.05$, respectively) (Fig. 1i). Only intermediate-risk AML patients showed an increase in ROSlo GSHhi fraction compared to favorable-risk AML patients ($p = 0.03$) (Fig. 1j). In support of the potential impact of redox metabolic profiling on clinical outcome, we observed that AML patients with an elevated percentage of GSHhi MITOhi CD34+ cells, homogenously treated with conventional chemotherapy, experienced poor clinical outcomes (log-rank $p = 0.04$) (Supplementary: Fig. 1i, j).

To investigate the functional capabilities of these redox metabolic subsets, we repeated the experiments using isolated CD34+ AML cells. The cells were labeled for ROS, MITO, and GSH and then sorted accordingly. Specifically, we focused on the ROS-low and ROS-high subsets in relation to their GSH content or mitochondrial functionality. We also compared the fractions of cells with low and high GSH within the CD34+ AML cells that exhibited high mitochondrial potential (Supplementary Fig. 1k–m). We then evaluated the survival and proliferative capacity of each subset using a colony-forming cell assay. Intriguingly, the ROSlo population revealed a higher and significant clonogenic potential in both scenarios with high GSH content ($p = 0.03$) and high mitochondrial potential ($p = 0.03$) (Supplementary: Fig. 1k, l). While a high level of GSH and high mitochondrial potential led to an increase in clonogenic output, the change was not statistically significant when compared to the fraction with low GSH and high mitochondrial potential (GSHlo MITOhi) (Supplementary: Fig. 1m).

Here, we observed that the leukemic hematopoietic compartment in AML patients shows a transition to lower ROS levels and higher

mitochondrial potential and GSH levels, suggesting a more aggressive disease. Our data also suggest a combination of metabolic markers that may help risk stratification and clinical prediction of AML patients.

## Metabolic profiling of circulating CD34+ AML cells using the SCENITH method unveils both high glucose dependence and glycolytic capacity with prognostic significance

To further explore the metabolic reprogramming of circulating LSCs, we employed the single-cell energetic metabolism by profiling the translation inhibition (SCENITH™)[28] approach. Using protein synthesis level detection upon blockade of different metabolic pathways (as detailed in the "Methods" section), we measured puromycin incorporation as a surrogate for ATP production and energetic metabolism in fresh blood at single-cell resolution. Interestingly, we detected high puromycin levels, displayed as median fluorescence intensity (MFI), in CD34+ cells in comparison to paired CD3+ cells after vehicle (Control, Co) or oligomycin treatment (O) ($p < 0.0001$, respectively; Fig. 2a), indicating higher protein synthesis and metabolism in the CD34+ AML cell fraction.

Accordingly, we observed that CD34+ cells relied primarily on glucose oxidation (glucose dependence; mean percentage 83.77%), exhibiting a high glycolytic capacity (glyco cap, mean percentage 78.58%); at the same time displaying a low rate of mitochondrial dependence (mito dep, mean percentage 21.42%) and a low fatty acid and AA oxidation capacity (FAAO capacity, mean percentage 16.23%), indicative of their limited ability to use fatty acids and amino acids as sources for ATP production (Fig. 2b).

When we compared the metabolic status of circulating CD34+ AML cells with the metabolic state of paired BM resident fractions, no significant differences were reported in puromycin levels (Supplementary Fig. 2a) or metabolic parameters (Fig. 2c). However, we observed in circulation a significant increase in mitochondrial dependence and a reduction in glycolytic capacity in the more immature CD34+CD38low/- fractions ($p < 0.05$, respectively) (Fig. 2d), which was slightly increased in paired BM fractions (Supplementary Fig. 2b). Our analysis revealed that more specific leukemic subsets, such as progenitors, myeloid cells, and primitive LSCs, did not exhibit significant differences in the metabolic parameters evaluated

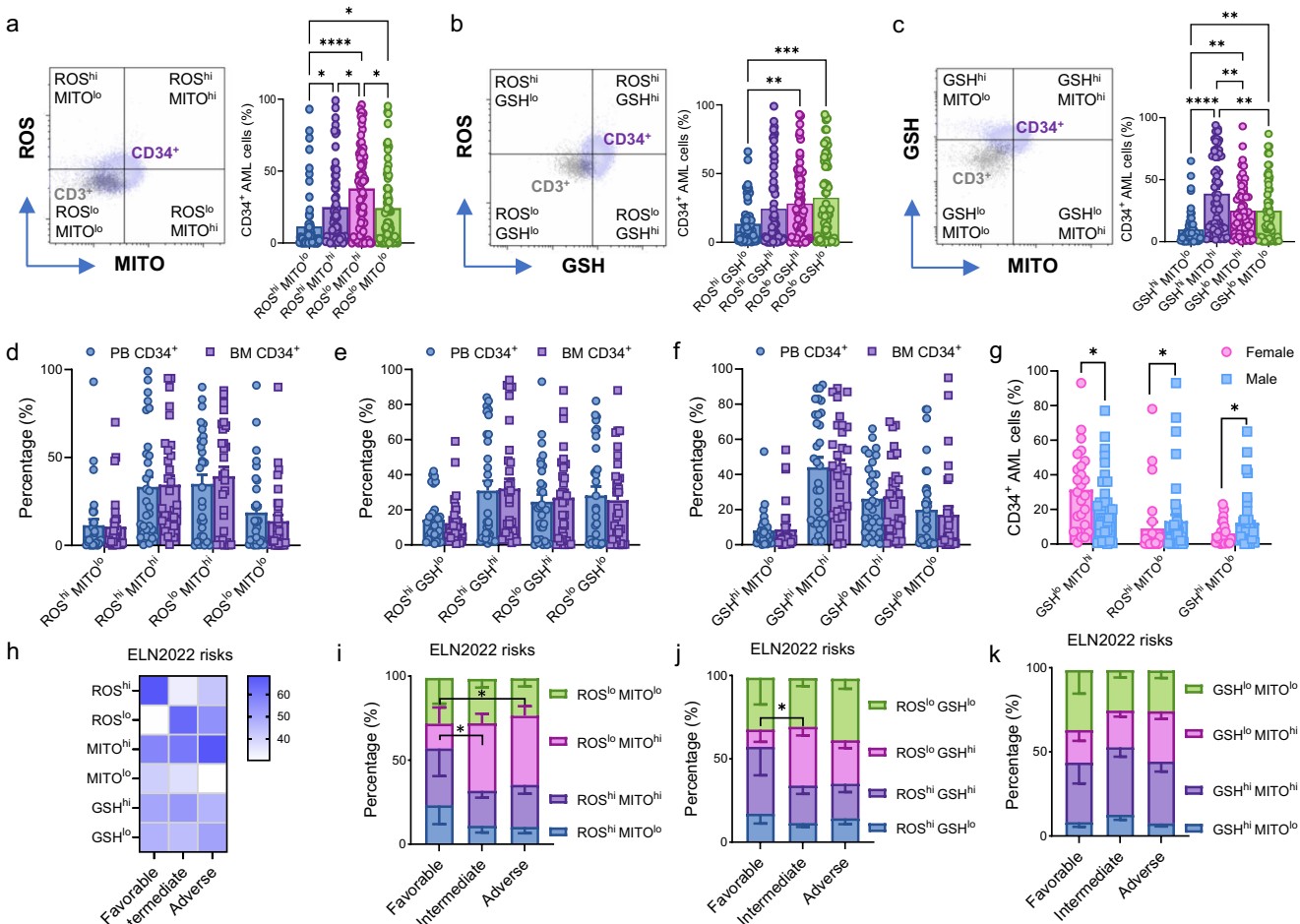

**Fig. 1 | Redox metabolic profile of CD34+ AML cells from fresh whole blood. a–c** On the left, representative dot plots illustrating the gating strategy used to profile CD34+ cells based on CD3+ cells for paired staining combinations of CellROX (ROS), MitoTracker CMXRos (MITO) and Thiol Tracker (GSH). On the right, percentages of CD34+ cells stained for ROS/MITO (**a**; $p = 0.02$, $p < 0.0001$, $p = 0.03$, $p = 0.03$, $p = 0.02$), ROS/GSH (**b**; $p = 0.007$, $p = 0.0002$), and GSH/MITO (**c**; $p < 0.0001$, $p = 0.001$, $p = 0.001$, $p = 0.005$, $p = 0.005$) in the blood of AML patients at diagnosis ($n = 62$). One-way ANOVA with Tukey's correction for multiple comparison. Graphs for profiling AML CD34+ cells from paired PB versus BM whole blood according to ROS/MITO (**d**), ROS/GSH (**e**), and GSH/MITO (**f**) combinations ($n = 31$). Two-way ANOVA with Šidák's multiple comparisons test. **g** Graph reporting

sex differences in the following CD34+ cell fractions: GSHlo MITOhi ($p = 0.04$), ROShi MITOlo ($p = 0.02$), GSHhi MITOlo ($p = 0.05$) considering female AML patients ($n = 26$) versus male AML patients ($n = 37$). Mann-Whitney unpaired t-test. **h** Heatmap reporting the mean percentage differences for ROS, MITO, and GSH in CD34+ cells considering both high and low expression for each marker in AML patients stratified by ELN risk. **i–k** Metabolic profile of AML CD34+ cells in the same AML patients stratified by ELN risk (favorable, $n = 6$; intermediate, $n = 28$; adverse, $n = 27$) for ROS/MITO (**i**; $p = 0.04$, $p = 0.04$), ROS/GSH ($p = 0.03$) and GSH/MITO (**k**). Two-way ANOVA with Dunnett's multiple comparison test. For all panels (*) $p < 0.05$; (**) $p < 0.01$; (***) $p < 0.001$; (****) $p < 0.0001$. Data presented as mean values ± SEM. Source data are provided as a Source Data file.

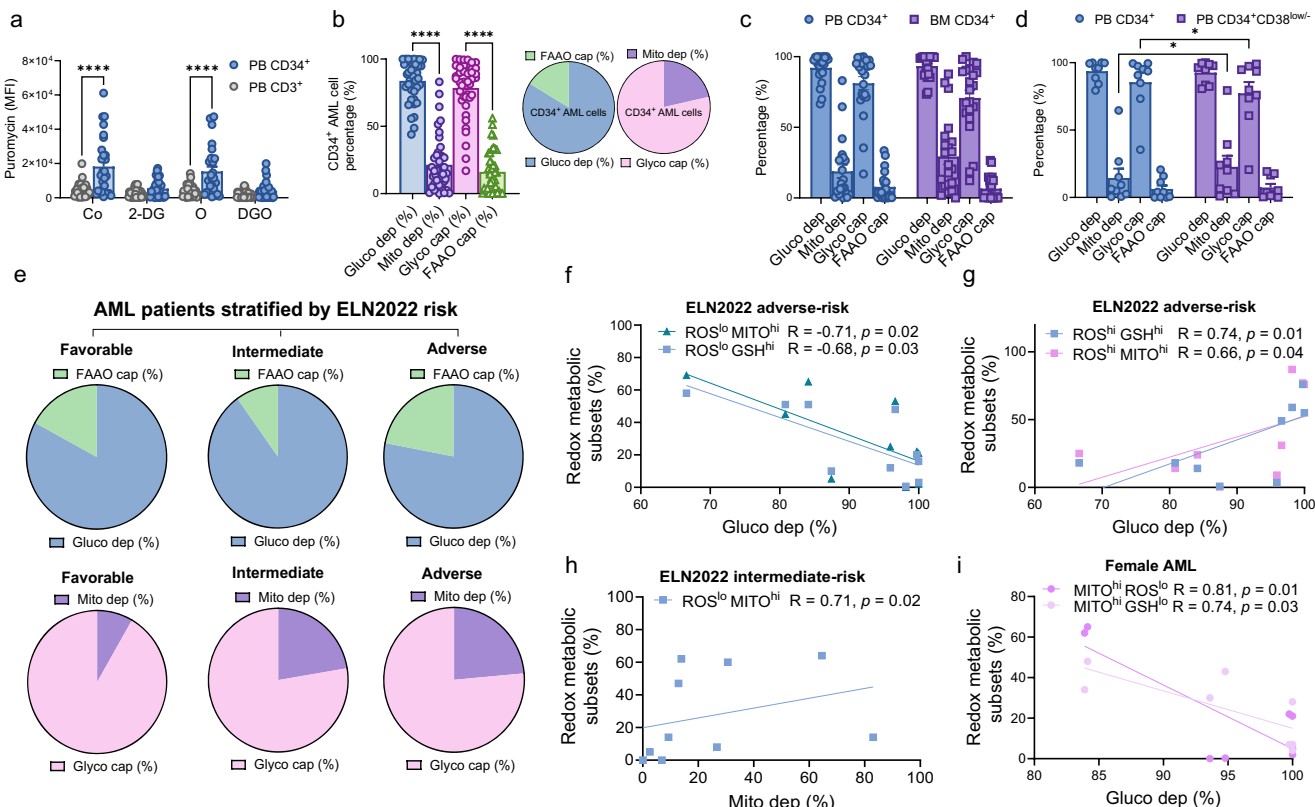

**Fig. 2 | Metabolic capacities and dependencies of CD34+ AML cells from fresh blood of AML patients at diagnosis using the SCENITH method. a** Translation level expressed by puromycin value (anti-Puro gMFI) after inhibition of metabolic pathways with Co (control, vehicle; $p < 0.0001$), 2-deoxy-D-glucose (2DG), oligomycin A (O; $p < 0.0001$) or both (DGO) comparing paired AML CD34+ and CD3+ cells from each patient (AML patients, $n = 29$). Two-way ANOVA with Šidák's multiple comparisons test. **b** Metabolic profile of AML CD34+ cells expressed in percentages with the corresponding pie charts representative of two-by-two dependent parameters measured, namely, glucose dependence (gluco dep) with fatty acid and AA oxidation capacity (FAAO cap) and mitochondrial dependence (mito dep) with glycolytic capacity (glyco cap) ($n = 42$; $p < 0.0001$). One-way ANOVA with Tukey's multiple comparison test. **c** Metabolic parameters according to the source, comparing paired AML PB CD34+ cells versus BM CD34+ cells ($n = 22$). **d** Differences in metabolic parameters between AML PB CD34+ and paired PB CD34+CD38low/- ($n = 9$) ($p = 0.04$; $p = 0.04$). Two-way ANOVA with Šidák's multiple comparisons test (**c**, **d**).

**e** Pie charts representing the metabolic profile in AML patients stratified according to ELN risk (favorable, $n = 6$; intermediate, $n = 18$; adverse, $n = 19$). **f** Inverse correlations between glucose dependence (%) in AML CD34+ cells measured by SCENITH and the percentages of ROSlo MITOhi (R = −0.71, $p = 0.02$) or ROSlo GSHhi (R = −0.68, $p = 0.03$) CD34+ cells from adverse-risk AML patients ($n = 10$). **g** Positive correlations between glucose dependence (%) in AML CD34+ cells with ROShi GSHhi (R = 0.74, $p = 0.01$) and ROShi MITOhi (R = 0.66 and $p = 0.04$) CD34+ cells from adverse-risk AML patients ($n = 9$). **h** Positive association between ROSlo MITOhi CD34+ cells and mitochondrial dependence (%) of CD34+ cells from intermediate-risk AML patients (R = 0.71, $p = 0.02$) ($n = 10$). **i** Inverse associations between MITOhi ROSlo CD34+ cells (R = 0.81, $p = 0.01$) and MITOhi GSHlo (R = 0.74, $p = 0.03$) in female AML patients ($n = 8$). Spearman's test reported correlations. For all panels (*) $p < 0.05$; (****) $p < 0.0001$. Data are presented as mean values ± SEM. Source data are provided as a Source Data file.

(Supplementary Fig. 2c–f). To gain insight into the potential clinical implications of metabolic profiling captured by SCENITH analysis, we analyzed metabolic data across ELN risk-adapted patient subsets. We observed that AML patients in the favorable group had a reduction in mitochondrial dependence at the expense of glycolysis. Specifically, favorable AML patients unveiled a mitochondrial dependence equal to 8.2% compared to the adverse and intermediate groups (mean percentage: 23.5% and 22.3%, respectively; $p < 0.05$ versus the adverse group) (Fig. 2e). Moreover, intermediate-risk AML patients revealed a higher glucose capacity compared to adverse group ($p < 0.05$; Fig. 2e). Then, we coupled SCENITH analysis with the redox metabolic profiling reported above to gain further insight into the global energetic metabolism of circulating LSCs detected directly in fresh whole blood. In the ELN adverse group, we found an inverse correlation between glucose dependence (%) and the percentage of ROSlo with GSHhi ($p = 0.03$) or MITOhi ($p = 0.02$) CD34+ cells (Fig. 2f). In contrast, the percentages of ROShi MITOhi/GSHhi CD34+ cells were positively correlated with glucose dependence ($p = 0.04$ and $p = 0.01$, respectively) (Fig. 2g). Of note, only intermediate-risk AML patients showed that ROSlo MITOhi CD34+ cell frequencies were positively associated

with mitochondrial dependence ($p = 0.02$) (Fig. 2h). Although sex-dependent analysis did not reveal any difference in metabolic parameters (Supplementary Fig. 2g), female AML patients exhibited an inverse correlation between glucose dependence (%) measured by SCENITH and the frequency of CD34+ cell subsets with high MITO and both low ROS ($p = 0.01$) and GSH levels ($p = 0.03$) (Fig. 2i).

All data reported above highlight the clinical relevance of real-time metabolic studies on fresh whole blood. Despite a high dependence on glucose and a preferential skewing toward glycolysis in AML total CD34+ cells at diagnosis, the more immature leukemic fractions show a prominent mitochondrial dependence, which appears relevant in AML risk stratification.

**Analysis of extracellular signals in circulation from AML patients at diagnosis reveals an increase in size, protein content and the presence of glutathione peroxidase 3 (GPX3) in EVAML**

To further profile AML patients through liquid biopsy, we focused our attention on circulating EVs isolated by size exclusion chromatography (SEC). For this purpose, we first compared the number and size of EVs isolated from the plasma of AML patients at diagnosis, herein referred

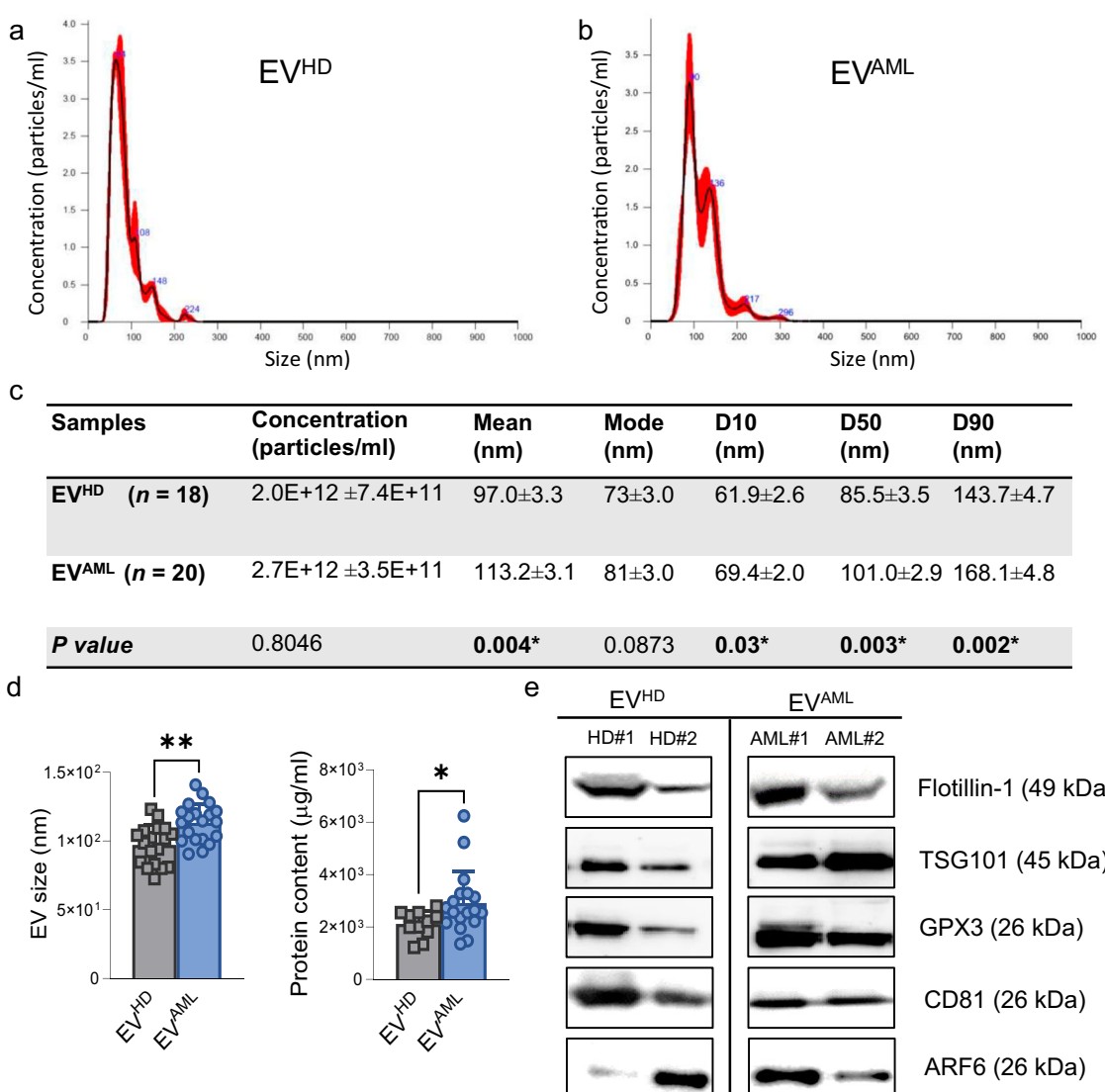

**Fig. 3 | General characterization of circulating EV^AML in comparison to EV^HD.**
**a**, **b** Representative histograms of plasma-derived EV distributions measured by NTA for EV^HD ($n = 18$) and EV^AML ($n = 20$). **c** Summary table report of NTA data for each group. NTA results present the total number of particles per ml as well as the average median size and mode. Size distribution where 10% (D10), 50% (D50), and 90% (D90) of the sample is also reported. **d** Plasma-derived EV size measured by NTA ($p = 0.004$) and EV protein content measured using Bradford's assay ($p = 0.03$) from AML patients ($n = 20$ for EV size; $n = 18$ for protein content) and HD ($n = 18$ for

EV size; $n = 11$ for protein content). **e** Representative western blot analysis of specific EV markers (namely, Flotillin-1, TSG101, CD81, ARF6) and the antioxidant GPX3 in two individuals per group ($n = 2$ biological replicates for each group). Uncropped scans of Western blots in the figures are provided as a Source Data file. Significant differences were reported using the Mann–Whitney test for unpaired samples with (*) $p < 0.05$ *and* (**) $p < 0.01$ considered significant. Data are presented as mean values ± SEM.

to as EV^AML, with those of EVs isolated from the plasma of a cohort of sex/age-matched HD, herein referred to as EV^HD.

Nanoparticle tracking analysis (NTA) revealed that EV concentrations were comparable between the two groups tested, as reported in Fig. 3a, b. However, all particle size distributions, as expressed by the diameter of particles (D10, D50, and D90), were significant higher in EV^AML, thus demonstrating larger EV populations in AML patients than in the control group (Table in Fig. 3c). Indeed, we observed an increase in the average particle size of EV^AML compared to EV^HD ($p = 0.004$; Fig. 3d). Additionally, the protein concentration reported for EVs from the cohort of AML patients tested was higher than that reported for EV^HD ($p = 0.03$; Fig. 3d). These NTA data were supported by TEM, which qualitatively identified EVs as a group of heterogeneous spheroids with distinct characteristics (Supplementary Fig. 3a). Then, based on MISEV (2023) guidelines[29], we also tested typical EV markers by western blot, including plasma membrane flotillin-1 and

exosomal markers such as TSG101 and CD81, which were detected in all EV samples. Consistent with the above findings on EV size, the marker for large EVs, namely, ADP-ribosylation factor 6 (ARF6), was found in EV^AML (Fig. 3e). Notably, we and others suggested the involvement of the glutathione peroxidase (GPX) family in the redox metabolism of LSCs[4,30], leading us to explore their presence in EVs with a special focus on plasma GPX3[31]. Interestingly, we revealed the presence of GPX3 in EV^AML (Fig. 2e). Western blot analyses revealed a substantial decrease in contaminants such as albumin and apolipoprotein A1 (apo-A1) in our EV fractions in comparison with the serum counterparts. Calnexin was only detected in cell lysates (Supplementary Fig. 3b).

Taken together, these data indicate that EV^AML isolated with our method show typical EV features and the cargo of the antioxidant GPX3, prompting us to further investigate EV-based liquid biopsy in AML and their potential metabolic roles.

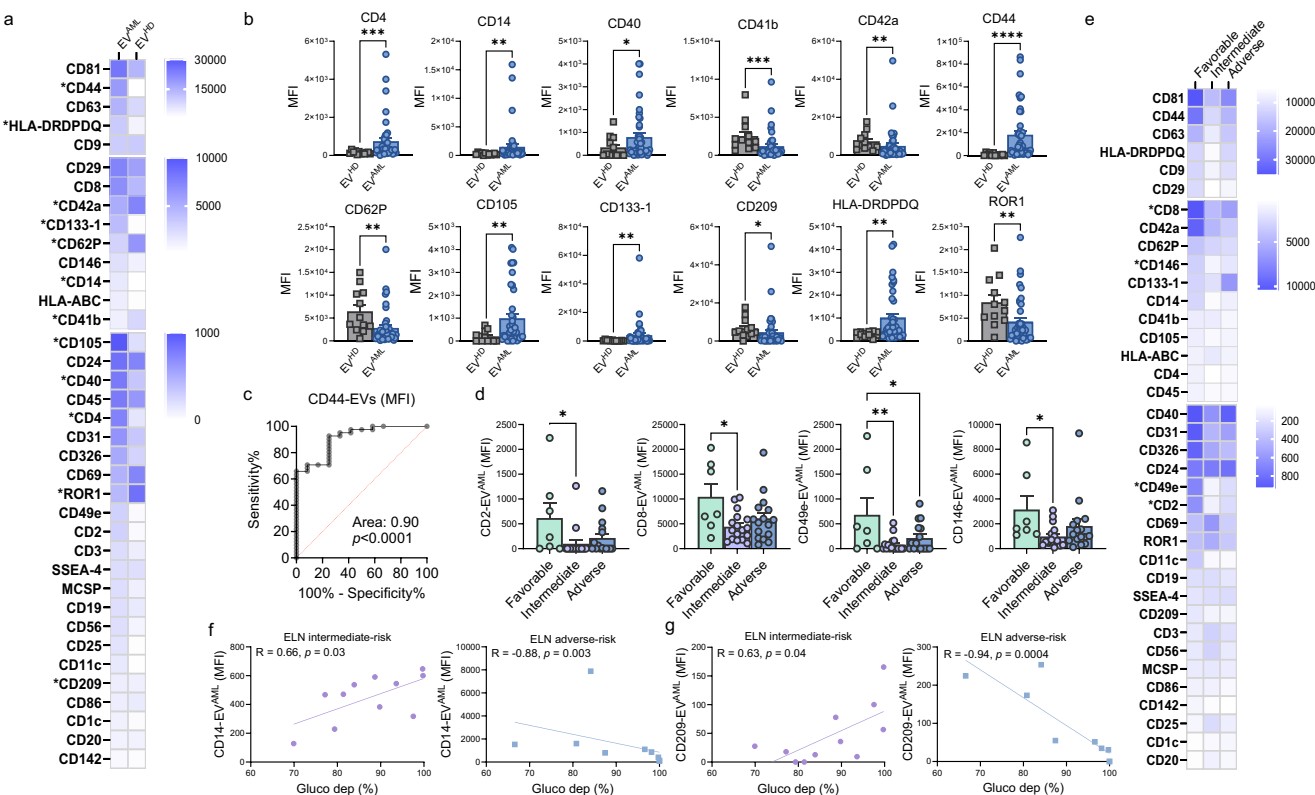

**Fig. 4 | Protein surface signatures on circulating EVs from HD and AML patients. a** Heatmap of the expression of 37 EV surface markers expressed on EV[HD] (n = 12) and EV[AML] (n = 41). **b** Background-corrected median APC fluorescence intensity for surface markers significantly different comparing EV[HD] (squares in gray; n = 12) and EV[AML] (circles in blue; n = 41): CD4 (p = 0.0007), CD14 (p = 0.0045), CD40 (p = 0.04), CD41b (p = 0.0005), CD42a (p = 0.003), CD44 (p < 0.0001), CD62P (p = 0.001), CD105 (p = 0.007), CD133-1 (p = 0.009), CD209 (p = 0.01), HLA-DRDPDQ (p = 0.008), and ROR1 (p = 0.004); **c** Area under the ROC curve for EV CD44 expression to discriminate AML patients (n = 41) versus HD subjects (n = 12) (AUC = 0.9; p = 0.0001); **d** Background-corrected median APC fluorescence intensity for surface markers (CD2, CD8, CD49e, CD146) significantly different in favorable-risk AML patients compared to intermediate-risk patients (p = 0.03, p = 0.01, p = 0.007, p = 0.04) or adverse-risk patients (p = 0.04). Two-way ANOVA with Tukey's multiple comparison test. **e** Heatmap for the expression of 37 EV

surface markers expressed on EV[AML] from patients stratified according to the ELN risk stratification: favorable-risk (n = 7), intermediate-risk (n = 18), and adverse-risk AML patients (n = 16). **f** Spearman's correlations between CD14 MFI on EV[AML] from intermediate-risk patients (R = 0.66, p = 0.03; n = 11) or adverse-risk patients (R = −0.88, p = 0.003; n = 8) with glucose dependence (%) reported on AML CD34[+] cells measured using SCENITH. **g** Spearman's correlations between CD209 MFI on EV[AML] from intermediate-risk patients (R = 0.63, p = 0.04; n = 11) or adverse-risk patients (R = −0.94, p = 0.0004; n = 8) with glucose dependence (%) reported on AML CD34[+] cells. Significant markers between HD and AML-derived EVs were reported as (*) p < 0.05, (**) p < 0.01, (***) p < 0.001, and (****) p < 0.001 using the Mann−Whitney test for unpaired samples or one-way ANOVA with Tukey's multiple comparison test unless stated. Data are presented as mean values ± SEM. Source data are provided as a Source Data file.

## Circulating EVs from newly diagnosed AML patients are mainly enriched in CD44, and multiplex protein analysis reveals potential prognostic value

To simultaneously profile the protein expression pattern of the previously isolated circulating EVs, a multiplex assay MACSPlex was performed to analyze the signal of 37 proteins on the EV surface to identify the cell origin.

First, we analyzed the overall MFI for each marker comparing EV[AML] versus EV[HD] (Fig. 4a). Considering the exosome markers CD9, CD63, and CD81, we did not detect any differences between the tested groups. Of note, EV[AML] were highly enriched in cancer stem cell markers, namely, CD44 (p < 0.0001) and CD133-1 (p = 0.009), in comparison to EV[HD]. Similarly, markers related to immune cell modulation, such as CD4 (T helper marker; p = 0.0007) and HLA-DRDPDQ (antigen-presenting cell marker; p = 0.008), as well as markers associated with monocytes, macrophages (CD14 and CD40; p = 0.04, respectively) and mesenchymal stromal cells (CD105; p = 0.007), were mostly expressed in EV[AML] and detected at low levels in EV[HD]. In contrast, CD62P (platelet activation marker; p = 0.001), CD41b and CD42a (platelet/mega-karyocyte markers; p = 0.0005 and p = 0.003) and CD209 (dendritic marker; p = 0.01) were significantly depleted in EV[AML] (Fig. 4a, b). Surprisingly, ROR1, a stem cell marker, was significantly reduced in EV[AML]

patients compared to EV[HD] (p = 0.003). Overall, ROC curve analysis confirmed the reliable diagnostic value for selected EV markers in AML patients, including CD4, CD105, and HLA-DRDPDQ, showing the best cutoff for CD44 MFI on EVs discriminating AML patients from HD subjects (AUC: 0.90, p < 0.0001; Fig. 4c).

Interestingly, we found a positive correlation between the frequency of BM blasts and the expression of CD44 on circulating EV[AML] (Supplementary Fig. 4a and Supplementary Table 2) suggesting CD44 as a putative biomarker for assessing BM blast levels in circulation.

According to ELN risk stratification, we found that the EV MFI of CD2, CD8, CD49e, and CD146 from the intermediate-risk AML groups and adverse-risk group (only for CD49e) were significantly reduced in comparison with favorable-risk AML patients (Fig. 4d, e). Remarkably, AML patients who had not been treated intensively and showed low expression of CD40 (HR = 0.23, 95% CI, 0.05−0.99, p = 0.04) or CD62P (OS, HR = 0.24, 95% CI, 0.05−1.02, p = 0.05) on circulating EV[AML] experienced poor outcomes, as reported by univariable Cox regression analysis for overall survival (Supplementary Fig. 4b, c). Finally, we explored the link between surface protein expression on EV[AML] and the metabolic profile reported above for circulating CD34[+] AML cells. We observed positive correlations between the MFI of CD86 detected on EV[AML] and the percentage of CD34[+] cells with ROS[lo] and both GSH[hi] and

MITO$^{hi}$. Additionally, the MFI of the cancer stemness marker CD24 was significantly associated with the percentage of GSH$^{lo}$ MITO$^{hi}$ CD34$^+$ cells (Supplementary Fig. 4d–f), suggesting a mirrored effect of cell metabolic status on EV surface expression markers. Moreover, we revealed that the glucose dependence of CD34$^+$ AML cells was strongly and negatively associated with the expression of EV markers detected in adverse-risk AML patients, including CD4, CD14, CD31, CD40, CD44, CD49e, CD142, CD133-1, and CD209. Conversely, intermediate-risk AML patients showed positive correlations between glucose dependence and EV markers such as CD14, CD209, MCSP, and SSEA-4 (Supplementary Table 3). Surprisingly, adverse and intermediate-risk patients revealed 'asymmetrical' correlations between CD14 and CD209 levels on EV$^{AML}$ and glucose dependence (Fig. 4f, g). To capture a more comprehensive and specific picture of AML, we examined paired EV$^{AML}$ obtained from both PB and BM plasma. Interestingly, we found that 9 markers were significantly higher in BM EV$^{AML}$, including markers related to antigen-presenting molecules (HLA-ABC and HLA-DRDPDQ; $p < 0.001$), T cells (CD4; $p < 0.01$), monocytes (CD14/CD40; $p < 0.05$), platelets (CD41b, CD62; $p < 0.05$) and mesenchymal stromal cells (CD29, CD105; $p < 0.05$) (Supplementary Fig. 4g).

Collectively, these results demonstrate that an abnormal signature characterizes the phenotype of EV$^{AML}$. Interestingly, the expression of EV markers is closely linked to cell metabolism, and the depletion of selected markers on EVs defines intermediate- and adverse-risk patients, supporting an EV signature with potential prognostic power.

## A distinct lipidomic profile is detected in EV$^{AML}$ and highlights putative biomarkers for AML stratification

The EV lipidome might reflect the disease-specific metabolism of the parent cells from which they originate[32]. Thus, we performed an untargeted lipidomic analysis on EV preparations from AML patients versus HD. In our dataset, based on the acceptance criteria outlined in the "Methods" section, the variance between theoretical and experimental MS1 was 4.30 ppm. The average score assigned by the MS-DIAL annotation software was 96.7%. Additionally, we observed an odd/even ratio of 3.28% for EV$^{AML}$ and 3.94% for EV$^{HD}$. Lipidomic analysis revealed 267 lipid species annotated at the molecular species level, grouped into 9 lipid classes (Supplementary Table 4, Source file). We reported the abundance of the following lipid classes in the overall isolated EVs: phosphatidylcholine (PC: 44.69%), free fatty acid (FA: 15.85%), diacylglycerol (DG: 11.61%), ceramide (Cer: 11.37%), and sphingomyelin (SM: 10.62%). As reported by the bar graph, within lipid classes, DGs were significantly overexpressed in EV$^{AML}$ compared to EV$^{HD}$ (Fig. 5a). According to network analysis, in AML patients, we observed an increase in Cer, which seems to occur along with the upregulation of metabolic pathways that have SM as substrates. Indeed, PC works as a substrate for DG and lysophosphatidylcholine (LPC). The increase in FA is related to the biosynthetic pathway with PC and phosphatidyli-nositol (PI) as precursors (Fig. 5b).

The entire lipidomic dataset was then processed using the MetaboAnalyst (5.0) platform to perform different types of univariate and multivariate analyses. The supervised PLS analysis shows the separation between the AML group and the HD group in the partial least squares discriminant analysis (PLS-DA) (accuracy = 0.91, R2 = 0.89, and Q2 = 0.61). The most highly expressed molecular species were used for the multivariate ROC analysis and unveiled a promising area under the curve (AUC = 0.98) (Fig. 5c) for further investigation as potential biomarkers. The corresponding VIP plot and heatmap were used to list the most important lipid molecular species causing clusterization (Fig. 5d). In particular, within the most highly expressed lipid classes in AML (namely, FA and DG), FA 18:1, DG 16:0_18:1, DG 16:0_18:2, DG 18:0_18:1, and DG 18:1_18:2 were the most differentially abundant lipid species in EV$^{AML}$ compared to EV$^{HD}$ (Fig. 5d, e). Additionally, despite no difference in the corresponding class, we detected overexpression in EV$^{AML}$ for other molecular species, including Cer (Cer 18:1;20/24) and

alkyl-phosphatidylcholine (PC O-18:1_20:4). Of interest, several species belonging to the PC and SM classes were depleted in EV$^{AML}$. Notably, concerning clinical disease features and ELN risk stratification, we found that favorable-risk AML patients revealed a decrease in the concentration of FA 16:0 compared to intermediate-risk patients ($p = 0.04$) and a higher concentration of LPC 18:3 compared to adverse-risk patients ($p = 0.04$). Notably, adverse-risk patients reported a lipidomic signature distinct from that of intermediate-risk patients, with a significant depletion in DG 18:1_20:4 ($p = 0.01$), FA 21:0 ($p = 0.04$), and PE 18:0_22:5 ($p = 0.03$) in the adverse group, but not in SM 18:2;20/24:3 ($p = 0.006$) (Fig. 5f).

To further investigate the cargo of our EV$^{AML}$, we conducted a targeted metabolomic analysis focusing on the content of mitochondrial TCA cycle intermediates in our EVs. This analysis was based on our previous data where TCA intermediates were found to be increased in AML[4]. Although we did not observe significant differences between EV$^{AML}$ and EV$^{HD}$, we noted that several metabolites (namely fumarate, aconitic acid, shikimic acid, citrate, isocitrate, and glucose) showed a two-fold increase in adverse or intermediate EV compared to favorable-risk EVs. Sucrose was detected in only three patients and was not present in EVs from HD (Supplementary Fig. 4h). Notably, α-ketoglutarate was significantly elevated in adverse EV$^{AML}$ compared to favorable EVs ($p = 0.03$) (Fig. 5g).

Taken together, lipidomic analysis of circulating EVs identifies a lipid-based abnormal signature of EVs from AML patients and suggests the alteration of the EV lipidome in relation to risk stratification at diagnosis. Furthermore, the targeted metabolomic studies demonstrate an elevated level of α-ketoglutarate in adverse EV$^{AML}$.

## Human leukemia cell lines show a different response to EV$^{AML}$ in redox metabolism

Given our results on the metabolic profiling of AML cells and the distinct phenotype and cargo of EV$^{AML}$, we then investigated the functional influences of EV$^{AML}$ on the redox metabolism of various human leukemia cell lines and primary CD34$^+$ AML cells. After coculturing AML cells in the presence of optimal concentrations of EVs, we first analyzed the cells by flow cytometry, proving EV uptake within 24 h for functional studies. EVs isolated from AML patients were stained with the green membrane dye PKH67 and then co-cultured with CD34$^+$ AML cells for 4 to 24 h at 37 °C. The uptake of EV$^{AML}$ was measured by flow cytometry (Supplementary Fig. 5a). To rule out passive uptake, a negative control was conducted using PKH67 dye without EVs, or by keeping the AML CD34$^+$ cells at 4 °C to inhibit uptake. The data confirmed the EV uptake within 24 h as an active process.

Then, we simultaneously detected ROS and GSH levels along with mitochondrial potential as reported above for whole blood on human cell lines from AML patients: MOLM-13, KG-1, OCI-AML3, MV4-11, and Kasumi-1 (Supplementary Fig. 5, 6). When the MOLM-13, KG-1 and OCI-AML3 were cocultured with EV$^{AML}$, we noticed that MOLM-13 only showed a lower percentage of ROS$^{hi}$ MITO$^{hi}$ in the presence of EV$^{AML}$ ($p < 0.01$; Supplementary Fig. 5b), whereas OCI-AML3 showed an increase in the fraction ROS$^{lo}$ MITO$^{hi}$ ($p < 0.05$; Supplementary Fig. 5d). Moreover, MOLM-13 and OCI-AML3 showed a significant increase in ROS$^{lo}$ GSH$^{hi}$ fraction in the presence of EV$^{AML}$ ($p < 0.05$, respectively; Supplementary Fig. 5e, g). In addition, MOLM-13 and KG-1 shared a considerable reduction in ROS$^{hi}$ GSH$^{lo}$ ($p < 0.01$ for MOLM-13 and $p < 0.05$ for KG-1; Supplementary Fig. 5e, f) and a concomitant decrease in the fraction GSH$^{lo}$ MITO$^{hi}$ ($p < 0.05$, respectively) (Supplementary Fig. 5h, i). Only the KG-1 cell line presented an increase in the fraction with high GSH content and high mitochondrial potential after cocultures with EV$^{AML}$ ($p < 0.05$: Supplementary Fig. 5i). Conversely, EV$^{AML}$ showed a limited effect on redox metabolic markers in the MV-4-11 and Kasumi-1 cell lines (Supplementary Fig. 6a–f). Only the fraction of ROS$^{hi}$ MITO$^{hi}$ cells decreased in response to EV$^{AML}$ in MV-4-11 cells ($p < 0.05$; Supplementary Fig. 6a).

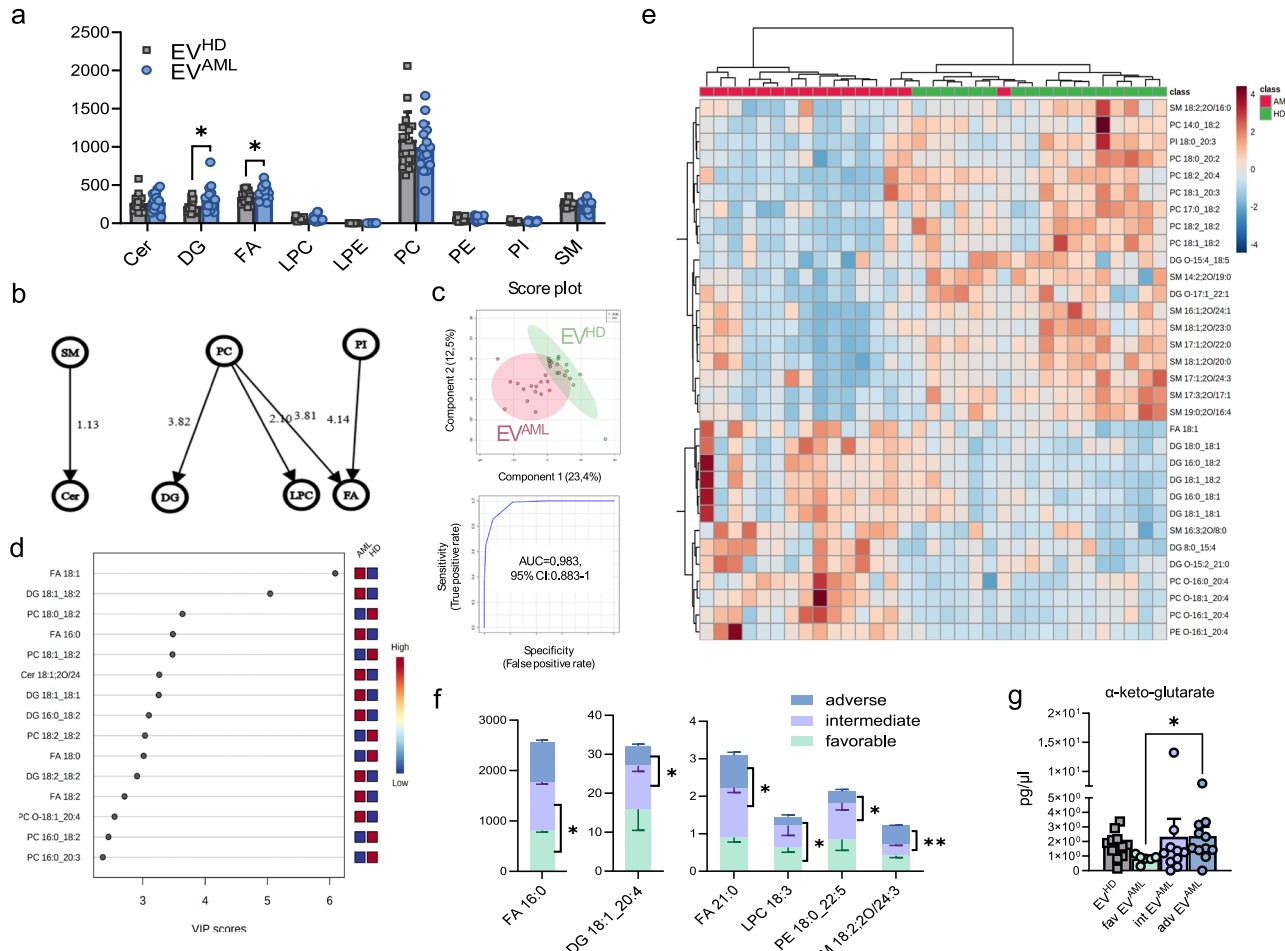

**Fig. 5 | Lipidomic analysis of EVs. a** Graph for the nine lipid classes detected in EV$^{AML}$ ($n = 16$) and EV$^{HD}$ ($n = 17$) ($p = 0.01$ for DG and $p = 0.03$ for FA). **b** Network of transformations between lipid classes. The lines connecting the nodes indicate the interaction's direction and status. The numbers (z-score) indicate the intensity of the reaction on an arbitrary scale. **c** Partial least squares-discriminant analysis (PLS-DA) for lipid species differentially expressed in EV$^{AML}$ compared to EV$^{HD}$ and multivariate ROC analysis of the EV lipidomic dataset between AML patients and HD (AUC = 0.98). **d** Variable importance in projection (VIP) scores of PLS-DA for lipid species in EV$^{AML}$ and EV$^{HD}$. The colored boxes on the right indicate the relative concentrations of the corresponding lipid species (red for high; yellow for intermediate and blue for low levels). **e** Corresponding heatmap for lipid species. **f** Selected lipid species levels differentially expressed in EVs, namely, FA 16:0

($p = 0.04$), LPC 18:3 ($p = 0.04$), DG 18:1_20:4 ($p = 0.01$), FA 21:0 ($p = 0.04$), PE 18:0_22:5 ($p = 0.03$) and SM 18:2;20/24:3 ($p = 0.006$), according to ELN risk status: favorable ($n = 3$), intermediate ($n = 6$) and adverse ($n = 5$) risk. Graphs are obtained based on the data matrix normalized by the median and autoscaled. Significant markers were reported as (*) $p < 0.05$, (**) $p < 0.01$ by Mann–Whitney test for unpaired samples or two-way ANOVA with Sidak's multiple comparisons tests. **g** Targeted metabolomic data on EV$^{AML}$ (n = 27) versus EV$^{HD}$ (n = 12). TCA cycle intermediate α-ketoglutarate expressed as pg/μl between favorable (n = 6), intermediate (n = 10), and adverse (n = 11) EV$^{AML}$ ($p = 0.03$). A significant difference was reported by the Kruskal-Wallis test with Dunn's post-test. Data are presented as mean values ± SEM. Source data are provided as a Source Data file.

In summary, EV$^{AML}$ showed a mild effect on human leukemia cell lines, mainly decreasing the ROS$^{hi}$ GSH$^{lo}$ fractions in the MOLM-13 and KG-1 cell lines and increasing the frequency of ROS$^{lo}$ GSH$^{hi}$ fractions in MOLM-13 and OCI-AML3. This suggests that EV$^{AML}$ may play a more significant role in the metabolism of specific AML cell subsets.

### EV$^{AML}$ from different AML risks have distinct metabolic effects on OCI-AML3 and MOLM-13, and adverse-risk EV$^{AML}$ enhance the engraftment of MOLM-13 in vivo

We then investigated how circulating EV$^{AML}$ from different ELN risk groups affects the metabolism of human cell lines. Our focus was on the two cell lines, OCI-AML3 and MOLM-13, derived from PB of AML patients and which were more responsive to EV$^{AML}$.

For OCI-AML3, all three types of EV$^{AML}$ were found to significantly increase the proportion of cells with low ROS levels and high GSH content ($p < 0.0001$, respectively). Additionally, only adverse EV$^{AML}$ increased the proportion of ROS low (ROS$^{lo}$) and high mitochondria

potential (MITO$^{hi}$) ($p < 0.05$), while significantly decreasing the proportion of ROS$^{hi}$ and GSH$^{hi}$ ($p < 0.01$) (Fig. 6a–c). For MOLM-13, all three types of EVs were observed to reduce the frequency of cells with high ROS and low GSH (ROS$^{hi}$ GSH$^{lo}$) ($p < 0.0001$ for fav EV$^{AML}$ and $p < 0.01$ for both intermediate and adverse EVs). However, only adverse EV$^{AML}$ significantly decreased the percentage of cells with low GSH and high mitochondria potential (GSH$^{lo}$ MITO$^{hi}$) ($p < 0.05$) (Fig. 6d–f). We therefore examined whether pretreatment with EV$^{AML}$ might enhance resistance to Venetoclax, a selective inhibitor of BCL-2 that has advanced treatment options for AML patients[33]. We performed our experiments on MOLM-13 since OCI-AML3 has been reported as resistant to Venetoclax[34]. Notably, pretreatment with EV$^{AML}$ increased the survival of MOLM-13 cells treated with Venetoclax for 24 h ($p < 0.01$). However, we did not observe a difference when we stratified EV$^{AML}$ according to risk (Fig. 6g).

Finally, we conducted an in vivo study to evaluate the effect of EV$^{AML}$ pre-treatment on MOLM-13 or OCI-AML3 cells xenografted in

NSG mice. We pre-treated Luc-mCherry MOLM-13 and OCI-AML3 cells with EV^AML or with vehicle control (PBS) prior to tail vein infusion and engraftment. Our results showed that pre-treatment with EV^AML from adverse-risk AML patients was more effective in increasing AML xenograft growth compared to treatment with a vehicle control (PBS). Interestingly, after 18 days, the treatment with EV^AML from adverse risk significantly improved the engraftment of Luc-mCherry MOLM-13 cells compared to untreated cells ($p < 0.001$, Fig. 6h, i). However, EV^AML from intermediate and favorable patients did not affect the engraftment of Luc-mCherry MOLM-13 cells (Fig.6h, i). A similar pattern, although characterized by signal dispersion, was observed when OCI-AML3 cell line was used.

According to in vitro experiments, our in vivo findings indicate that EV^AML from adverse-risk AML patients may have an increased capacity for accelerating in vivo leukemia cell engraftment compared to EV^AML collected from intermediate-risk or favorable AML patients.

### EVs from AML patients alter the redox metabolism of CD34+ AML cells, modulating the GSH/GPX4 axis

We cocultured human primary CD34+ cells isolated from AML patients at diagnosis or CB-derived CD34+ AML cells with EV^AML to test the metabolic modulation mediated by EVs exclusively on the hematopoietic cell compartment. Firstly, we tested whether EV^AML can interfere with HSPC differentiation, and we did not find any significant difference in the HSPC subset proportion (Supplementary Fig. 7a, b). Then, as reported above, we developed a gating strategy combining side by side the different staining suitable among the cell lines and human primary cells. Notably, no differences were observed when we cocultured EV^AML with normal CB CD34+ cells regarding redox metabolic markers (Supplementary Fig. 7d, e).

When we cocultered CD34+ AML cells with EV^AML, combining ROS with mitochondrial functionality, we did not find any significant differences (Fig. 7a). However, we found a significant decrease in ROS levels and an increase in MITO levels after EV^AML treatment, as reported by MFI values ($p < 0.05$; Fig. 7d). As a result, within the leukemic CD34+ progenitor compartment, coculture with EV^AML significantly reduced the proportion of ROS^hi GSH^lo cells (mean percentage: 53.91%) compared to that in AML CD34+ cells treated with vehicle (mean percentage: 76.21%) ($p < 0.01$; Fig. 6b), as previously reported in Supplementary Fig. 5e, f for human cell lines (MOLM-13 and KG-1). Moreover, we observed an increase in the frequency of ROS^hi GSH^hi fractions in the presence of EV^AML ($p < 0.01$; Fig. 7b). Again, similar to what we observed for the KG-1 cell line (Supplementary Fig. 5i), we also confirmed in AML CD34+ cells an increase in GSH^hi MITO^hi fractions ($p < 0.002$) and a reduction in GSH^lo MITO^hi fractions ($p < 0.01$; Fig. 7c).

When we analyzed the metabolic effects by sorting the data based on EV^AML from different ELN risk groups, we observed that all three subtypes of EV^AML significantly increased the frequency of CD34+ cells with high levels of GSH and mitochondria (GSH^hi MITO^hi), and significantly decreased the frequency of CD34+ cells with high levels of ROS and low levels of GSH (ROS^hi GSH^lo), as well as the proportion of cells with GSH^lo and MITO^hi (Supplementary Fig. 8b, c). However, only intermediate EV^AML were able to increase the proportion of CD34+ cells with low ROS and high MITO, whereas only favorable EV^AML increased the proportion of CD34+ cells with high ROS and high GSH (Supplementary Fig. 8a, b). Significantly, unlike EVs from intermediate ($p < 0.001$) and adverse-risk patients ($p < 0.01$), EV^AML from favorable AML patients increased the proportion of CD34+ cells with high levels of ROS and high mitochondrial potential (ROS^hi MITO^hi; Fig. 7e) as compared to untreated cells ($p < 0.01$; Supplementary Fig. 8a) and decreased the proportion of CD34+ cells with low levels of ROS and high mitochondrial potential as compared to intermediate ($p < 0.01$) and adverse-risk EV^AML ($p < 0.05$; Fig. 7e). These data suggest a metabolic vulnerability driven by EVs from different ELN risk categories.

Then, we performed gene expression profile analysis to observe whether circulating EVs directly alter the tumor transcriptional program from a metabolic point of view. We observed 15 differentially expressed genes in AML CD34+ cells in the presence of EV^AML, as reported by the volcano plot (Supplementary Fig. 8d). Notably, these results were further corroborated by cBioPortal data, which showed that AML patients (n = 25) with high mRNA expression of the genes upregulated by EV^AML in our cohort (namely, *CAB39, ADA, SLC2A14, CHMP2A, NAT8L, GPX4, CTCF, TPX2, NDUFS8, STAT3,* FANCD2 and *BRCA1*) had decreased OS compared to unaltered AML patients (n = 148) (log-rank, $p < 0.04$) (Supplementary Fig. 8d, e).

Then, based on the GPX3 cargo of EV^AML and GSH/GPX axis modulation driven by EV^AML, we assessed the activity of RSL3, a pharmacological inhibitor resulting in posttranslational degradation of GPX4[35]. Since GPX4 expression is closely associated with GSH levels in antioxidant metabolism[36,37], we tested whether GPX4 inhibition might revert EV^AML-driven effects on AML CD34+ cells. In preliminary experiments, we set up the optimal dose of RSL3 that does not affect cell viability even though it causes GSH depletion. We found that RSL3 significantly reduced the GSH levels detected in AML CD34+ cells ($p < 0.05$; Fig. 7f). Importantly, EV^AML were not able to restore GSH levels after preincubation with RSL3, suggesting EV^AML functions via the GPX system ($p < 0.05$).

Collectively, these results demonstrate that circulating EVs from AML patients may trigger CD34+ AML cells toward an increase in both mitochondrial potential and GSH levels, reducing ROS levels and showing a leukemia-dependent mechanism partially reverted by GPX4 inhibition. Understanding the metabolic dependencies mediated by EVs from different ELN risk groups can allow for the development of tailored therapeutic strategies aimed at exploiting new metabolic vulnerabilities.

### Mitochondrial dependence is increased in CD34+ AML cells cocultured with circulating EV^AML, subverting energetic metabolism

To further assess the metabolic response to EV^AML, we performed Seahorse analysis on both cell lines, i.e., KG-1 and MOLM-13, which were shown to be more responsive to EV^AML, and on primary CD34+ AML cells. The Cell Mito stress assay demonstrated a slight increase in the overall oxygen consumption rate (OCR) in both cell lines tested (Fig. 8a–d). In KG-1 cells, we found an increase in basal respiration in the presence of EV^AML (Fig. 8b), whereas MOLM-13 cells showed a significant increase in the maximal respiration rate after coculture with EV^AML (Fig. 8d). As shown in Fig. 8e, f, mitochondrial respiration, as measured by the OCR, was increased in circulating CD34+ AML cells cocultured with EV^AML compared to that in CD34+ cells cultured with vehicle. Spare respiratory capacity (SRC) showed a 2.5-fold increase in CD34+ cells treated with EV^AML, but the difference did not reach statistical significance. However, maximal respiration was significantly increased in the presence of EV^AML (Fig. 8e, f; $p < 0.05$). Importantly, we applied the SCENITH method to total paired MNCs from PB and BM AML samples resembling their microenvironment after treatment with circulating EV^AML. EV^AML did not affect metabolic parameters such as glucose dependence and FAAO cap (Fig. 8g). Surprisingly, circulating EV^AML shifted the metabolism of PB CD34+ fractions toward higher mitochondrial dependency and lower glycolytic capacity, showing an EV-driven metabolic adaptation that mainly occurs in the leukemic fractions outside of the BM niche (Fig. 8h). Indeed, no effects were reported in BM CD34+ cells from the paired AML patients between mito dep or glyco cap (Supplementary Fig. 8h). Our data showed that BM AML cells had lower uptake compared to PB AML cells from the same patients, which might explain the lower responsiveness of BM CD34+ cells (Supplementary Fig. 8i). Overall, the data suggest that circulating EV^AML influences the metabolic profile of CD34+ AML cells and that

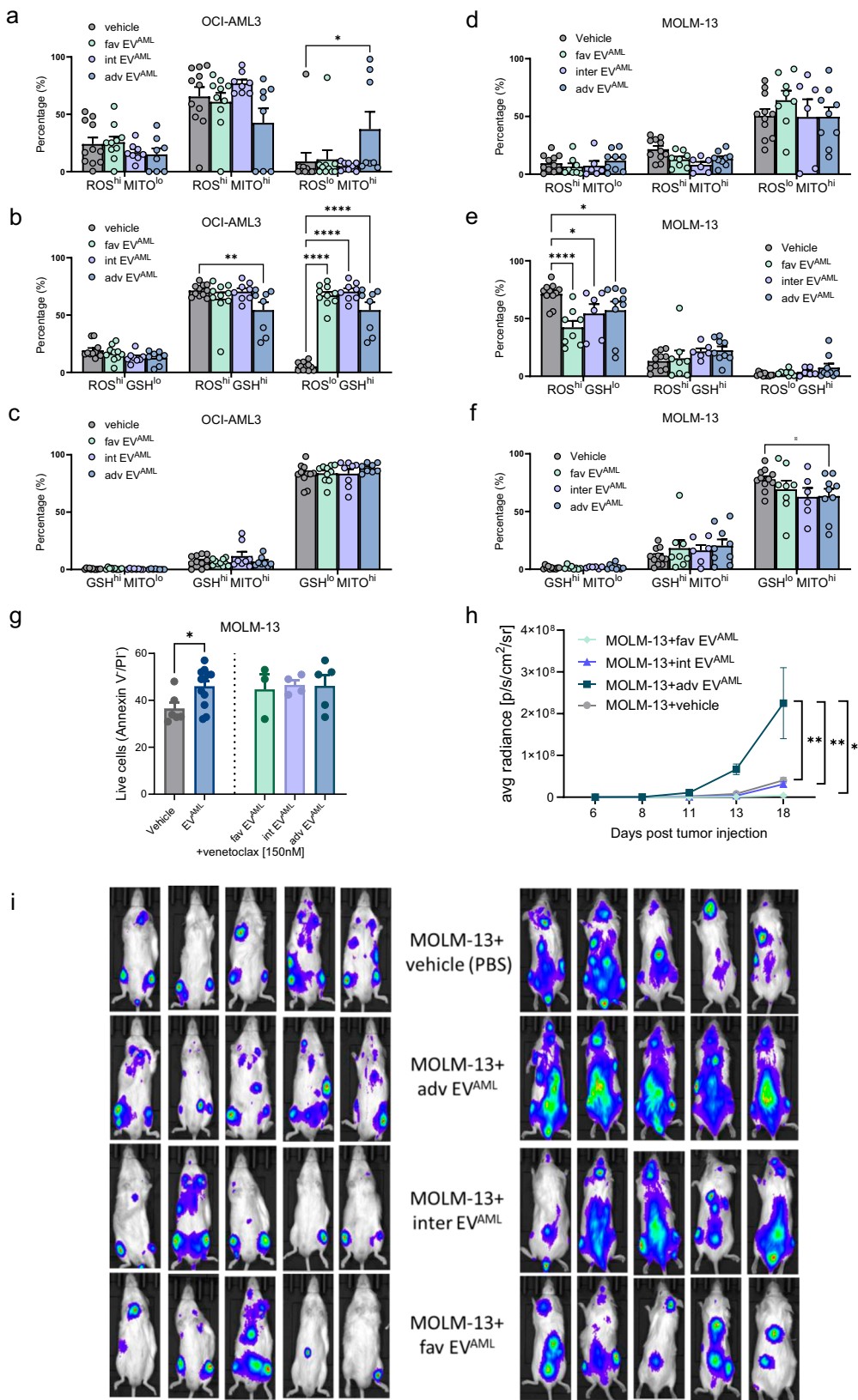

the effects of circulating EV[AML] might subvert the metabolic dependency reported in AML whole blood samples.

## Discussion

Despite the progress in unraveling metabolic rewiring in AML[5,6], there is still much to be learned about exploiting these metabolic features in clinical practice. Here, we demonstrate that a combined flow cytometry-based redox metabolic approach in parallel on AML CD34[+] (CD38[low/-]) cells and EV[AML] is readily applicable in fresh blood and might be exploited to reveal critical metabolic vulnerabilities for prognostic purposes.

By focusing primarily on CD34[+]CD38[low/-] stem and progenitor sub-sets, we provided evidence supporting a consistent and standardized

**Fig. 6 | Redox metabolic profiling for OCI-AML3 and MOLM-13 treated for 24 h with EV^AML from adverse, intermediate or favorable-risk AML patients and in vivo engraftment assay.** OCI-AML3 (**a**, **b**, **c**) or MOLM-13 cell lines (**d**, **e**, **f**) were treated with favorable EV^AML (*n* = 10), intermediate EV^AML (*n* = 8) or adverse EV^AML (*n* = 8) from 8 independent biological experiments. ROS/MITO subsets expressed as percentages were reported for OCI-AML3 (**a**; *p* = 0.03) or MOLM-13 (**d**). Percentage of ROS/GSH subsets for OCI-AML3 (**b**; *p* = 0.002, *p* < 0.0001) and MOLM-13 (**e**). Percentages of GSH/MITO subsets for OCI-AML3 (**c**) and MOLM-13 (**f**; *p* = 0.04). **a**–**f** Two-way ANOVA reported significant differences with Dunnett's multiple comparisons test. **g** MOLM-13 pre-treated with EV^AML for 4 h before adding Venetoclax (150 nM) for 24 h. MOLM-13 were treated with vehicle or favorable EV^AML (*n* = 3) or intermediate EV^AML (*n* = 4) or adverse EV^AML (*n* = 5) from 5 independent

biological experiments. A significant difference was reported using the Mann–Whitney test for unpaired samples (*p* = 0.03). **h**, **i** Mice transplanted with luc-mCherry MOLM-13 cells pre-treated for 24 h with vehicle control (PBS, n = 5), adverse-risk EV^AML (n = 5) or intermediate-risk EV^AML (n = 5) or favorable-risk EV^AML (n = 5). The graph shows the quantification of bioluminescence as the average total photon flux per second from days 6 to 18 after initiation (**h**). Whole-animal bioluminescence imaging from day 18. Ventral (left) and dorsal view images (right) from mice for each group are shown. Each animal's region of interest (ROI) was defined at every time point (inset) (**i**). Two-way ANOVA with Tukey's multiple comparison test. Data are presented as mean values ± SEM. Source data are provided as a Source Data file.

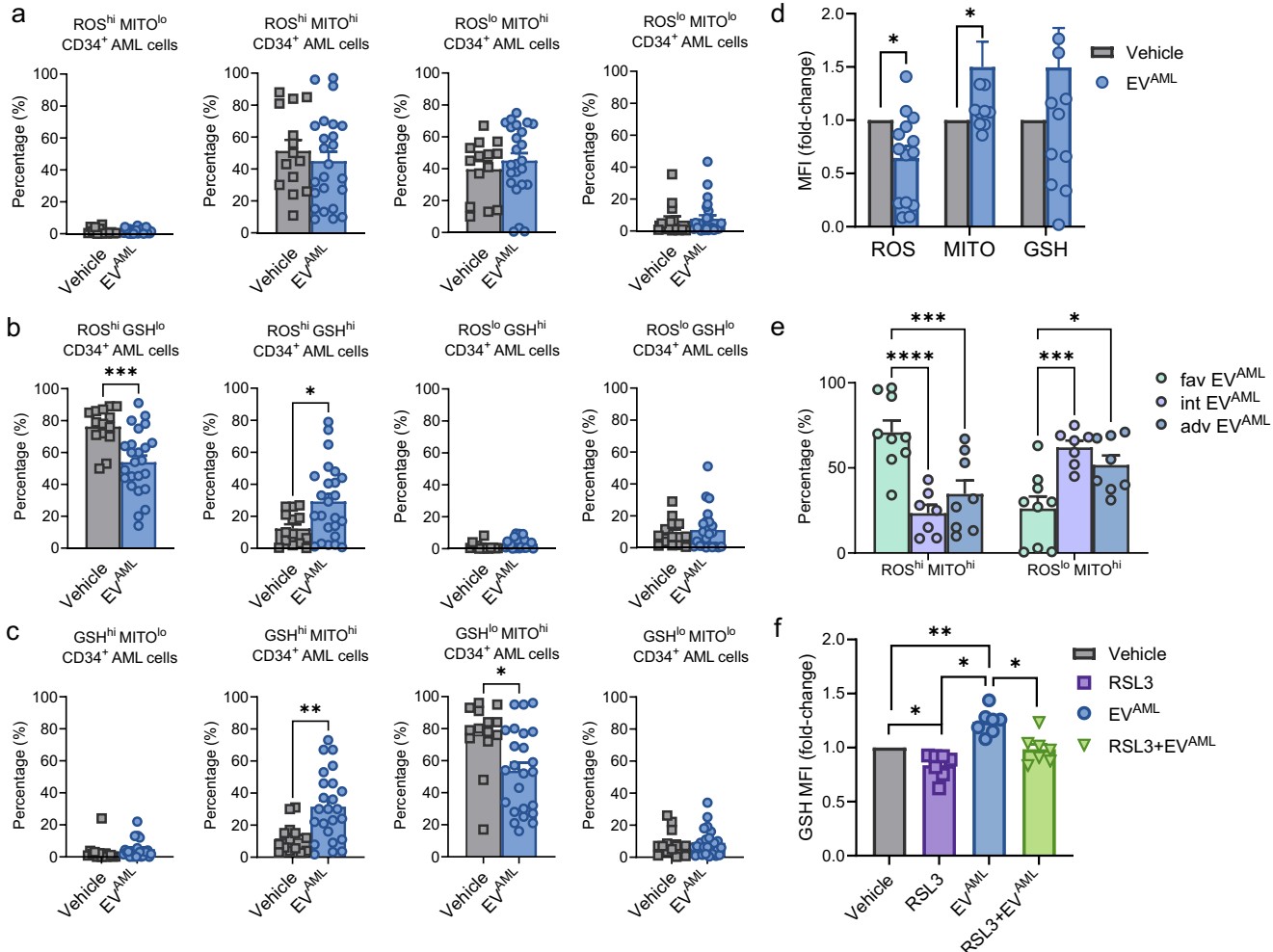

**Fig. 7 | Redox metabolic phenotype of CD34^+ AML cells in coculture with vehicle (PBS) or EV^AML.** Percentages of CD34^+ cells isolated from AML patients at diagnosis in coculture with vehicle (PBS) (*n* = 14) or EV^AML (*n* = 24) for 24 h before staining for **a** ROS/MITO: ROS^hi MITO^lo, ROS^hi MITO^hi, ROS^lo MITO^hi, and ROS^lo MITO^lo subsets; **b** ROS/GSH to determine the ROS^hi GSH^lo (*p* = 0.0006), ROS^hi GSH^hi (*p* = 0.02), ROS^lo GSH^hi, and ROS^lo GSH^lo subsets; **c** GSH/MITO: GSH^hi MITO^lo, GSH^hi MITO^hi (*p* = 0.004), GSH^lo MITO^hi (*p* = 0.01), and GSH^lo MITO^lo subsets. Significant differences were reported as (*) *p* < 0.05, (**) *p* < 0.01, (***) *p* < 0.001 using the Mann–Whitney test for unpaired samples from 14 biological experiments. **d** The expression levels for ROS (*n* = 14; *p* = 0.02), MITO (*n* = 12; *p* = 0.02), and GSH (*n* = 13) in CD34^+ cells treated with EV^AML are expressed as MFI normalized to the MFI of untreated cells used as a control (MFI fold change). Significant differences were reported as (*) *p* < 0.05 using the Mann–Whitney test for unpaired samples.

**e** Percentages of ROS^hi MITO^hi (*p* < 0.0001 and *p* = 0.0004) and ROS^lo MITO^hi (*p* = 0.0007 and *p* = 0.01) for CD34^+ AML cells treated with favorable EV^AML (*n* = 9), intermediate EV^AML (*n* = 7) or adverse EV^AML (*n* = 8) from 10 independent experiments. Two-way ANOVA reported significant differences with Šidák's multiple comparisons test. *P* values < 0.01 (**), <0.001 (***), <0.0001 (****) were considered significant. **f** gMFI for ThiolTracker of AML CD34^+ cells treated for 24 h with vehicle (PBS), RSL3 (1 μM; *p* = 0.02), EV^AML (*p* = 0.0003) or after pre-treatment with RSL3 (1 μM) for 6 h before adding EV^AML (*p* = 0.0004 versus EV^AML). The expression levels for each marker are expressed as gMFI normalized to the MFI of untreated cells used as a control (MFI fold change). One-way ANOVA with Tukey's multiple comparisons (*n* = 7). *P* values < 0.05 (*) and <0.01 (**) were considered significant. Data are presented as mean values ± SEM. Source data are provided as a Source Data file.

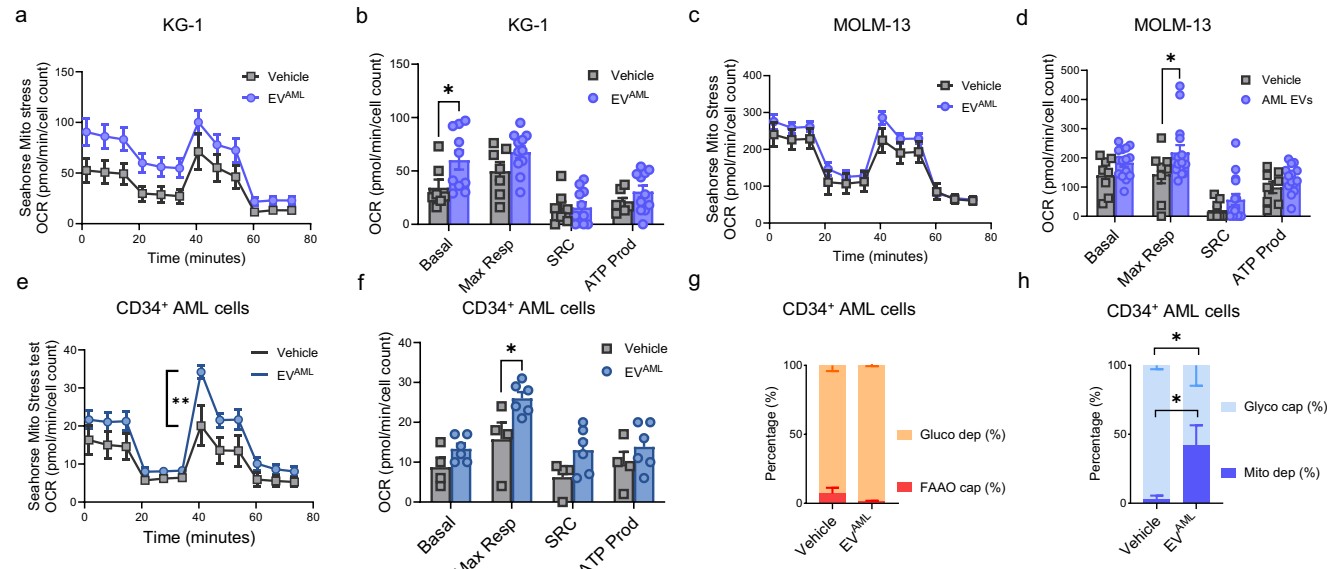

**Fig. 8 | Metabolic studies on leukemic cell lines (KG-1 and MOLM-13) or CD34+ AML cells isolated from AML patients in coculture with vehicle (PBS) or EV^AML for 24 h. a–d** Seahorse XFp Cell Mito Stress profile and analyses of KG-1 (n = 7 vehicle vs. n = 11 with EV^AML) or MOLM-13 cells (n = 8 vehicle vs. n = 14 with EV^AML); **a–c** Oxygen consumption rate (OCR) for KG-1 (**a**) or MOLM-13 (**c**). Bioenergetic parameters extracted from the OCR plot: basal respiration (basal), maximal respiration (Max Resp), spare respiratory capacity (SRC), and ATP-linked OCR (ATP prod) for KG-1 (**b**; p = 0.03) and MOLM-13 cells (**d**; p = 0.04). **e, f** Seahorse XFp Cell Mito Stress profile of AML CD34+ cells and relative bioenergetic parameters extracted from the oxygen consumption rate (OCR) plot (n = 4 independent biological experiments; p = 0,001 and p = 0.02). **g, h** Metabolic profile using SCE-NITH in MNCs cocultured with vehicle (PBS) or EV^AML and then stained for CD34+. Stacked graph with percentages of CD34+ cells using glucose dependence (gluco dep) or FAAO capacity on the left and glycolytic capacity or mitochondrial dependence (percentages) on the right (n = 4 vehicle vs. n = 6 with EV^AML; p = 0.02) from 4 biological experiments. Statistical significance was reported by two-way ANOVA with Sidak's multiple comparisons test or using the Mann–Whitney test for unpaired samples. Data are presented as mean values ± SEM. Source data are provided as a Source Data file.

approach to evaluating metabolic status in liquid biopsy for AML, minimizing the impact of interindividual variations in different marker expressions. Achieving a better understanding of the metabolic profiles of different LSC phenotypes is crucial to identifying new therapeutic targets. Intriguingly, our data from PB are mirrored by paired BM leukemic cells, sustaining the exploitation of liquid biopsy based on redox metabolism for AML.

To explore the metabolic landscape of AML, we used an innovative approach called SCENITH that functionally profiles real-time multicellular metabolism[28]. In contrast to other approaches[6,8], SCE-NITH technology applied to hematological malignancies may provide a live snapshot of the central energy metabolism of AML cell subsets and LSCs without creating metabolic artifacts. This innovative method enables the utilization of fresh whole blood and facilitates processing within just a few hours. In line with previous works[38], our data indicate that AML CD34+(CD38^low/-) cells have enhanced glycolytic activity and altered glucose metabolism that might preferentially promote a resistant phenotype to chemotherapeutic drugs. However, as reported and clarified by DeBerardinis et al.[39], the increase in aerobic glycolysis does not predict the loss of mitochondrial oxidative metabolism. Indeed, our findings are corroborated by higher mitochondrial dependence observed in intermediate-/adverse-risk AML patients or in the BM fractions, suggesting high mitochondrial dependence for the residual LSCs, as previously suggested by others and us[4,5,40]. According to our research, Jayavelu et al.[41] have identified a new high-risk subtype of AML, defined as 'Mito-AML', characterized by high expression of mitochondrial proteins and associated with poor outcomes. Our data support that increased mitochondrial potential or dependence is associated with intermediate and adverse-risk AML. In addition, we reported a high glucose dependence exclusively in intermediate-risk AML patients, revealing that metabolic profiling can potentially predict clinical features.

To date, there is an urgent need for a redefinition of diagnostic and prognostic systems for AML management to improve risk stratification[42,43]. In particular, the intermediate-risk group remains largely undefined because of their heterogeneity and because several patients fall into this group after exclusion from the other two groups[42]. Here, we aim to employ liquid biopsy to uncover distinct redox metabolic profiles in leukemic stem/progenitor subsets and EV signatures in AML patients according to their risk profiles. Additionally, we found sex-associated differences implicated in redox metabolism, with high levels of mitochondrial potential exclusively in female AML patients, providing evidence for further including the sex dimension in developing a new therapeutic strategy for AML. In line with that, liquid biopsy based on metabolic features could improve the current methods to track and characterize AML cells, which are currently based on molecular characterization. Thus, we assessed EV-based liquid biopsy signatures to associate risk categories with the metabolic features detected in the LSC compartment. Since EVs can mirror the parental cell phenotype and cargo[19], the study of circulating EVs might be crucial for identifying resident chemoresistant cells. Importantly, in line with the fact that a highly immature phenotype is associated with more aggressive disease, we unveiled that all favorable-risk AML patients showed an increase in the expression of almost all markers (including CD2, CD8, CD49e, and CD146) in comparison to intermediate- or adverse-risk AML patients. Therefore, our study provides evidence of a prognostic 'index' based on marker levels expressed on EV^AML, similar to the 'stemness index' widely used for cancer stem cell studies[44] or to prognostic gene signatures suggested by Lasry et al.[43].

It has also been reported that EVs can induce phenotypic changes in recipient cells and modify cellular metabolism[45]. Specifically, we observed that EVs isolated from AML patients with favorable, inter-mediate, and adverse prognoses induce metabolic changes that

depend on the recipient cells, whether primary AML CD34[+] cells or cell lines. This suggests that the complexity of the leukemia scenario is influenced not only by the origin of the EVs but also by the intrinsic characteristics of the recipient cells. However, we demonstrated that EV[AML] might be a key mediator of metabolic reprogramming, enhancing the mitochondrial dependence of circulating CD34[+] AML cells and suggesting a more aggressive and chemoresistant phenotype. Indeed, AML cells pre-treated with EV[AML] displayed increased resistance to Venetoclax treatments. This indicates that EV[AML] may directly affect the response to drugs. Additionally, we found that EV[AML] from adverse-risk AML patients enhanced the engraftment of AML cell lines such as MOLM-13, supporting our interests in EV cargo and the EV roles in intracellular metabolism. Thus, the understanding of the complex metabolic cross-talk between leukemic cells and EVs is certainly a key point for a comprehensive characterization of the AML metabolic landscape.

To the best of our knowledge, only a few works have analyzed the lipid profile in AML patients, but they have focused on plasma/serum or cellular components[46–48]. Reflecting the metabolic state of parent cells, lipids are essential components of cells even though they have been poorly investigated[49]. Our in-depth analysis of the EV lipidomic profile may help to identify EV signatures in AML patients, providing insights into the cellular origin or metabolic status. We know that one limitation of our study is that EV phenotype, cargo, and function might be modulated by several variables, including the methods used for isolation. To circumvent some of these hurdles, we tried to adjust and optimize our methods according to the minimal requirements and following previous reports and guidelines[50,51]. Thus, the presence of specific lipid species, even though considered contaminants[52], might reveal relevance in a clinical setting. Here, we demonstrated that quantitative lipidomic analysis of EVs may support risk stratification for AML, as suggested by the depletion of lipid species in adverse-risk patients compared to favorable-risk and intermediate-risk patients. We can speculate that LSCs might be more resistant to oxidative stress induced by drugs due to the induction driven by EV[AML] enriched in monounsaturated lipids, which are less susceptible to lipid peroxidation[53]. Our initial metabolomic study on EV[AML] showed higher concentrations of the metabolite α-ketoglutarate in EV[AML] patients from adverse-risk AML patients. Alpha-ketoglutarate is a crucial intermediate of the TCA cycle and a precursor of glutamine, serving as an antioxidant in various cellular processes and crucial for mitochondrial metabolism[4,54]. Further exploration of the metabolomic cargo of EV[AML], coupled with the identification of lipid species, holds the potential to elucidate the metabolic impacts of EV[AML] on AML cells.

Indeed, the enrichment of specific lipids in EV[AML] can also be justified by the involvement of antioxidant defense[55], as suggested by the abundance of certain alkyl-phosphatidylcholines (PC-Os) in EV[AML] or by the presence of GPX3. According to our previous work[4], Vignon et al.[30] showed that activation of the antioxidant enzyme GPX3 in AML cells cocultured with BMSCs leads to decreased ROS levels. Interestingly, GPX3 EVs may predict the response to therapy in advanced hepatocellular carcinoma[56], further supporting the involvement and clinical relevance of factors related to redox metabolism detected in EVs. Accordingly, we demonstrated the involvement of the GSH/GPX4 axis in EV[AML] function. GSH is a cofactor of selenoenzyme GPX4, which regulates lipid peroxidation of cell membranes during increased oxidative stress, and both factors are involved in ferroptosis[36,37]. In our system with CD34[+] AML cells, we identified GPX4 as a crucial mediator of the GSH increase after EV[AML] treatment. Our work might suggest GPX4 inhibition as a promising strategy for targeting LSCs.

Although our findings need to be confirmed in future studies powered by adequate sample sizes and across AML subtypes with different molecular statuses, our study highlights a connection between leukemic redox metabolism and EV signatures. This link may have a potential prognostic relevance for AML. An approach based on

liquid "cell-EV" biopsy has the potential to further decipher the metabolic landscape in AML, leading to a redefined metabolic AML stratification.

## Methods

### Human patient samples and cell lines
The research was approved by the institutional review board of the Area Vasta Emilia Centro (AVEC) Ethical Committee (approval code: 94/2016/O/Tess). Patients with AML and healthy donors (HD) were recruited at IRCCS Azienda Ospedaliero-Universitaria, Seràgnoli Hematology Institute in Bologna. Clinical samples and data were collected after written informed consent was obtained. According to normal clinical practice, both PB samples ($n = 114$) and paired BM aspirates were collected from AML patients at diagnosis. The demographic and clinical findings of the AML patients are summarized in Supplementary Table 1 and Source Data File. PB samples were also collected from sex/age-matched HDs upon signed informed consent ($n = 30$). The gene expression profiles of AML patients were collected from the TCGA[57] database through cBioPortal (https://www.cbioportal.org/). Thus, our case set was a group of 173 AML cases with mRNA data (RNA-seq V2) from 200 AML patients. Cord blood units were obtained and used to obtain normal HSCs. The human AML cell lines MOLM-13, OCI-AML3, KG-1, Kasumi-1 and MV-4-11 were purchased from DSMZ (Braunschweig, Germany). Cell lines were grown in a humidified incubator at 37 °C and 5% $CO_2$, identified by short tandem repeat profiling and tested for mycoplasma.

### Blood sample collection and plasma preparation
After collection, venous ethylenediaminetetraacetic acid (EDTA)-blood tubes (BD Vacutainer®) were kept at room temperature (RT). Within 2 h, the blood samples were centrifuged at RT for 10 min (min) at $1000 \times g$, and the supernatants were then transferred to a new tube for two consecutive centrifugations at $2500 \times g$ for 15 min at RT to obtain platelet-free plasma. The plasma samples were then aliquoted and stored at −80 °C until use.

PB was centrifuged over a Ficoll-Paque gradient (Lympholyte CL5020). Mononuclear cells (MNCs) were collected and resuspended in RPMI 1640 medium (Euroclone) supplemented with 20% Gibco™ exosome-depleted fetal bovine serum (FBS) (Thermo Fisher Scientific), 2 mmol/L L-glutamine (Thermo Fisher Scientific), 500 IU/mL penicillin, and 0.5 mg/mL streptomycin (penicillin–streptomycin, MP Biomedicals, USA) and frozen for further experiments. CD34[+] cells (mean purity > 90%) were purified from AML MNCs and from cord blood units by immunomagnetic separation (Miltenyi Biotec,) according to the manufacturer's recommendations.

### EV isolation
Plasma samples were defrosted at RT, and EV isolation was achieved by size-exclusion chromatography (SEC; qEVoriginal 70 nm column, Izon Science) following the manufacturer's instructions. Briefly, the column was equilibrated with PBS before loading plasma samples (500 µl) on top of the column. Next, four fractions were collected after void volume. EV-enriched fractions were pooled and concentrated by ultrafiltration using Amicon Ultra-2 30 K centrifugal filters (Merck Life Science S.r.l.) with centrifugation at $4000 \times g$ for 40 min and then at $1000 \times g$ for 2 min before use or stored at −80 °C until use. Isolated EVs were analyzed by nanoparticle tracking analysis (NTA) using the NanoSight LM10 system (NanoSight Ltd.) equipped with a 405 nm laser and NTA 2.3 analytic software. The protein content of the fractions was determined using the Bradford assay (Bio-Rad) according to the manufacturer's instructions.

### Coculture experiments with EV[AML]
To study the effects of EV[AML] on AML/CB CD34[+] cells and human AML cell lines (MOLM-13, KG-1, OCI-AML3, Kasumi-1, and MV-4-11), we set up

cocultures using EV concentrations ranging from 2 to 10 µg, which we preliminarily demonstrated to have no effects on cell viability. Cell lines were cultured under standard conditions in 20% Gibco™ exosome-depleted FBS (Thermo Fisher Scientific). CD34+ cells were seeded at $1 \times 10^6$ cells/ml in the presence of cytokines such as interleukin (IL)-3 (50 ng/ml, PeproTech) and stem cell factor (SCF) (50 ng/ml; Miltenyi Biotec). Where indicated, MNCs were cocultured with EV$^{AML}$ for 24 h and stained with CD45, CD34, and CD38 before performing experiments. For GPX4 inhibition, cells were treated with (1S,3 R)-RSL3 (RSL3) at 1 µM for 6 h before adding EV$^{AML}$. RSL3 was purchased from MedChemExpress and dissolved in DMSO. After 24 h, the cells were collected, counted, and stained for the analyses described below. In selected experiments, Venetoclax (ABT-199; #6960, Tocris Bioscience) was added to MOLM-13 for 24 h at 150 nM after pre-treatment with vehicle or EV$^{AML}$ for 4 h.

### Seahorse metabolic extracellular flux profiling

CD34+ AML cells or human leukemia cell lines (MOLM-13 and KG-1) cocultured with vehicle (phosphate-buffered saline, PBS without EVs) or EV$^{AML}$ were seeded in 24-well plates (-500.000 cells/well). The Cell MitoStress Test (XFp Cell Mito Stress Test Kit, Agilent Technologies, Santa Clara, CA, USA) was performed following the standard protocol. The oxygen consumption rate (OCR) and extracellular acidification rate (ECAR) were detected after injection of oligomycin (O, 1.5 µM), carbonyl cyanide-p-trifluoromethoxyphenylhydrazone (FCCP, 0.5 µM), and the combination of rotenone & antimycin (Rot/AA, 0.5 µM). The OCR and ECAR were measured with an XFp analyzer and analyzed using Agilent Seahorse Analytics software after normalization to cell counts.

### Flow cytometry analysis and fluorescence-activated cell sorting (FACS)

Fresh blood (500 µl) or cell suspensions (AML/CB CD34+ or cell lines), after cocultures, were stained with CellROX Green for ROS levels (ROS: 0.8 µM), MitoTracker™ Red CMXRos for mitochondria potential/functionality (MITO: 50 nM), and ThiolTracker™ Violet as glutathione detection reagent (GSH: 16 µM) according to the manufacturer's instructions (all from Thermo Fisher Scientific) with minor modifications. Then, at least $1 \times 10^5$ cells were incubated for 30 min in a 37 °C cell culture incubator and washed. Fresh whole blood, after lysis with BD FACS™ Lysing solution (BD Biosciences, San Jose, CA, USA), and cells from cultures were stained with the following monoclonal antibodies: CD45-PE (clone: HI30; BD Biosciences, San Jose, CA, USA), CD34-APC (clone 8G12; BD Biosciences), CD38-PE/cyanine7 (clone HIT2; BioLegend) and CD3-Alexa Fluor 700 (clone SK7; BioLegend) following gating strategy 1–2 reported in Supplementary Fig. 1a–e. In selected experiments, we used a panel with the following antibodies: CD34-APC (clone 8G12; BD Biosciences), CD33-BV650 (clone: WM53, Sony Biotechnology), CD117-BV605 (clone: 104D2, Sony Biotechnology), CD45-BV750 (clone: HI30, Sony Biotechnology), HLA-DR APC/H7 (clone: L243, BD™), CD123 PE (clone: 9F5; BD™). Cells were incubated with antibodies at a 1:100 dilution, or as specified otherwise, and were stained at RT for 15 min, washed, and analyzed using CytoFLEX and Kaluza software version 2.1 (Beckman Coulter, Milan, Italy) or using the Sony ID7000™ Spectral Cell Analyzer equipped with 5 lasers (355 nm, 405 nm, 488 nm, 561 nm, 637 nm). To overcome the heterogeneity among AML patients, the gating strategy was standardized and built considering the staining of CD3+ lymphocytes within each AML sample as reported in gating strategies (Supplementary Fig. 1a–e). In a set of experiments, PB CD34+ AML cells that had been previously purified were thawed and prepared for FACS sorting. The CD34+ AML cells were stained with DAPI (Sigma–Aldrich, Milan, Italy; 1:1000) to exclude dead cells and were stained with a two-by-two combination of ROS, MITO, and GSH dyes as specified above. Sorting was performed using a 100 µm nozzle with a pressure of 20 PSI. A BD FACSAria™ Fusion

Special Order (SORP) cell sorter cytometer (BD Biosciences) equipped with three lasers was used (405 nm, 488 nm, and 640 nm). Sorted fractions are shown in Supplementary Fig. 1k–m. After completing the sorter, cells were centrifuged, resuspended in RPMI medium, counted, and plated in clonogenic assays.

### Clonogenic assays

CD34+ AML cells (1000 cells/plate) were cultured at 37 °C and 5% CO$_2$ in 35-mm dishes in methylcellulose-based medium (human StemMACS HSC-CFU lite w/ Epo, Miltenyi Biotech). After 10 days of incubation, colony-forming unit (CFU) growth was evaluated by standard morphologic criteria using an inverted microscope (Axiovert 40, Zeiss).

### EV uptake assay

The PKH67 Green fluorescent cell linker (Sigma–Aldrich, Milan, Italy) was used to label the EV membrane. The EV$^{AML}$ were labeled with dye (1:80) for 5 min in Diluent C. After adding 1% bovine serum albumin (BSA) to quench the EV$^{AML}$, the samples were washed with serum-free media, and the redundant dye was removed by ultrafiltration twice with Ultra-15 centrifugal filters and Amicon Ultra-2 (Merck Life Science S.r.l., Milan, Italy) for 40 min. Then, PKH67-labeled EV$^{AML}$ were added to the cell culture with AML CD34+ cells or leukemia cell lines for up to 24 h. A tube containing only PBS and the dye was used as a negative control. EV$^{AML}$ stained with the dye or negative control were used to treat CD34+ cells or human cell lines for 4 h at 37 °C or 4 °C. After coculturing, the uptake of EVs was detected using a CytoFLEX flow cytometer.

### RNA isolation and NanoString analysis

After coculturing with vehicle or EV$^{AML}$, isolated CD34+ cells were collected and lysed using TRIzol (Thermo Fisher Scientific), and RNA was extracted in phenol-chloroform for NanoString analysis using the nCounter Human Metabolic Panel. Analysis of detected gene counts was performed by nSolver Analysis Software 4.0 (NanoString Technologies). After the imaging quality check, raw gene counts were normalized to technical controls and 4 housekeeping genes with the lowest coefficient of variation among the ones included in the panel. After completion of normalization processes, counts were log2 transformed, and a build ratio analysis was performed by comparing the expression profiles of EV$^{AML}$-treated CD34+ cells and control (vehicle) CD34+ cells. Bioinformatic analyses on gene expression profiling (GEP) were conducted by R Software v4.0.4 using the following R packages: ggplot2 and topGO.

### SCENITH staining on fresh blood or leukemic cells and data acquisition

SCENITH™ was performed as described in ref. 28. The SCENITH™ reagent kit (inhibitors, puromycin, and antibodies) was obtained from www.SCENITH.com/try-it and used according to the provided protocol for fresh whole blood or in vitro cells (CD34+ cells or MNCs). Briefly, fresh whole blood was treated for 40 min with control (Co), 2-deoxy-glucose (2-DG; 100 mM), oligomycin (O; 1 µM), a combination of 2DG and oligomycin (DGO), harringtonine (H; 2 µg/mL) and puromycin (final concentration 10 µg/mL) for 40 min. For MNCs ($1 \times 10^6$) or CD34+ cells ($2 \times 10^5$), harvested at desired time points after cocultures, were treated for 15 min with control (Co), 2-deoxy-glucose (2-DG; 100 mM), oligomycin (O; 1 µM), a combination of 2DG and oligomycin (DGO) or harringtonine (H; 2 µg/mL) before adding puromycin (final concentration 10 µg/mL; clone: R4743L-E8) for other 30 min. After puromycin treatment, the blood or cells were washed in cold PBS and stained with eBioscience™ Fixable Viability Dye eFluor™ 780 (1:1000; Thermo Fisher Scientific) for 15 min at 4 °C in PBS. Primary antibodies against surface markers as reported above, including CD34-APC and CD38-PE, were incubated for 25 min at 4 °C in staining buffer (5% FBS in PBS). In selected experiments, we used the following antibodies: CD34-

BUV395 (clone: 563; BD Biosciences), CD33-BUV563 (clone: WM53; BD Biosciences), CD45-BUV805 (clone: HI30; BD Biosciences), CD38-BV711 (clone: HIT2; Biolegend), CD117 cFluor BYG667 (clone: 104D2; Cytek Biosciences), HLA-DR-PE-Fire 810 (clone: L243; Biolegend), and CD123-cFluor R720 (clone: 6H6; Cytek Biosciences), at a dilution of 1:50. Cells were fixed and permeabilized using the Foxp3 Transcription Factor Staining Buffer Set (Thermo Fisher Scientific, Waltham, MA, USA) according to the manufacturer's instructions. Intracellular staining of puromycin (10 μg/ml) was performed for 1 h in diluted (10×) permeabilization buffer at 4 °C. Finally, data acquisition was performed using the CytoFLEX flow cytometer or 5 L Cytek Aurora spectral cytometer (Cytek Bioscience). Unstained cell controls used for autofluorescence extraction were generated for each time point, culture condition (control), and metabolic inhibitor treatment (C, 2DG, O, DGO). Geometrical mean fluorescence intensity (gMFI) expression values were imported from the population of interest and used for calculations to derive SCENITH parameters as follows:

[Co = gMFI of anti-Puro-Fluorochrome upon Co treatment]
[2-DG = gMFI of anti-Puro-Fluorochrome upon 2-DG treatment]
[O = gMFI of anti-Puro-Fluorochrome upon O treatment]
[DGO = gMFI of anti-Puro-Fluorochrome upon 2DG + O (DGO) treatment]
[Glucose dependence (gluco dep) = 100(Co − 2DG)/(Co-DGO)]
[Mitochondrial dependence (mito dep) = 100(Co − O)/(Co-DGO)]
[Glycolytic Capacity (glycol cap) = 100 − mito dep]
[Fatty acid and AA oxidation capacity (FAAO cap) = 100 − gluco dep]

### Lipid extraction and liquid chromatography quadrupole time-of-flight mass spectrometry (LC/MS QTOF) analysis

Lipid extraction was performed on EV samples using the method described by ref. 58 with minor modifications. First, a final volume of the methanol/methylterbutyl ether/chloroform extraction mixture was prepared by adding standard Splash Lipidomix (Avanti Polar). Subsequently, each EV sample was spiked with 600 μL of fresh-prepared MMC extraction mixture. After vortexing for 30 s, the samples were placed in a T-Shaker (Euroclone) and processed for 20 min at 1500 rpm at 20 °C. Afterward, the samples were centrifuged for 10 min at 16000 × g at 4 °C, and the supernatant was transferred to an auto-sampler vial and dried with a gentle stream of N2. The residue was resuspended in 200 μL of a 9:1 methanol/toluene solution and immediately subjected to LC/MS analysis.

LC/MS analyses were carried out using the validated analytical method described by ref. 59, adapted to the different instrumental configurations. In brief, LC/MS QTOF analysis was performed using a 1260 Infinity II LC System coupled with an Agilent 6530 Q-TOF spectrometer (Agilent Technologies, Santa Clara, CA, USA). Spectrometric data (MS) were acquired in the 40–1700 m/z range in both negative and positive polarity. An iterative MS/MS acquisition mode (DDA) on the pooled sample was used. Raw data were processed using MS-DIAL software (4.48)[60] to perform peak-picking, alignment, lipid annotation, quantification and polarity amalgamation. Lipid annotation at the molecular species level was carried out according to the recommendations of the Lipid Standard Initiative[61] using MS and MS/MS acquired data. The acceptance criteria for identifying lipids were as follows: (1) The variance between theoretical MS1 and experimental MS1 must be less than 25 ppm. (2) The total score assigned by the MS-DIAL annotation software must exceed 80%.

A data matrix containing the concentration in nmol/mL was processed with LipidOne[62] for in-depth analysis of lipid building blocks, the MetaboAnalist 5.0 web platform[63] for multivariate statistical, chemoinformatic analysis, and BioPAN[64] for performing pathway analysis. For the network analysis, the diagrams were generated using the Graph Editor tool (https://csacademy.com/app/graph_editor/, accessed on 10 January 2023) and included a statistical reinterpretation of data from LipidOne. Network transformations were analyzed by assessing the ratios of product to reactant masses for each sample. To determine which biochemical reactions were enhanced or inhibited, we compared the mean mass ratios between EV$^{AML}$ and EV$^{HD}$, following the methodology outlined by A. Nguyen et al. [65]. The volcano plot and network graphs were created with Excel (Microsoft) by processing the data obtained with LipidOne.

### Multiplex EV surface marker analysis

MACSPlex analysis was performed using the MACSPlex Exosome Kit, human (Miltenyi Biotec, Bergisch Gladbach, Germany), which detects 37 exosomal surface epitopes plus two isotype controls, according to manufacturer's instructions. Briefly, after overnight incubation at RT of isolated EVs diluted in MACSPlex buffer with MACSPlex exosome capture beads, MACSPlex Exosome Detection Reagent for CD9, CD63, and CD81 was added to each tube, followed by incubation for 1 h at RT. Flow cytometric analysis was carried out on a CytoFLEX flow cytometer and analyzed by Kaluza Analysis 2.1. For analysis, median fluorescence intensity (MFI, APC) was evaluated for each capture bead subset and corrected by subtracting the respective MFI of the blank control.

### Transmission electron microscopy (TEM)

TEM was performed on EVs isolated by SEC and left to adhere to a 400-mesh holey film grid for 20 min. The grids were then incubated with 2.5% glutaraldehyde containing 2% sucrose and stained with 2% uranyl acetate (for 2 min). The samples were observed using a Tecnai G2 transmission electron microscope (FEI Company). Images were captured with a Veleta digital camera (Olympus Soft Imaging System).

### Western blot analysis

EV protein extracts were separated by sodium dodecyl sulfate–polyacrylamide gel electrophoresis (SDS–PAGE, Bio-Rad) and transferred onto nitrocellulose membranes. Membranes were incubated overnight with the following antibodies: mouse anti-ADP-ribosylation factor 6 (ARF6; clone: 3A-1, 1:1000; Santa Cruz Biotechnology), mouse anti-tumor susceptibility gene 101 (TSG101; clone: C-2, 1:1000; Santa Cruz Biotechnology), mouse anti-CD81 (clone: B-11, 1:1000; Santa Cruz Biotechnology), mouse anti-apoA-I (clone: B-10, 1:1000; Santa Cruz Biotechnology, Dallas, Texas, USA), rabbit anti-flotillin (clone: EPR6041, 1:1000; Abcam), rabbit anti-Calnexin antibody (clone: EPR3633(2), 1:1000; Abcam), rabbit anti-HSA (MA5-29022, 1:1000; Invitrogen) and rabbit anti-glutathione peroxidase GPX3 (PA5-119141, 1:500; Thermo Fischer Scientific, Waltham). Horseradish peroxidase (HRP)-conjugated anti-rabbit immunoglobulin G (1:10000; GE Healthcare, Amersham, UK) or anti-mouse IgG (1:10000; GE Healthcare) were used as secondary antibodies. An enhanced chemiluminescence prime (ECL™) reaction kit (GE Healthcare) was utilized for detection by a ChemiDoc XRS+ System (Bio-Rad), and ImageJ software was applied to perform signal quantification.

### Measurement of TCA cycle intermediates by Gas chromatography (GC)-triple quadrupole Mass Spectrometry GC-QQQ-MS in EVs

Analysis of the TCA cycle intermediates was performed by GC-QQQ-MS at Swedish Metabolomics Centre. For Gas Chromatography–Mass Spectrometry (GC-MS) analysis an 11-point calibration curve (cis-aconitic acid, a-keto-glutaric acid, citric acid, fumaric acid, glucose, glucose 6-phosphate, isocitric acid, lactic acid, malic acid, shikimic acid, succinic acid, sucrose, urea) spanning from 25–2500 pg/μL was prepared by serial dilutions and spiked with internal standards (Fumaric acid (13C4), L-Malic acid (13C4), D-Glucose (13C6), α-ketoglutaric acid (13C4), Succinic acid (D4), citric acid (D4), Sucrose (13C12) at a final concentration of 350 pg/μL. EV isolated following our protocols were diluted in 80% methanol (1:10), vortex, and then centrifuged for 10 min at 4 °C at 18,000 × g. Supernatants were frozen until use. Up to 200 μl of the

extracted sample were transferred to GC vials, and the above-mentioned isotopically labeled internal standards were added. The sample was evaporated to dryness under a stream of nitrogen. Derivatization was performed according to Gullberg et al. [66]. In detail, 10 µL of methoxyamine (15 µg/µL in pyridine) was added to the dry sample that was shaken vigorously for 10 min before left to react in room temperature. After 16 h, 20 µl of a mixture of MSTFA and methyl stearate (1050 pg/µL in heptane) (1:1) was added and the sample was vortexed. One µL of the derivatized sample was injected by an Agilent 7693 autosampler, in splitless mode into an Agilent 7890 A gas chromatograph equipped with a multimode inlet (MMI) and 10 m × 0.18 mm fused silica capillary column with a chemically bonded 0.18 µm DB 5-MS UI stationary phase (J&W Scientific). The injector temperature was 260 °C. The carrier gas flow rate through the column was 1 ml min-1, the column temperature was held at 70 °C for 2 min, then increased by 40 °C min$^{-1}$ to 320 °C and held there for 2 min. The column effluent is introduced into the electron impact (EI) ion source of an Agilent 7000 C QQQ mass spectrometer. The thermal AUX 2 (transfer line) and the ion source temperatures were 250 °C and 230 °C, respectively. Ions were generated by a 70 eV electron beam at an emission current of 35 µA and analyzed in dMRM-mode. The solvent delay was set to 2 min. For a list of MRM transitions see Supplementary Table 5.

Data were processed using MassHunter Qualitative Analysis and Quantitative Analysis (QqQ; Agilent Technologies, Atlanta, GA, USA) and Excel (Microsoft, Redmond, Washington, USA) software. Methyl stearate was used as internal standard for compounds lacking a stable isotope internal standard.

### Flow cytometric analysis of CB CD34$^+$ cells after EV treatment for HSPC subsets

After treatment with EV$^{AML}$ for 24 h, CB CD34+ cells were washed and incubated with the following antibodies: CD90 FITC (clone: 5E10; BD Pharmingen™), CD45RA BV510 (clone: HI100; Biolegend), CD34 Pe/Cy7 (clone: 4H11; eBioscience™), CD38 APC (clone: HIT2; Biolegend), CD10 APC-H7 (clone HI10a; BD™), CD123 PE (clone: 9F5; BD™). All the antibodies were diluted at 1:100 and incubated at room temperature for 15 min in the dark following gating strategy 3 reported in Supplementary Fig. 7a and by Trino et al. [67]. After incubation, cells were washed and resuspended in staining buffer. Then, 10,000 events were acquired on CytoFLEX and analyzed using Kaluza software version 2.1 (Beckman Coulter, Milan, Italy).

### Human leukemia cell line transduction

To label the cells for in vivo transplantation, the OCI-AML3 and MOLM-13 cell lines were stably transduced with a lentiviral vector expressing stably luciferase and mCherry, MI-Luciferase-IRES-mCherry (w168-1; Addgene plasmid, #75020), carrying mCherry as a selective marker. To infect human AML cell lines, 293T were cotransfected with 10 µg of lentiviral vector, 3 µg PMD2G envelope plasmid, 2.5 µg of REV packaging plasmid, and 5 µg of RRE transfer plasmid using calcium phosphate transfection system. After 16 h, the media was removed and replaced with 5 mL of fresh medium. Viral supernatant was then collected at 24 and 48 h and added directly to human AML cell lines plated at 1 × 10$^5$ supplemented with 8 µg/mL of polybrene (Merck). Spin-infection was performed for 1.30 h at 37 °C twice daily for one separate day. After 48 h from the first infection cycle, infected cells were sorted using FACS Fusion II (BD Bioscience).

### In vivo experiments

Experiments involving animals were approved by the Italian Ministry of Health and have been done in accordance with the applicable Italian laws (D.L.vo 26/14 and following amendments), the Institutional Animal Care and Use Committee, and the institutional guidelines at the European Institute of Oncology. In this study, given the higher incidence of AML and better engraftment efficiency observed in male cohorts, male mice were prioritized to ensure consistency across experiments. Mice were housed in specific pathogen-free facilities in individually ventilated cages, maintained under 12 h light-dark cycles, controlled temperature (19–23 °C) and humidity (55 ± 10%) with free access to standard rodent chow. To determine the in vivo effects of human cell lines treated with EV$^{AML}$ on leukemia progression and engraftment, tandem tagged luciferase-mCherry human cell lines (OCI-AML3 and MOLM-13) were seeded in a 24-well plate (20 × 10$^4$) and treated with EV$^{AML}$ (15 µg) from AML patients stratified according to ELN2022 risk (n = 5 for each risk group, respectively) or PBS/vehicle as control (n = 5). After 24 h, human cell lines were washed and counted before transplantation. Luc-mCherry AML cells (1 × 10$^5$) treated with EV$^{AML}$ or vehicle control (PBS) were injected with a 26-gauge needle into the lateral tail vein of 6–8-week-old male NOD-scid IL2R gamma null (NSG) mice (n = 5/group from two independent experiments). Mice were randomly assigned to different condition cohorts. All the mice were monitored twice a week and imaged using bioluminescence imaging to document engraftment. Animals were anesthetized for imaging in vivo using 2% Isofluorine (flow rate 1 L/min O$_2$). Spectral imaging was commenced 10 min post IP injection to allow stabilization of light output. Mice were injected with 150 mg/kg of D-Luciferin and imaged twice a week to monitor disease status and treatment efficacy. As a surrogate for tumor burden, bioluminescence was quantified using Living Image Software (version 4.7.2, PerkinElmer). Mice were euthanized when showing any signs of illness or distress (at a maximum radiance of 10$^{11}$ photons per second per cm$^2$ per steradian) and were sacrificed using CO$_2$ chambers.

### Statistical analysis

The mean and standard error (SEM) of each treatment group were calculated for all experiments. Unpaired Student's t tests or ANOVA tests were performed using GraphPad 9.3.1. Where specified, statistics were performed using 2-way ANOVA or the nonparametric Mann–Whitney test (* $p < 0.05$, ** $p < 0.01$, *** $p < 0.001$, **** $p < 0.0001$). Pearson and Spearman's correlations were applied as appropriate. No data were excluded from the analyses. Time-to-event overall survival (OS) and progression-free survival (PFS) endpoints were assessed following the Kaplan–Meier methods and relative survival was provided. Univariate and multivariable semiparametric Cox models were produced for quantitative analysis and hazard ratio (HR) estimates. All estimates objected to inferential analysis were reported with their relative 95% confidence intervals.

### Reporting summary

Further information on research design is available in the Nature Portfolio Reporting Summary linked to this article.

## Data availability

The publicly available datasets used in this study can be accessed at cBioPortal for Cancer Genomics: CAB39, ADA and 10 other genes in Acute Myeloid Leukemia (TCGA, NEJM 2013). The gene expression profile data generated in this study are available in the Gene Expression Omnibus (GEO) repository at https://www.ncbi.nlm.nih.gov/geo/ under accession number GSE245810. All relevant experimental data regarding the isolation protocol and general characterization have been submitted to the EV-TRACK knowledgebase (EV-TRACK ID: EV230004) EV-TRACK (evtrack.org). Lipidomic and metabolomic data on EVs have been deposited to the EMBL-EBI MetaboLights database[68] (https://doi.org/10.1093/nar/gkad1045, PMID:37971328) with the identifier MTBLS11523 for lipidomics (https://www.ebi.ac.uk/metabolights/MTBLS11523) and MTBLS11746 for metabolomics (https://www.ebi.ac.uk/metabolights/MTBLS11746). The protein surface data on EVs generated in this study are available in Zenodo (Zenodo) under accession code https://doi.org/10.5281/zenodo.14017145. Source data are provided with this paper. The remaining

data are available within the Article, Supplementary Information or Source Data file. Source data are provided with this paper.

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

## Acknowledgements

APC fees were funded by the Italian Ministry of Health (RC-2023-2778890). The project was supported by funding from Associazione Italiana contro le leucemie-linfomi e mieloma (AIL) through the "Lascito R. Benvenuti", Bologna AIL and Fondazione Terzo Pilastro Internazionale. D.F. was supported by Società Italiana di Ematologia (SIE) e Associazione "Amici di Beat Leukemia Dr. Alessandro Cevenini ONLUS", by Fondazione Umberto Veronesi and FATRO/Foundation Corrado and Bruno Maria Zaini-Bologna. A.C.([6]) (AIRC IG 2017 id. 20654) and S.B. (AIRC Fellowship for Italy) were supported by Associazione Italiana per la Ricerca sul Cancro (AIRC). F.B. was supported by Italian Ministry of Health (Ricerca Corrente e 5xmille). We acknowledge all AML patients and their families, as well as our current and former colleagues from our Institution. We also acknowledge Dr. Giulia Corradi and Andrea Polimeno for their efforts and support in preparing the crowdfunding campaign related to this project. We sincerely thank the numerous donors of the "EVviva la Ricerca" (IdeaGinger) crowdfunding campaign for allowing us to buy scientific instruments for EV studies. Swedish Metabolomics Centre, Umeå, Sweden (www.swedishmetabolomics centre.se) is acknowledged for measuring TCA cycle intermediates by GC-QQQ-MS. We thank Dr. Luca Dozza for biostatistical analysis, Diatech Lab Line S.r.l. for supporting reagents for Nanostring analysis, and DiBio imaging Facility (University of Padua) for TEM analysis.

## Author contributions

D.F. collected, annotated, and analyzed all the laboratory and clinical data, performed the experiments in vitro, interpreted the results, and wrote the manuscript. R.M.P., H.B.R., S.B., L.U., and C.E. provided lipidomic data and analyzed and edited the manuscript. P.F., D.L., and F.B. provided mouse models and performed in vivo and post-sorter experiments. S.R. performed sorter of leukemic fractions. F.B. and I.V. provided data analysis with NTA. F.P., F.M., and S.B. performed WB analysis. M.B. provided flow-cytometry data on AML patients and reviewed the manuscript with S.S. G.C. and C.S. provided clinical data from AML patients. P.G.G. and R.J.A. provided reagents for SCENITH and metabolic analysis on LSC subsets. B.D. and A.C.([8]) performed NanoString nCounter analysis. L.C. offered help with the manuscript preparation and M.C. edited the final manuscript version. A.C.([6]) enrolled AML patients, edited the manuscript, and provided financial support.

## Competing interests

The authors declare no competing interests.

## Ethics

The research was approved by the institutional review board of the Area Vasta Emilia Centro (AVEC) Ethical Committee (approval code: 94/2016/O/Tess).

## Additional information

[1]Department of Medical and Surgical Sciences, Institute of Hematology "L. and A. Seràgnoli", University of Bologna, Bologna, Italy. [2]Department of Chemistry, Biology and Biotechnology, Biochemical and Biotechnological Sciences Section, University of Perugia, Perugia, Italy. [3]Laboratory of Hematology-Oncology, European Institute of Oncology IRCCS, Milan, Italy. [4]Onco-Tech Lab, European Institute of Oncology IRCCS and Politecnico di Milano, Milan, Italy. [5]Aix Marseille Univ, CNRS, INSERM, CIML, Centre d'Immunologie de Marseille-Luminy, Marseille, France. [6]IRCCS Azienda Ospedaliero-Universitaria di Bologna, Istituto di Ematologia "Seràgnoli", Bologna, Italy. [7]Biosciences Laboratory, IRCCS Istituto Romagnolo per lo Studio dei Tumori (IRST) "Dino Amadori", Meldola, Italy. [8]Laboratory of Translational Research, Azienda USL-IRCCS di Reggio Emilia, Reggio Emilia, Italy. [9]Department of Experimental Oncology, European Institute of Oncology IRCCS, Milano, Italy. [10]Centro di Eccellenza sui Materiali Innovativi Nanostrutturati (CEMIN), University of Perugia, Perugia, Italy. [11]Present address: IRCCS Azienda Ospedaliero-Universitaria di Bologna, Istituto di Ematologia "Seràgnoli", Bologna, Italy.
✉e-mail: antonio.curti2@unibo.it

