## [Transparent Peer Review file · Nature Communications]

Parallel single-cell metabolic analysis and extracellular vesicle profiling reveal novel vulnerabilities with prognostic significance in acute myeloid leukemia

Corresponding Author: Dr Antonio Curti

Version 0:

Reviewer comments:

Reviewer #1

(Remarks to the Author)

Summary:

In this article, the authors aim to elucidate the metabolome of leukemic stem cell like cells (LSC) and to get a better understanding of cargo and function of the LSC-derived extracellular vesicles (EVs) using a liquid biopsy approach. This topic is of scientific interest since the switch to an accelerated metabolic state is defined as a hallmark of cancer, thus potentially providing approaches to novel and urgently needed treatment options in acute myeloid leukemia. Furthermore, the authors tried to demonstrate that an EV conciliated liquid biopsy might possibly be used as less invasive biomarker to study the AML disease course compared to bone marrow biopsies.

Results:

The authors showed a specific metabolic profile in LSCs comprising of lower levels of reactive oxygen species (ROS), higher mitochondrial capability and increased glutathione levels. Utilizing a SCENITH approach (in brief, cells are treated with inhibitors of specific metabolic pathways (i.e. glycolysis), puromycin uptake is measured by flow cytometry as a surrogate parameter for adenosinetriphosphate (ATP) production, therefore making predictions of the dependency on certain metabolic pathways), the authors could showcase a high reliance of circulating LSCs on glucose and glycolysis at initial AML diagnosis, with increasing mitochondrial dependence within the more immature CD34+/CD38low fraction. NTA revealed a bigger EV size of so called AML EV within the patient's plasma accompanied by higher protein amounts and enrichment in glutathione peroxidase 3 (GPX3) indicating a higher antioxidative capability. Using a multiplex protein analysis the researches could show that the EVs from AML plasma show a distinct and sound expression of surface markers that are associated with stem cell cancers (CD44, CD133) and immune cell modulation (CD4) as well as markers that are related to monocytes, macrophages and mesenchymal stroma cells. In addition, the authors could show an association of certain EV markers with ELN risk categories therefore potentially suggesting a prognostic tool. Further, "AML" EV treatment of CD34+ cells demonstrated a decrease in ROS levels and an increase in mitochondrial capacity. To validate that the observed changes in the redox potential is mediated by the earlier described GPX/GSH system they added an GPX4-inhibitor reverting the changes in redox capability thus demonstrating GPX4 mediated changes. Lastly coculture of CD34+ cells from PB with EV from AML patient plasma showed a shift of metabolism towards a mitochondrial dependency and lower glycolytic capacity, demonstrating a metabolic change (outside of the bone marrow).

What are the noteworthy results?

Distinct metabolic and lipidomic patterns as well as a unique marker signature of EVs derived from AML patient plasma at diagnosis might indicate a potential role of the EVs as a tool for disease monitoring. Correlation of ELN subtypes to specific EV marker expression with suggested prognostic relevance.

Will the work be of significance to the field and related fields?

The work has several limitations (please see below) and a dissection of the identified phenotypical changes including functional analyses are lacking, therefore substantial revisions including experiments would need to be added to truly validate the above-described findings.

How does it compare to the established literature? If the work is not original, please provide relevant references.

Mainly descriptive and correlative analysis of EVs with a small proportion of functional experiments. EVs are known circulating biomarkers in several diseases, including AML.

Does the work support the conclusions and claims, or is additional evidence needed?

Substantial additional evidence is needed to confirm the found phenotype, including linking this specifically and to dissect effects of the isolated extracellular vesicle fraction deriving from multiple cells of origin.

Are there any flaws in the data analysis, interpretation and conclusions? Do these prohibit publication or require revision?

The manuscript mainly focusses on descriptive and correlative analyses of so called "AML EV" which are in fact a mixture of EVs derived from many different cells types within the PB of not further aligned AML PB fractions (i.e. PB blast count unknown). Further, substantial additional regulatory cargo analysis of EVs (f.e. RNA) and functional analyses are missing. Interesting topic but results not suitable for a publication within the journal nature communication.

Major concerns:

1. EV isolation method and proof of a pure, intact vesicle fraction

The authors use the commonly used method size exclusion chromatography (SEC) to isolate EVs from PB of healthy donors compared to AML patients. According to MISEV criteria, which are cited within the paper, 1 transmembrane, 1 cytosolic protein and 1 protein found in most common co-isolated contaminants are mandatory (within 1 western blot) as general EV characterization, and further highly recommended at least one non-EV "negative" protein marker. This important EV characterization step within 1 western blot is missing and not shown within the manuscript and needs to be added (including information about technical and biological replicates). Moreover, a whole cell lysate control and a for example "BSA only" control would also be desirable. Further, all additionally mentioned EV characterization steps of the MISEV guidelines need to be addressed and functional validation of an intact isolated EV fraction such as uptake and tagged EV internalization experiments into target cells need to be shown (or at least "not shown", page 9, paragraph 3).

2. EV fraction isolated from PB is not AML-specific

The authors do not preselect leukemic blasts from PB, which is a major flaw of the study. I would highly recommend to pre-select AML cells to produce AML-specific EVs after which those could be analyzed accurately rather than analyzing a mixture of PB EVs which are randomly produced by all existing EV-producing cells. Further, effects and observed characteristics could be due to effects of leukemia derived EVs on other cell fractions. Further, no standardization/alignment nor PB blast count is mentioned within the whole manuscript.

3. Definition of leukemic stem cells and MACS separation

CD34+CD38-/low cells were propagated as leukemic stem cell like cells, but this is a very broad cell population and AML LSCs can have CD34-CD38-, CD34+CD38-, CD34+CD38+, or CD34-CD38 phenotypes with intraindividual differential expression. Furthermore, magnetic bead isolation of cells could have potentially altered the metabolic status of the cells as well. This needs to be ruled out through comparison to other isolation methods such as FACS cell sorting.

4 Further "AML" EV Cargo and functional Analysis missing

The PB so called "AML", but rather all cell-derived EVs were not further analyzed concerning cargo, especially small RNA and miRNA cargo, which could also lead to the phenotypical changes which have been demonstrated upon EV treatment of AML cell lines. Further content and substantial functional analysis need to be added to support the claims of the manuscript.

5 SCENITH approach

CD3+ Cells as Control, more differentiated cells than CD34/CD38 low LSC. Highly proliferating LSC-Clone are expected to have a very high metabolic activity because of the increased cell proliferation.

Is the methodology sound? Does the work meet the expected standards in your field?

Proper EV characterization und functional validation missing.

Is there enough detail provided in the methods for the work to be reproduced?

Mainly yes, detailed patient information missing (f.e. concerning PB blood counts).

Reviewer #2

(Remarks to the Author)

In their manuscript, Forte et al. present a novel flow cytometry-based redox metabolic approach to investigate the metabolic status of AML cells and their interactions with extracellular vesicles (EVs). The authors successfully correlate these findings with the European LeukemiaNet (ELN) 2022 AML risk classification. They demonstrate that AML cells exhibit high GSH levels and mitochondrial functionality, specifically associating with intermediate- and adverse-risk patients and predicting poor prognosis. The manuscript also includes a comprehensive characterization of the isolated EVs, providing evidence that these EVs significantly alter AML cell metabolic status and are correlated with patient prognosis. The potential use of liquid biopsy to characterize AML blast metabolic status and predict AML prognosis makes this study particularly appealing to the readers of Nature Communications. However, the conclusion lacks critical data to support the claims made in the current version.

Major comments:

1. The authors assert that their results reflect LSC-like (CD34+CD38low/-) cells. However, the use of CD34+ cells with limited data (Figure 2d, Extended data Figure 1a-c and Extended data Figure 2b) using CD34+CD38low/- cells does not convincingly represent LSCs, but rather the bulk AML blasts. Thus, the claim of LSC-like cells throughout the manuscript is inaccurate. Moreover, it is essential to clarify whether all the blasts from the 100 AML samples collected (Supplementary Table 1) are indeed positive for CD34. What are the results of AML with blasts negative for CD34 (e.g., acute

monocytic/monoblastic leukemia, AML with NPM1 mutation)? In case this study is confined to AML blasts expressing CD34, the authors should explicitly specify this limitation.

2. To establish the specificity of their results for LSC-like cells, the authors need to compare LSC-like cells with the rest of the bulk AML cells (CD34+CD38+ or non CD34+CD38low/-) in Figure 1 and 2; also demonstrating the difference, if any, between LSC-like cells and the remaining bulk AML cells after EVs treatment.
3. Providing the functional data demonstrating the significance of separating CD34+ cells based on the expression of ROS/GSH and ROS/MITO in Figure 1, and using SCENITH in Figure 2, would enhance the significance of their method.
4. Given that both the intermediate group and adverse group exhibit greater mitochondrial dependence compared to the favorable group (Figure 2e), it is necessary to explain the lack of difference between the favorable group and adverse group in Figure 1 I and L. If this lack of difference is due to the methods employed, the authors should clarify which approach is more effective.
5. It is interesting that EV markers are differentially expressed in adverse-risk and intermediate-risk AML patients. To further strengthen their findings, the authors should explore whether EVs from these two AML categories have differing effects on modifying AML LSC metabolism. Additionally, if there is a method to eliminate EVs and test the metabolic status and engraftment efficiency of AML cells (e.g., transplantation with or without EVs), it would be crucial in highlighting the role of EVs.

Minor comments:

1. Please provide a representative flow gating strategy for experiments using CD34+CD38low/- cells.
2. On line 123, the authors stated that “no significant differences were found in paired samples between the two sources”. However, statistical data for the PB CD34+ group and BM CD34+ group are still necessary (Figure 1d-f). In addition, please provide statistical data for Figure 2b, 2c, and Extended Data Figure 2b (group Mito dep and Glyco cap).
3. On line 217, “Fig. 2e” should be “Fig. 3e”.
4. Please provide quantitative data for Figure 3e.
5. The figure legend of Figure 6f is missing.
6. The resolution of Figure 5f and Figure 6e is too low to be appreciated.
7. Please discuss why EV-driven metabolic adaptation has no effects on BM CD34+ blasts.
8. Many references (e.g., #8-15, 24, 25) lack complete information, such as page numbers and years.

Reviewer #3

(Remarks to the Author)

Forte et al. analyzed peripheral blood from a cohort of AML patients at diagnosis for risk stratification. Specifically, they assessed markers of metabolic activity of circulating tumor cells and the molecular composition of extracellular vesicles. They applied a battery of assays/technologies to characterize different molecular layers. A key finding were distinct metabolic and redox profiles of intermediate risk patients, which adds to the growing body of work in this field. However, I do have concerns regarding this manuscript that should be addressed prior to publication.

Major points:

- 1) The statistical power of this study appears relatively low. This is, even though the overall study comprises 100 patients as per Suppl. Tab. 1., several analyses were performed with much smaller sub-cohorts with no further explanation. This is a major limitation and should be discussed more transparently.
- 2) In view of the above, some claims appear overstated, in particular the ‘prognostic relevance’ of EV profiles.
- 3) The authors claim that their approach is ‘real-time’. However, there is no mention about the timing of the assay and no data are provided that would support an application in a real-world clinical setting.
- 4) I was unable to review the quality of the lipidomics data as the Supplementary File mentioned in the reporting summary is missing (Suppl. Tab. 3 is insufficient for that purpose). Guidance may be found in PMID: 35934691. I would also strongly encourage the authors to make the MS raw files publicly available.
- 5) The authors state ‘the most highly expressed molecular species were tested as putative biomarkers detected in EVs...’. It is not clear on which cohort this test was performed. If done on the same dataset as the PLS-DA analysis, this is a redundant result and no evidence for a potential biomarker panel; particularly given the pertinent risk of overfitting in PLS-DA.
- 6) Jayavelu et al. recently described a ‘Mito-AML’ subtype (PMID: 35245447). The authors should discuss their work in the context of this study.

Minor points:

- 7) L280: please specify 'network analysis' and associated significance levels. What is the meaning of the numbers in Fig. 5b?
- 8) The resolution in some panels of Figs. 5 and 6 is not sufficient.
- 9) Several data are 'not shown'. Why?
- 10) For better readability, the authors should consider dropping some of the acronyms, e.g. 'LB'.
- 11) Statements such as 'for the first time' are superfluous.

Reviewer #4

(Remarks to the Author)

In this study, Dorian Forte and colleagues sought to simultaneously characterize the energy-related metabolism, redox potential, and mitochondrial function in CD34+/CD38Low cells isolated from the peripheral blood of a cohort of AML patients. Furthermore, in order to provide a comprehensive profile of these patients, a significant part of the study was dedicated to the characterization of the nature, size, and content of extracellular vesicles obtained from the same patients through size exclusion chromatography. This was done in comparison to extracellular vesicles isolated from healthy donors, with the ultimate objective of establishing associations between the observed metabolic disturbances in CD34+/CD38Low AML blasts and molecular cues potentially conveyed by extracellular vesicle cargos.

Through this investigation, they established that: i) the less mature CD34+/CD38Low fraction of AML blasts exhibited a decrease in glycolytic capacity and an increased reliance on mitochondria, which was associated with higher levels of the antioxidant glutathione, ii) extracellular vesicles from AML patients were more abundant than those from healthy donors, iii) these vesicles displayed various protein and lipid markers that could assist in better stratifying AML patients, iv) the extracellular vesicles had the capability to support mitochondrial metabolism in CD34+ cells.

This study is well-conducted overall, technically sound, and relies heavily on the use of primary patient materials. While most of the findings are correlative in nature, the study holds the technical merit in its exploration of the influence of extracellular vesicles on disease onset, a topic that is not trivial to investigate and remains incompletely understood in the current literature. To try to establish a more causal relationship with their observations, the authors also isolated extracellular vesicles from patients for further investigation of their metabolic impact through co-culture with AML cell lines and primary cells.

Additional major points:

1- Although the authors consistently correlate cell phenotypes and extracellular vesicle biomarkers with well-established patient stratification methods like the ELN 2022 categories, there is an overall absence of dedicated analyses intended to link metabolic phenotypes and vesicle size distribution with patients' mutational status. It would be more informative for the reader to understand if specific mutations, combinations of mutations, or classes of mutations are more closely associated with particular metabolic phenotypes or the composition of extracellular vesicles. Could the authors address this critical point using, for instance, more-in-depth associative statistical analyses?

2- Does the CD34+ cell fraction exhibiting low ROS levels and high mitochondrial potential have the capacity to resist apoptosis more effectively? Does this observation correspond to a reduced mitochondrial apoptotic priming, as measured by BH3 profiling?

3- Is the increased extracellular vesicle size distribution observed in AML patients, compared to healthy donors, associated with hyperleukocytosis or high blast counts in patients? In other words, is there a quantitative relationship between leukemia burden and vesicle size?

4- On the one hand, the authors demonstrated that specific extracellular vesicle markers are more prevalent in AML patients and are linked to the metabolic state of CD34+ cells. On the other hand, they showed that when isolated extracellular vesicles were co-cultured with AML CD34+ cells, they influenced their mitochondrial metabolism. However, it would provide valuable insights to determine whether extracellular vesicles with specific surface markers are more likely to induce these metabolic changes in CD34+ AML cell metabolism. For example, based on their vesicle profiling data, could the authors selectively isolate (by immunoprecipitation, for instance) extracellular vesicles that express distinct classes of molecules associated with specific metabolic phenotypes and conduct co-culture-based metabolic experiments with CD34+ cells using these marker-enriched immunoprecipitated vesicles?

5- In line with the previous question, the results section does not clearly indicate the impact of extracellular vesicles isolated from AML patients on cord-blood-derived CD34+ cells. Although the authors reported that they do not influence their redox metabolism, it remains uncertain whether they affect other metabolic characteristics or the growth of healthy cord-blood-derived CD34+ cells. Does the observed metabolic reprogramming induced by extracellular vesicles from AML patients selectively affect cells with a (pre-)malignant phenotype?

6- Throughout the study, the authors frequently asserted that extracellular vesicles from AML patients might play a crucial role as a "mediator of metabolic reprogramming," associated with "a more aggressive and chemoresistant phenotype," without providing substantial evidence to support this claim. In addition to the changes in mitochondrial metabolism outlined by the authors, it would be valuable to see additional functional evidence that these extracellular vesicles convey in vitro a signal of enhanced proliferation and chemoresistance. For instance, are primary AML cells, healthy CD34+ cells, or AML cell lines more proliferative or more resistant to cytarabine and daunorubicin (or any other front-line therapies) when exposed to extracellular vesicles isolated from patients with a poor prognosis?

Version 1:

Reviewer comments:

Reviewer #1

(Remarks to the Author)

In general, the authors have improved the manuscript in terms of EV characterization and clarified their point on using a broader non-AML specific EV population, due to the aim of establishing a new diagnostic/prognostic liquid biopsy tool. Further, the authors have responded to all my major concerns.

1. The WB EV characterization has been improved significantly. WB analyses included for calnexin (cell marker), Apo-A1, and HSA (non-EV marker). I would suggest including the cell lysate and EV samples on the same gel/WB. Also, given the four independent experiments, a densitometric analysis could be shown.

2. The authors added important information about PB and BM blast percentage and correlations of EV-marker expression between both sources which might be used for disease monitoring. No further concerns.

3-5. The authors fully addressed my comments.

Reviewer #2

(Remarks to the Author)

The authors have performed extensive work to address the reviewers' critiques, which is highly appreciated. However, the main concern remains.

The study indicates that EVs isolated from favorable, intermediate, and adverse AMLs induce different and inconsistent metabolic changes. For example, while "all three types of EVs were observed to reduce the frequency of cells with high ROS and low GSH (ROShiGSHlo)", "only adverse EVAML increased the proportion of ROSlo and high mitochondria (MITOhi) ($p < 0.05$), while significantly decreasing the proportion of ROSHi and GSHhi ($p < 0.01$) (Fig. 6a-c)". These inconsistencies raise questions about the significance of these changes. I still believe it is essential to provide functional data demonstrating the importance of separating CD34+ blasts based on the expression of ROS/GSH and ROS/MITO is essential. Some changes seem meaningless without this context. Craig Jordan's group has demonstrated the feasibility of utilizing Redox-stained AML cells for functional studies, and combining this with GSH or MITO should be feasible.

Furthermore, the majority of experiments were performed on CD34+ AML blasts. Referring to these cells as AML stem cells or LSC-like is misleading, as they are merely bulk blasts. Using the term AML cells is more appropriate for the experiments described in the manuscript. In supplemental Figure 5, the authors investigated human cell lines reacting to EV from AMLs, and concluded, "This suggests that EVAML might have a more evident role in specific LSC metabolism" (line 423-424). This conclusion is inaccurate, as the experiments were conducted on bulk cell line cells.

In Figure 2d, the authors observed a significant increase in mitochondrial dependence and a reduction in glycolytic capacity in the more immature CD34+CD38low/- fraction. However, in Supplemental Figure 2e, no significant difference was observed, and the authors claimed, "our analysis revealed that more specific leukemic subsets, such as progenitors, myeloid cells, and primitive LSCs, did not exhibit significant differences in the metabolic parameters evaluated (Supplementary Fig. 2c-f). This observation reinforces the validity of using bulk staining for CD34+ and CD38low/- markers to identify leukemic subsets in the circulation of AML patients for metabolic studies". This discrepancy is confusing and further supports my above point – using the term AML blasts.

Reviewer #3

(Remarks to the Author)

The authors have thoroughly revised their manuscript. In particular, the sample size is now described more transparently and the authors have toned down their claims regarding the prognostic value.

Two concerns remain:

1) The authors should more explicitly describe their acceptance criteria for lipid identifications, as some species with odd-numbered fatty acyl chains might be unlikely in the samples (e.g. one third of the reported DG species), see also <https://doi.org/10.1038/s41580-024-00758-4>

2) I would still appreciate more detail on the network analysis presented in Fig. 5b.

Reviewer #4

(Remarks to the Author)

The authors satisfactorily addressed the points raised by the reviewer and substantially revised their manuscript, resulting in substantial improvements that justify its publication.

Version 2:

Reviewer comments:

Reviewer #1

(Remarks to the Author)

The authors have effectively responded to the reviewer's comments and made significant revisions to their manuscript, leading to notable enhancements that warrant its publication.

Reviewer #2

(Remarks to the Author)

The authors have thoroughly addressed all of my concerns. I have no further questions. I greatly appreciate their efforts, and the manuscript has been significantly enhanced.

Reviewer #3

(Remarks to the Author)

The authors have sufficiently addressed my points.

Dear Reviewers,

Please find below a point-by-point detailed rebuttal to your comments. Please note that we added a new Figure (6) to accommodate additional experimental data. All changes have been highlighted in the manuscript and below here as required. In red is the copy-paste section from the manuscript with the suggested corrections.

REVIEWER COMMENTS

Reviewer #1, expertise in extracellular vesicles and AML (Remarks to the Author):

Summary:

In this article, the authors aim to elucidate the metabolome of leukemic stem cell like cells (LSC) and to get a better understanding of cargo and function of the LSC-derived extracellular vesicles (EVs) using a liquid biopsy approach. This topic is of scientific interest since the switch to an accelerated metabolic state is defined as a hallmark of cancer, thus potentially providing approaches to novel and urgently needed treatment options in acute myeloid leukemia. Furthermore, the authors tried to demonstrate that an EV conciliated liquid biopsy might possibly be used as less invasive biomarker to study the AML disease course compared to bone marrow biopsies.

Results:

The authors showed a specific metabolic profile in LSCs comprising of lower levels of reactive oxygen species (ROS), higher mitochondrial capability and increased glutathione levels. Utilizing a SCENITH approach (in brief, cells are treated with inhibitors of specific metabolic pathways (i.e. glycolysis), puromycin uptake is measured by flow cytometry as a surrogate parameter for adenosinetriphosphate (ATP) production, therefore making predictions of the dependency on certain metabolic pathways), the authors could showcase a high reliance of circulating LSCs on glucose and glycolysis at initial AML diagnosis, with increasing mitochondrial dependence within the more immature CD34+/CD38^{low} fraction. NTA revealed a bigger EV size of so called AML EV within the patient's plasma accompanied by higher protein amounts and enrichment in glutathione peroxidase 3 (GPX3) indicating a higher antioxidative capability. Using a multiplex protein analysis the researches could show that the EVs from AML plasma show a distinct and sound expression of surface markers that are associated with stem cell cancers (CD44, CD133) and immune cell modulation (CD4) as well as markers that are related to monocytes, macrophages and mesenchymal stroma cells. In addition, the authors could show an association of certain EV markers with ELN risk categories therefore potentially suggesting a prognostic tool. Further, "AML" EV treatment of CD34+ cells demonstrated a decrease in ROS levels and an increase in mitochondrial capacity. To validate that the observed changes in the redox potential is mediated by the earlier described GPX/GSH system they added an GPX4-inhibitor reverting the changes in redox capability thus demonstrating GPX4 mediated changes. Lastly coculture of CD34+ cells from PB with EV from AML patient plasma showed a shift of metabolism towards a mitochondrial dependency and lower glycolytic capacity, demonstrating a metabolic change (outside of the bone marrow).

What are the noteworthy results?
Distinct metabolic and lipidomic patterns as well as a unique marker signature of EVs derived from AML patient plasma at diagnosis might indicate a potential role of the EVs as a tool for disease monitoring. Correlation of ELN subtypes to specific EV marker expression with suggested prognostic relevance.

Will the work be of significance to the field and related fields?
The work has several limitations (please see below) and a dissection of the identified phenotypical changes including functional analyses are lacking, therefore substantial revisions including experiments would need to be added to truly validate the above-described findings.

How does it compare to the established literature? If the work is not original, please provide relevant references.
Mainly descriptive and correlative analysis of EVs with a small proportion of functional experiments. EVs are known circulating biomarkers in several diseases, including AML.

Does the work support the conclusions and claims, or is additional evidence needed?
Substantial additional evidence is needed to confirm the found phenotype, including linking this specifically and to dissect effects of the isolated extracellular vesicle fraction deriving from multiple cells of origin.

Are there any flaws in the data analysis, interpretation and conclusions? Do these prohibit publication or require revision?
The manuscript mainly focusses on descriptive and correlative analyses of so called "AML EV" which are in fact a mixture of EVs derived from many different cells types within the PB of not further aligned AML PB fractions (i.e. PB blast count unknown). Further, substantial additional regulatory cargo analysis of EVs (f.e. RNA) and functional analyses are missing. Interesting topic but results not suitable for a publication within the journal nature communication.

Major concerns:

1. EV isolation method and proof of a pure, intact vesicle fraction
The authors use the commonly used method size exclusion chromatography (SEC) to isolate EVs from PB of healthy donors compared to AML patients. According to MISEV criteria, which are cited within the paper, 1 transmembrane, 1 cytosolic protein and 1 protein found in most common co-isolated contaminants are mandatory (within 1 western blot) as general EV characterization, and further highly recommended at least one non-EV "negative" protein marker. This important EV characterization step within 1 western blot is missing and not shown within the manuscript and needs to be added (including information about technical and biological replicates). Moreover, a whole cell lysate control and a for example "BSA only" control would also be desirable. Further, all additionally mentioned EV characterization steps of the MISEV guidelines need to be addressed and functional validation of an intact isolated EV fraction such as uptake and tagged EV internalization experiments into target cells need to be shown (or at least "not shown", page 9, paragraph 3).

We thank Reviewer #1 for bringing up this point.

In response to the reviewer's suggestion, we have added positive controls for the Western blot. We used unprocessed serum from AML patients and compared it to plasma-derived EVs for the detection of albumin and apolipoprotein-A1. Our results show a reduction of both albumin and apolipoprotein-A1 in EVs in comparison to serum. Additionally, we have included a cell-specific marker that is expected to be absent in EVs, calnexin, for EV purity. We have also provided more detailed information about our technical and biological replicates. We updated our reference to the latest MISEV guidelines (2023) (*J Extracell Vesicles*. 2024 Feb;13(2):e12404. doi: 10.1002/jev2.12404).

In summary, according to MISEV 2023, we have identified at least one protein from each of the following categories in the EVs isolated for this study: 1) transmembrane protein (e.g. CD81), 2) cytosolic protein (e.g. flotillin-1), 3) major components of non-EV co-isolated structures (e.g. apolipoprotein and albumin). Additionally, as an optional characterization, we have included endoplasmic reticulum (ER) membrane protein (e.g., calnexin) as a cell-specific 'exclusion' marker to demonstrate EV purity further.

We added the following test in the manuscript, and we updated the **Supplementary Figure 3**

Page 6; line 235:

Western blot analyses revealed a substantial decrease in contaminants such as albumin and apolipoprotein A1 (apo-A1) in our EV fractions in comparison with the serum counterparts. Calnexin was only detected in cell lysates (Supplementary Fig. 3b).

b) Western blot analyses for contaminants. The absence of cell-specific marker calnexin (90kDa) in EV from HD and AML patients in comparison to the positive control with cell lysate. Human serum albumin (HSA; 66 kDa) and apo-A1 (28kDa) reduction in EV^{HD} and EV^{AML} compared to positive control with AML serum (representative data for two subjects for each group from 4 independent experiments).

As suggested by Reviewer #1, we have removed the data not shown (page 6, Lines 235-238) (page 9, line 357) and included the figure depicting the uptake of EVs labeled with the green membrane dye PKH67 in AML CD34⁺ cells within 24 hours.

We have included the following text into the manuscript:

EV isolated from AML patients were stained with the green membrane dye PKH67 and then co-cultured with AML CD34⁺ for 4 to 24 hours at 37°C. The uptake of EV^{AML} was measured by flow cytometry (Supplementary Fig.7a-b). To rule out passive uptake, a negative control was conducted using PKH67 dye without EVs, or by keeping the AML CD34⁺ cells at 4°C to inhibit uptake. The data confirmed the EV uptake within 24 hours as an active process.

Figure legend Supplementary Figure 7

EV^{AML} can be taken up by CD34⁺ cells. Primary human AML CD34⁺ cells were treated with PKH67-labeled EVs or with green dye PKH67 (without EVs) for 4 hours (a) or 24 hours (b) at 4°C or 37°C. Representative flow cytometry histograms showing the PKH67-labeled EV^{AML} uptake by CD34⁺ cells in the FITC channel (green histograms) in comparison to CD34⁺ cells treated with PKH67 without EV^{AML} (blue histograms) or with unstained EV^{AML} (light blue histograms) or control cells (gray histograms). AML cell lines (e.g. MOLM-13) provided similar results (data not shown). Data are presented as the mean values of 4 independent experiments.

2. EV fraction isolated from PB is not AML-specific
 The authors do not preselect leukemic blasts from PB, which is a major flaw of the study. I would highly recommend to pre-select AML cells to produce AML-specific EVs after which those could be analyzed accurately rather than analyzing a mixture of PB EVs which are randomly produced by all existing EV-producing cells. Further, effects and observed characteristics could be due to effects of leukemia derived EVs on other cell fractions. Further, no standardization/alignment nor PB blast count is mentioned within the whole manuscript.

We agree with Reviewer #1 that circulating EVs are not specific for AML. However, the major goal of our study is to exploit a method for identifying disease biomarkers in EV-based liquid biopsy of AML patients at diagnosis. Moreover, towards a translation of EVs into clinical practice, we aimed to determine whether bulk fractions of EVs in circulation could provide predictive signals distinct from those obtained by the cells without relying on BM samples. This approach could offer less invasive and more accessible diagnostic and prognostic tools. However, technical limitations must be addressed, particularly the difficulty in obtaining a clinically relevant number of EVs. We tried to isolate EVs from AML-immortalized cell lines, but our data were inconclusive, and the expression of markers on this type of EVs can not be considered relevant in a clinical setting. We opted not to preselect a fraction of EVs

because it would provide an incomplete picture and be less informative than using PB plasma.

We appreciate Reviewer #1's comment on this section, and in response, we have tried different approaches to address our study's flaws.

We have included a table with specific information on morphology examination (BM and PB blasts) for all the samples used for EV isolations so that we can perform the analysis requested.

We included the following table in the manuscript (**Supplementary Table 2**) for all the AML patients on whom we performed multiplex analysis of isolated EVs:

UPN	Morphology examination (%)		Age (Y)	Sex	diagnosis	ELN_2022	WBC (x 10 ⁹ /L)	EV profiling by MACSplex
	BM blasts	PB blasts						
ID_1	60	20	65	M	novo	adv	16,43	Y
ID_2	24	13	62	M	novo	adv	1,19	Y
ID_3	30	20	79	F	novo	adv	1,63	Y
ID_4	70	88	60	M	novo	adv	18,58	Y
ID_5	N/A	34	75	M	novo	adv	10,45	Y
ID_6	90	90	53	M	novo	adv	25,47	Y
ID_7	80	8,6	69	F	novo	adv	1,08	Y
ID_8	30	7	58	F	sec	adv	19,72	Y
ID_9	50	27	49	M	ter	adv	45,21	Y
ID_10	60	3	66	M	novo	adv	1,67	Y
ID_11	50	30	64	F	novo	adv	1,37	Y
ID_12	80	35	73	F	sec	adv	1,82	Y
ID_13	25	15	63	F	sec	adv	5,60	Y
ID_14	80	51	52	M	novo	int	28,80	Y
ID_15	80	12	79	F	novo	int	10,28	Y
ID_16	70	15	84	M	sec	int	2,07	Y
ID_17	100	66	69	M	novo	int	0,94	Y
ID_18	80	8	48	F	novo	int	2,53	Y
ID_19	80	67	72	M	sec	int	27,48	Y
ID_20	N/A	68	69	M	novo	int	2,90	Y
ID_21	20	2	52	M	novo	int	0,73	Y
ID_22	70	46	68	M	sec	int	20,00	Y
ID_23	100	95	35	M	novo	int	164,10	Y
ID_24	50	0	70	M	novo	int	1,79	Y
ID_25	40	0	38	F	novo	int	1,69	Y
ID_26	80	89	22	M	novo	int	11,40	Y
ID_27	N/A	N/A	54	M	novo	fav	7,20	Y

ID_28	80	20	82	F	novo	fav	5,30	Y
ID_29	100	66	68	F	novo	fav	115,57	Y
ID_30	30	4	62	M	novo	fav	1,00	Y
ID_31	90	87	55	M	novo	int	11,15	Y
ID_32	*4	2	54	F	novo	int	3,59	Y
ID_33	60	2	64	M	novo	int	2,87	Y
ID_34	90	30	70	M	novo	adv	10,09	Y
ID_35	40	10	64	M	novo	int	0,91	Y
ID_36	*15	N/A	77	M	novo	int	2,05	Y
ID_37	90	43	83	F	novo	fav	31,94	Y
ID_38	70	44	83	M	novo	adv	7,68	Y
ID_39	*8	0	46	M	novo	adv	2,04	Y
ID_40	10	1,5	76	M	sec	fav	111,05	Y
ID_41	50	18	18	F	novo	fav	N/A	Y

* hemodiluted bone marrow aspirates or myeloid sarcoma (MS) or with specific AML-defining recurrent genetic abnormality; Abbreviations: N/A not available; sec: secondary AML; ter: Therapy-related AML; novo: de novo AML; Y: yes; F: female or M: male; adv/int/fav: adverse, intermediate and favorable ELN risk.

The characterization of markers expressed on EVs can provide biological information about the cell or tissue of origin and their functional states. We performed a specific analysis comparing markers expressed on EVs using multiplex analysis (MACSplex with 37 markers) and BM blast counts to explore whether EV profiling can reflect the BM cellular composition and associate with blast counts.

We observed that EV^{AML} were highly enriched in the leukemic marker CD44. Importantly, we found that CD44 expression on PB EV^{AML} positively correlated with BM blasts detected by morphology examination (See **Supplementary Table 2** and **Supplementary Fig. 4a**). This finding might suggest the potential of CD44 as a reliable marker for monitoring disease status in AML patients. Moreover, using the expression of specific markers on EVs as a liquid biopsy to surrogate BM composition offers significant advantages in terms of non-invasiveness, accessibility, and potential for regular monitoring.

Supplementary Fig. 4a

a) Spearman correlation between CD44 expression on PB EV expressed as MFI and BM blast percentages detected by morphology examination (n=38, AML patients, R = 0.37, P = 0.02).

Page 8, lines 292-295

Interestingly, we found a positive correlation between the frequency of BM blasts and the expression of CD44 on circulating EV^{AML} (Supplementary Fig. 4) suggesting CD44 as a putative biomarker for assessing BM blast levels in circulation.

To address Reviewer #1's concern regarding PB-derived EVs, we also isolated EV^{AML} from the BM of AML patients. We conducted a paired comparison of EV^{AML} isolated from both PB and BM from the same AML patients (n=10). This comparison allows for a direct evaluation of the differences and similarities between EVs from these two sources, and it helps to validate the use of EV-based biomarkers.

Our results are now included in the manuscript as follows:

Page 7 line 288

To capture a more comprehensive and specific picture for AML, we examined paired EV^{AML} obtained from both PB and BM plasma. Interestingly, we found that nine markers were significantly higher in BM EV^{AML}, including markers related to antigen-presenting molecules (HLA-ABC and HLA-DRDPDQ; P < 0.001), T cells (CD4; P < 0.01), monocytes (CD14/CD40; P < 0.05), platelets (CD41b, CD62; P < 0.05) and mesenchymal stromal cells (CD29, CD105; P < 0.05) (Supplementary Fig. 4g).

Figure legend in Supplementary Fig. 4g

Background-corrected median APC fluorescence intensity for 37 surface markers detected in EV^{AML} from PB (light blue) versus BM plasma (blue). Multiple t-test with Wilcoxon matched-pairs signed rank test (n=10).

3. Definition of leukemic stem cells and MACS separation CD34+CD38-/low cells were propagated as leukemic stem cell like cells, but this is a very broad cell population and AML LSCs can have CD34-CD38-, CD34+CD38-, CD34+CD38+, or CD34-CD38 phenotypes with intraindividual differential expression. Furthermore, magnetic bead isolation of cells could have potentially altered the metabolic status of the cells as well. This needs to be ruled out through comparison to other isolation methods such as FACS cell sorting.

We agree with Reviewer #1 and have considered the potential effects of magnetic bead isolations in our experiments. To ensure a more reliable metabolic characterization of AML cells, we primarily used fresh whole blood for our experiments without isolating or manipulating the cells. This is reported in Figure 1 (by redox metabolic staining) and Figure 2 (by SCENITH approach), respectively. Both methods require fresh whole blood to be processed in less than 1 hour.

Additionally, we set up a preliminary set of experiment to overcome any interference related to the isolation method used and to investigate how magnetic bead isolation could affect the metabolism of CD34+ cells. We used total MNC (from 3 AML patients) after Ficoll density isolation, and we split the samples to test metabolic parameters directly to the MNC after staining with CD34 antibody (defined as 'unprocessed CD34+ cells') and after isolating CD34+ cells using magnetic separation (defined as 'magnetically separated CD34+ cells'). As shown in Figure 1, for Reviewer #1's reference, we observed no significant differences in metabolic parameters detected using SCENITH when comparing the 'unprocessed' MNCs stained with CD34+ and the CD34+ after magnetic separation. This indicates that our isolation method did not likely affect the metabolic status of the cells separated.

We also believe that using another isolation method, such as FACS sorting, might further impact cell metabolism due to the pressure exerted by the device and the time required for sorting.

Metabolic parameters were identified in paired mononuclear cells (MNC) and immunomagnetically isolated CD34 cells from AML patients at diagnosis (n = 3) using the SCENITH approach. The figure provided for the reviewer only indicates no significant differences in mitochondrial dependency, glucose dependence, fatty acid and AA oxidation, and glycolytic capacity between the two conditions.

To date, the definition of LSC is still elusive. Trying to define LSC by its surface phenotype is challenging, considering that a plethora of additional surface candidates have been studied. However, it is well known that the majority of LSC are enriched within the CD34⁺ compartment (Arnone M et al. *Cancers (Basel)*. 2020;12(12):3742/ Shuchi Agrawal-Singh, et al. *Mol Oncol*. 2023 Dec; 17(12): 2493–2506). By focusing on CD34 expression, we ensured a consistent and standardized approach to evaluate disease metabolic status, minimizing the impact of individual variations in different marker expressions.

However, according to the Reviewer's suggestion, we removed the 'LSC-like' definition from the text, where we used only CD34 and CD38 markers, and substituted it with AML CD34⁺(/CD38^{low/-}) cells and we explained our choice in the manuscript (Results section) as reported below.

Page 4, lines 103-107

To minimize the impact of intratumoral heterogeneity, our study focused on the CD34⁺(CD38^{low/-}) fractions, representing the most common immunophenotype in AML, for a more consistent and standardized approach. These fractions are known to exhibit chemotherapy resistance and possess the highest leukemogenic ability¹, making them crucial targets for understanding disease progression and developing effective treatments.

Moreover, we investigated the metabolic parameters of the other leukemic fractions identified by combining CD45 expression with primary markers (CD34, CD117, CD38) and myeloid markers (CD33, HLA-DR, CD123, gating strategy 2), as reported in the manuscript (Method and Result sections and **Supplementary Fig. 1a and e**).

Accordingly, we have added our gating strategy in the **Supplementary Fig. 1a and e**

Gating strategy 1

a) Gating strategy 1. Blast populations were identified by CD45^{low/-}/SSC gating strategy to define and analyze mainly CD34⁺ stem cells, immature CD34⁺ CD38^{low/-} stem cells and CD34⁺ CD38⁺ progenitor cells. For redox metabolic analysis, the CD34⁺ leukemic subsets were analyzed in combination with CellROX (ROS), MitoTracker CMXRos (MITO), and Thiol Tracker (GSH) using two-by-two gating on CD3⁺ cells as a reference marker (lympho CD3⁺ gate).

e) Gating strategy 2. Blast populations were identified by CD45^{low/-}/SSC gating strategy to define and analyze immature and progenitor leukemic cells (CD34⁺ and/or, CD117⁺), myeloid cells (CD33⁺, HLA-DR⁺), and primitive LSC (CD123⁺). For redox metabolic analysis, the leukemic subsets were analyzed in combination with CellROX (ROS), MitoTracker CMXRos (MITO), and Thiol Tracker (GSH) using two-by-two gating based on CD3⁺ cells as reference marker (lympho CD3⁺ gate).

When we explored the redox metabolic parameters (ROS/GSH/MITO) in different leukemic fractions, we observed that CD34⁺CD38⁻ AML blasts showed less GSH content, mitochondrial potential and low ROS levels compared to other fractions. We have now included this data and the information in our manuscript.

We added the following text in the manuscript (Result section) and the data (**Supplementary Fig. 1f-h**).

Page 4, lines 127

Then, we further investigated more specific leukemic subpopulations capturing progenitors (CD34⁺CD38⁺, CD34⁺ and/or CD117⁺), myeloid cells (CD33⁺, HLA-DR⁺), and primitive LSC (CD123⁺) (Supplementary Fig. 1e-h).

Within the leukemic blasts, we observed a higher frequency of CD34⁺CD38⁻ with low GSH content and low mitochondrial potential (GSH^{lo} MITO^o) compared to CD34⁺ ± CD38^{low/-} ± CD117⁺ subsets ($P < 0.05$). Also, the percentage of CD34⁺CD38⁻ cells with low ROS levels and low mitochondrial potential were increased compared to CD34⁺CD117⁺ blasts ($P < 0.05$). No other differences were reported in the redox metabolic fraction for the leukemic subsets explored (Supplementary Fig. 1f-h).

For SCENITH analysis, we found that the selected subsets do not show significant differences in the metabolic parameters tested, supporting our bulk staining for CD34⁺ and CD38^{low/-} to identify leukemic fractions in the circulation of AML patients.

We added the following text in the manuscript (Result section, page 6, lines 195-198) and the Figure in **Supplementary Fig. 2c**:

Of note, our analysis revealed that more specific leukemic subsets such as progenitors, myeloid cells, and primitive LSCs, did not exhibit significant differences in the metabolic parameters evaluated. This observation reinforces the validity of using bulk staining for CD34⁺ and CD38^{low/-} markers to identify leukemic subsets in the circulation of AML patients for metabolic studies.

c-f) Comparison between leukemic cell subsets derived from PB for glucose dependence (Gluco dep; c), fatty acid and aminoacid oxidation (FAAO cap; d), mitochondria dependence (Mito dep; e) and glycolytic capacity (Glyco cap; f) expressed in percentage for CD34⁺CD38^{low/-}, CD34⁺CD38⁺, CD34⁻CD38⁺, CD34⁻CD38⁻, CD34⁻CD117⁺, CD34⁺CD117⁺ and CD34⁺CD117⁺HLA-DR⁺CD33⁺CD123⁺ subsets in fresh PB from 13 AML patients at the diagnosis. One-way ANOVA followed by Šidák's multiple comparisons test.

4 Further “AML” EV Cargo and functional Analysis missing The PB so called “AML”, but rather all cell-derived EVs were not further analyzed concerning cargo, especially small RNA and miRNA cargo, which could also lead to the phenotypical changes which have been demonstrated upon EV treatment of AML cell lines. Further content and substantial functional analysis need to be added to support the claims of the manuscript.

We thank Reviewer #1 for bringing this point to our attention.

We have partially addressed this point in the response to point 2 above.

In response to the Reviewer's request on EV cargo, we have decided to conduct a metabolomic analysis on EVs isolated from AML patients (n=27) and HD donors (n=12), aligning with our focus on metabolism. Considering the risk assessment, we have conducted targeted metabolomic studies on EVs from AML patients, and we focused on TCA cycle intermediates that we previously found involved in AML (*Forte et al. Cell metabolism 2020*).

We decided to add these data as follows in the results sections:

Page 9, lines 361-373

*To further investigate the cargo of our EV^{AML} further, we conducted a targeted metabolomic analysis focusing on the content of mitochondrial Tricarboxylic acid (TCA) cycle intermediates in our EVs. This analysis was based on our previous data where TCA intermediates were found to be increased in AML². Although we did not observe significant differences between EV^{AML} and EV^{HD}, we noted that several metabolites (namely fumarate, aconitic acid, shikimic acid, citrate, isocitrate, and glucose) showed a two-fold increase in adverse or intermediate EV compared to favorable EVs. Sucrose was detected in only three patients and was not present in EVs from HD (**Supplementary Fig. 4h**). Notably, alpha-ketoglutarate was significantly elevated in adverse EV^{AML} compared to favorable EVs ($P < 0.05$) (Fig. 5g).*

Taken together, lipidomic analysis of circulating EVs identifies a lipid-based abnormal signature of EVs from AML patients and suggests the alteration of the EV lipidome in relation to risk stratification at diagnosis. Furthermore, the targeted metabolomic studies demonstrate an elevated level of α -ketoglutarate in adverse EV^{AML}.

h) Targeted metabolomic data on EV^{AML} (n=27) versus EV^{HD} (n=12). TCA cycle intermediates including urea, succinate, glucose, sucrose, lactic acid, fumarate, malate, α -keto-glutarate, aconitic

acid, shikimic acid, citrate, isocitrate, glucose-6-phosphate expressed as pg/ μ l. No significant differences were reported between groups by two-way ANOVA.

We have included a panel g in **Figure 5** (added to lipidomic data on EVs) to show the difference between favourable and adverse-risk AML patients in alpha-keto-glutarate, and we discussed this data in the Discussion.

Figure 5g

Targeted metabolomic data on EV^{AML} (n=27) versus EV^{HD} (n=12). TCA cycle intermediate α -ketoglutarate expressed as pg/ μ l between favorable (n=6), intermediate (n=10), and adverse (n=11) EV^{AML}. Significant difference was reported by Kruskal-Wallis test with Dunn's post-test.

In the discussion section, page 14, lines 631-636

Our initial metabolomic study on EV^{AML} showed higher concentrations of the metabolite alpha-ketoglutarate in EV^{AML} patients from adverse-risk AML patients. Alpha-ketoglutarate is a crucial intermediate of the TCA cycle and a precursor of glutamine, serving as an antioxidant in various cellular processes and crucial for mitochondrial metabolism^{2, 3}. Further exploration of the metabolomic cargo of EV^{AML}, coupled with the identification of lipid species, holds the potential to elucidate the metabolic impacts of EV^{AML} on AML cells.

We have included the relevant information in the Supplementary Method section (page 9, lines 881-885)

Measurement of TCA cycle intermediates by Gas chromatography (GC)-triple quadrupole Mass Spectrometry GC-QqQ-MS in EVs

Analysis of the TCA cycle intermediates was performed by GC-QqQ-MS at Swedish Metabolomics Centre, for detailed method description see Supplementary information.

Information reported in the **Supplementary amethod and Supplementary Table 5:**

Measurement of TCA cycle intermediates by Gas chromatography (GC)-triple quadrupole Mass Spectrometry GC-QqQ-MS in EVs

For Gas Chromatography–Mass Spectrometry (GC-MS) analysis an 11-point calibration curve (cis-aconitic acid, a-keto-glutaric acid, citric acid, fumaric acid, glucose, glucose 6-phosphate, isocitric acid, lactic acid, malic acid, shikimic acid, succinic acid, sucrose, urea) spanning from 25-2500 pg/μL was prepared by serial dilutions and spiked with internal standards (Fumaric acid (13C4), L-Malic acid (13C4), D-Glucose (13C6), α-ketoglutaric acid (13C4), Succinic acid (D4), citric acid (D4), Sucrose (13C12) at a final concentration of 350 pg/μL. EV isolated following our protocols were diluted in 80% methanol (1:10), vortex, and then centrifuged for 10 minutes at 4 °C at 14000 RPM. Supernatants were frozen until use. Up to 200 μl of the extracted sample were transferred to GC vials, and the above-mentioned isotopically labeled internal standards were added. The sample was evaporated to dryness under a stream of nitrogen. Derivatization was performed according to Gullberg et al ⁴. In detail, 10μL of methoxyamine (15 μg/μL in pyridine) was added to the dry sample that was shaken vigorously for 10 minutes before left to react in room temperature. After 16 hours and 20 μl of a mixture of MSTFA and methyl stearate (1050 pg/μL in heptane) (1:1) was added and the sample was vortexed. One μL of the derivatized sample was injected by an Agilent 7693 autosampler, in splitless mode into an Agilent 7890A gas chromatograph equipped with a multimode inlet (MMI) and 10 m x 0.18 mm fused silica capillary column with a chemically bonded 0.18 μm DB 5-MS UI stationary phase (J&W Scientific). The injector temperature was 260 °C. The carrier gas flow rate through the column was 1 ml min⁻¹, the column temperature was held at 70 °C for 2 minutes, then increased by 40 °C min⁻¹ to 320 °C and held there for 2 min. The column effluent is introduced into the electron impact (EI) ion source of an Agilent 7000C QQQ mass spectrometer. The thermal AUX 2 (transfer line) and the ion source temperatures were 250 °C and 230 °C, respectively. Ions were generated by a 70 eV electron beam at an emission current of 35 μA and analyzed in dMRM-mode. The solvent delay was set to 2 minutes. For a list of MRM transitions see Suppl. Table 5 below.

Data were processed using MassHunter Qualitative Analysis and Quantitative Analysis (QqQ; Agilent Technologies, Atlanta, GA, USA) and Excel (Microsoft, Redmond, Washington, USA) software. Methyl stearate was used as internal standard for compounds lacking a stable isotope internal standard.

Reference for supplementary method

Gullberg J, Jonsson P, Nordström A, Sjöström M & Moritz T. Design of experiments: an efficient strategy to identify factors influencing extraction and derivatization of Arabidopsis thaliana samples in metabolomic studies with gas chromatography/mass spectrometry. *Anal Biochem* 2004 331 283-295.

Compound	RT (min)	Quantifier	CE	Qualifier	CE
aconitic acid	5.24	375→147	10	375→211	10
alpha-keto-glutaric acid	4.75	288→73	20	288→198	10
alpha-ketoglutaric acid (IS)	4.75	308→147	10	-	-
citric acid	5.42	183→73	10	183→183	10
citric acid (IS)	5.42	276→185	10	-	-
fumaric acid	4.00	245→73	20	245→245	10
fumaric acid (IS)	4.00	249→147	10	-	-
glucose	5.7	319→129	10	319→157	10
glucose (IS)	5.7	323→132	10	-	-
glucose-6-phosphate	6.7	387→387	10	387→73	10
isocitric acid	5.47	245→73	20	245→83	20
lactic acid	2.72	219→147	10	219→191	10
malic acid	4.5	335→147	10	335→73	10
malic acid (IS)	4.5	339→147	20	-	-
methyl stearate (IS)	6.2	298→101	20	-	-
shikimic acid	5.4	462→204	10	462→254	20
succinic acid	3.9	262→73	10	262→113	10
succinic acid (IS)	3.9	251→147	20	-	-
sucrose	7.31	437→257	20	437→303	10
sucrose (IS)	7.31	442→262	10	-	-
urea	3.35	261→147	20	261→245	10

Supplementary Table 5

MRM transitions for GC-QQQ-MS analysis. RT=retention time; CE = Collision Energy.

CD3+ Cells as Control, more differentiated cells than CD34/CD38 low LSC. Highly proliferating LSC-Clone are expected to have a very high metabolic activity because of the increased cell proliferation.

Our data indicate that CD3 cells have a lower metabolic rate compared to LSC-like subsets (see figure 2a). We agree with the Reviewer's comment and acknowledge that our statement on CD3+ as controls is misleading.

According to Reviewer's suggestion, we removed the sentence regarding CD3+ as a reference population in SCENITH.

Page 5, lines 180-181

Interestingly, we detected high puromycin levels, displayed as median fluorescence intensity (MFI), in CD34+ cells in comparison to paired CD3+ cells ~~used as a reference population~~ after vehicle (Control, Co) or oligomycin treatment (O) ($P < 0.0001$, respectively; Fig. 2a), indicating higher protein synthesis and metabolism in the CD34+ AMLAML CD34+ stem LSC-like cell fraction.

Is the methodology sound? Does the work meet the expected standards in your field?
Proper EV characterization und functional validation missing.

Is there enough detail provided in the methods for the work to be reproduced?
Mainly yes, detailed patient information missing (f.e. concerning PB blood counts).

We thank the Reviewer for highlighting these missing points. Accordingly, we have added a more detailed description of AML patients' characteristics, and expanded the number of evaluated AML samples. These allowed us to better characterize leukemic cell subsets, and corroborated the metabolic profiling of other leukemic subsets. We also included the profile of BM-derived EVs and the metabolomic data on isolated EVs.

Reviewer #2, expertise in leukemia stem cells (Remarks to the Author):

In their manuscript, Forte et al. present a novel flow cytometry-based redox metabolic approach to investigate the metabolic status of AML cells and their interactions with extracellular vesicles (EVs). The authors successfully correlate these findings with the European LeukemiaNet (ELN) 2022 AML risk classification. They demonstrate that AML cells exhibit high GSH levels and mitochondrial functionality, specifically associating with intermediate- and adverse-risk patients and predicting poor prognosis. The manuscript also includes a comprehensive characterization of the isolated EVs, providing evidence that these EVs significantly alter AML cell metabolic status and are correlated with patient prognosis. The potential use of liquid biopsy to characterize AML blast metabolic status and predict AML prognosis makes this study particularly appealing to the readers of Nature Communications. However, the conclusion lacks critical data to support the claims made in the current version.

Major comments:

1. The authors assert that their results reflect LSC-like (CD34+CD38low/-) cells. However, the use of CD34+ cells with limited data (Figure 2d, Supplementary Figure 1a-c and Supplementary Figure 2b) using CD34+CD38low/- cells does not convincingly represent LSCs, but rather the bulk AML blasts. Thus, the claim of LSC-like cells throughout the manuscript is inaccurate. Moreover, it is essential to clarify whether all the blasts from the 100 AML samples collected (Supplementary Table 1) are indeed positive for CD34. What are the results of AML with blasts negative for CD34 (e.g., acute monocytic/monoblastic leukemia, AML with NPM1 mutation)? In case this study is confined to AML blasts expressing CD34, the authors should explicitly specify this limitation.

We thank the Reviewer #2 for highlighting this missing point. We have updated the table reporting AML patient characteristics by adding the missing information in the Supplementary Table 1 (overall AML patients) and 2 (specifically for EV studies).

Supplementary Tables 1

Characteristics	N (%), median [range]
n	114
Age	68.5 [18-85]
Gender (%)	
F	40 (35%)
M	74 (65%)
Diagnosis (%)	
de novo	81 (71%)
secondary	31 (27%)
therapy-related	2 (2%)
WBC (x 10⁹/L)	3.64 [0.73-340]

Characteristics		N (%), median [range]
Platelet count (x 10 ⁹ /L)		46 [8-296]
Hemoglobin level (g/dl)		9 [4.7-14.6]
ELN 2022 risk group		
at diagnosis (%)	favorable	18 (16%)
	intermediate	44 (38%)
	adverse	52 (46%)

Supplementary Table 2

Information on morphology examination (BM and PB blast counts) for all AML samples used for EV isolations and EV profiling

UPN	Morphology examination (%)		Age (Y)	Sex	diagnosis	ELN_2022	WBC (x 10 ⁹ /L)	EV profiling by MACSplex
	BM blasts	PB blasts						
ID_1	60	20	65	M	novo	adv	16,43	Y
ID_2	24	13	62	M	novo	adv	1,19	Y
ID_3	30	20	79	F	novo	adv	1,63	Y
ID_4	70	88	60	M	novo	adv	18,58	Y
ID_5	N/A	34	75	M	novo	adv	10,45	Y
ID_6	90	90	53	M	novo	adv	25,47	Y
ID_7	80	8,6	69	F	novo	adv	1,08	Y
ID_8	30	7	58	F	sec	adv	19,72	Y
ID_9	50	27	49	M	ter	adv	45,21	Y
ID_10	60	3	66	M	novo	adv	1,67	Y
ID_11	50	30	64	F	novo	adv	1,37	Y
ID_12	80	35	73	F	sec	adv	1,82	Y
ID_13	25	15	63	F	sec	adv	5,60	Y
ID_14	80	51	52	M	novo	int	28,80	Y
ID_15	80	12	79	F	novo	int	10,28	Y
ID_16	70	15	84	M	sec	int	2,07	Y
ID_17	100	66	69	M	novo	int	0,94	Y

ID_18	80	8	48	F	novo	int	2,53	Y
ID_19	80	67	72	M	sec	int	27,48	Y
ID_20	N/A	68	69	M	novo	int	2,90	Y
ID_21	20	2	52	M	novo	int	0,73	Y
ID_22	70	46	68	M	sec	int	20,00	Y
ID_23	100	95	35	M	novo	int	164,10	Y
ID_24	50	0	70	M	novo	int	1,79	Y
ID_25	40	0	38	F	novo	int	1,69	Y
ID_26	80	89	22	M	novo	int	11,40	Y
ID_27	N/A	N/A	54	M	novo	fav	7,20	Y
ID_28	80	20	82	F	novo	fav	5,30	Y
ID_29	100	66	68	F	novo	fav	115,57	Y
ID_30	30	4	62	M	novo	fav	1,00	Y
ID_31	90	87	55	M	novo	int	11,15	Y
ID_32	*4	2	54	F	novo	int	3,59	Y
ID_33	60	2	64	M	novo	int	2,87	Y
ID_34	90	30	70	M	novo	adv	10,09	Y
ID_35	40	10	64	M	novo	int	0,91	Y
ID_36	*15	N/A	77	M	novo	int	2,05	Y
ID_37	90	43	83	F	novo	fav	31,94	Y
ID_38	70	44	83	M	novo	adv	7,68	Y
ID_39	*8	0	46	M	novo	adv	2,04	Y
ID_40	10	1,5	76	M	sec	fav	111,05	Y
ID_41	50	18	18	F	novo	fav	N/A	Y

* hemodiluted bone marrow aspirates or myeloid sarcoma (MS) or with specific AML-defining recurrent genetic abnormality; Abbreviations: N/A not available; sec: secondary AML; ter: Therapy-related AML; novo: de novo AML; Y: yes; F: female or M: male; adv/int/fav: adverse, intermediate and favorable ELN risk.

We also included a table on **the Source Data File** reporting the samples used for the experiments, including the samples with CD34⁺ cells.

UPN	Age (Y)	Sex	ELN_2022	WBC (x 10 ⁹ /L)	EV profiling by MACSplex	EV isolation for in vitro studies	AML with CD34 ⁺ used for staining on whole blood or separated for in vitro studies	Metabolic profile (SCENITH) on whole blood (CD34 ⁺ cells)	Metabolic profile (REDOX profiling) on whole blood (CD34 ⁺ cells)	Metabolomic and Lipidomic on EVs
ID_1	65	M	adv	16,43	Y	Y	Y	Y	Y	Y
ID_2	62	M	adv	1,19	Y	Y	Y	Y	Y	Y
ID_3	79	F	adv	1,63	Y	Y	Y	Y	Y	

ID_4	60	M	adv	18,58	Y	Y	Y	Y	Y	
ID_5	75	M	adv	10,45	Y	Y	Y	Y	Y	Y
ID_6	53	M	adv	25,47	Y	Y	Y	Y	Y	Y
ID_7	69	F	adv	1,08	Y	Y	Y	Y	Y	Y
ID_8	58	F	adv	19,72	Y	Y	Y	Y	Y	
ID_9	49	M	adv	45,21	Y	Y	Y	Y	Y	Y
ID_10	66	M	adv	1,67	Y	Y	Y	Y	Y	
ID_11	64	F	adv	1,37	Y	Y	Y	Y	Y	
ID_12	73	F	adv	1,82	Y	Y	Y		Y	
ID_13	63	F	adv	5,60	Y	Y				Y
ID_14	52	M	int	28,80	Y	Y	Y	Y	Y	
ID_15	79	F	int	10,28	Y	Y	Y	Y	Y	Y
ID_16	84	M	int	2,07	Y	Y	Y	Y	Y	Y
ID_17	69	M	int	0,94	Y	Y	Y	Y	Y	Y
ID_18	48	F	int	2,53	Y	Y	Y	Y	Y	Y
ID_19	72	M	int	27,48	Y	Y	Y	Y	Y	Y
ID_20	69	M	int	2,90	Y	Y	Y	Y	Y	
ID_21	52	M	int	0,73	Y	Y	Y		Y	Y
ID_22	68	M	int	20,00	Y	Y	Y		Y	Y
ID_23	35	M	int	164,10	Y	Y				Y
ID_24	70	M	int	1,79	Y	Y	Y	Y		Y
ID_25	38	F	int	1,69	Y	Y	Y	Y		
ID_26	22	M	int	11,40	Y	Y				Y
ID_27	54	M	low	7,20	Y	Y	Y	Y	Y	Y
ID_28	82	F	low	5,30	Y	Y	Y	Y	Y	Y
ID_29	68	F	low	115,57	Y	Y	Y		Y	Y
ID_30	62	M	low	1,00	Y	Y				Y
ID_31	55	M	int	11,15	Y	Y	Y		Y	
ID_32	54	F	int	3,59	Y	Y	Y		Y	
ID_33	64	M	int	2,87	Y	Y	Y	Y	Y	
ID_34	70	M	adv	10,09	Y	Y				
ID_35	64	M	int	0,91	Y	Y	Y	Y		
ID_36	77	M	int	2,05	Y	Y	Y	Y		
ID_37	83	F	low	31,94	Y	Y				
ID_38	83	M	adv	7,68	Y	Y				
ID_39	46	M	adv	2,04	Y	Y	Y			
ID_40	76	M	low	111,05	Y	Y	Y			
ID_41	18	F	low	N/A	Y	Y				Y
ID_42	71	F	adv	26,57			Y		Y	
ID_43	72	M	adv	2,45			Y		Y	
ID_44	71	M	adv	2,28					Y	
ID_45	77	F	adv	3,22			Y		Y	

ID_46	72	M	adv	9,78			Y		Y	
ID_47	60	M	adv	5,89			Y		Y	
ID_48	62	F	adv	1,07			Y		Y	
ID_49	50	M	adv	1,70			Y		Y	
ID_50	75	M	adv	1,87			Y	Y	Y	
ID_51	55	F	adv	2,38			Y		Y	Y
ID_52	66	F	int	9,01			Y		Y	
ID_53	72	F	int	0,96			Y		Y	
ID_54	78	M	int	19,34			Y		Y	
ID_55	67	F	int	1,88			Y		Y	
ID_56	58	F	int	3,18			Y		Y	
ID_57	71	M	int	1,06			Y		Y	
ID_58	75	M	int	1,26			Y		Y	
ID_59	64	F	int	6,61			Y	Y	Y	
ID_60	75	M	int	0,98			Y	Y	Y	
ID_61	47	F	int	25,07			Y		Y	
ID_62	49	F	int	1,76			Y		Y	
ID_63	71	M	int	0,78			Y	Y	Y	
ID_64	67	F	int	0,97			Y		Y	
ID_65	44	M	int	2,23			Y		Y	
ID_66	37	M	low	22,46			Y		Y	
ID_67	65	M	adv	2,77		Y				
ID_68	72	M	int	3,64		Y				Y
ID_69	67	F	adv	22,18		Y				
ID_70	79	F	adv	2,68		Y				
ID_71	75	F	low	4,95		Y				Y
ID_72	73	F	low	145,30		Y				Y
ID_73	54	M	int	68,20		Y				Y
ID_74	55	M	low	2,00		Y				Y
ID_75	27	M	adv	235,00		Y				Y
ID_76	52	F	low	3,60		Y				Y
ID_77	51	M	adv	88,63		Y				Y
ID_78	84	M	int	204,43		Y				Y
ID_79	50	M	adv	12,30		Y				Y
ID_80	63	M	int	178,00		Y				Y
ID_81	30	F	low	262,49		Y				Y
ID_82	58	M	adv	340,00		Y				
ID_83	69	F	adv	37,04			Y		Y	
ID_84	85	F	adv	7,63			Y		Y	
ID_85	74	M	adv	38,00			Y		Y	
ID_86	71	M	adv	1,25			Y		Y	

ID_87	55	M	adv	1,82			Y		Y	
ID_88	71	M	adv	2,66			Y		Y	
ID_89	71	M	int	3,64			Y		Y	
ID_90	74	M	int	280,00			Y			
ID_91	62	M	int	3,50			Y		Y	
ID_92	58	M	adv	1,14			Y		Y	
ID_93	61	M	adv	20,51						
ID_94	77	M	int	9,12			Y		Y	
ID_95	73	M	adv	2,90						
ID_96	85	F	low	0,96			Y		Y	
ID_97	46	M	adv	2,90			Y	Y		
ID_98	79	M	low	1,90			Y	Y		
ID_99	84	F	low	31,94			Y	Y		
ID_100	83	M	adv	19,29			Y	Y		
ID_101	35	F	adv	2,21			Y	Y		
ID_102	71	F	adv	3,42			Y	Y		
ID_103	74	M	adv	3,74			Y	Y		
ID_104	70	M	adv	10,00			Y	Y		
ID_105	75	M	adv	26,85			Y	Y		
ID_106	74	F	adv	324,78			Y	Y		
ID_107	74	M	adv	1,58			Y	Y		
ID_108	76	M	adv	35,03			Y	Y		
ID_109	61	M	low	1,27			Y	Y		
ID_110	81	M	low	53,15					Y	
ID_111	77	F	int	1,23			Y			
ID_112	73	M	int	67,68			Y			
ID_113	68	M	int	2,61			Y			
ID_114	76	M	int	21,02			Y			

According to Reviewer #2 suggestion, we investigated the metabolic parameters in different subsets of leukemic blasts to better characterize LSC-like subsets.

To date, the definition of LSC is still elusive. Trying to define LSC by its surface phenotype is challenging, considering that a plethora of additional surface candidates have been studied. However, it is well known that the majority of LSC are enriched within the CD34⁺ compartment (*Arnone M et al. Cancers (Basel). 2020;12(12):3742/ Shuchi Agrawal-Singh, et al. Mol Oncol. 2023 Dec; 17(12): 2493–2506*). By focusing on CD34 expression,

we ensured a consistent and standardized approach to evaluating disease metabolic status, minimizing the impact of individual variations in different marker expressions.

However, according to the Reviewer's suggestion, we removed the 'LSC-like' definition from the text, where we used only CD34 and CD38 markers, and substituted it with AML CD34⁺(/CD38^{low/-}) cells and we explained our choice in the manuscript (Results section) as reported below.

Page 4, lines 103-107

To minimize the impact of intratumoral heterogeneity, our study focused on the CD34⁺(CD38^{low/-}) fractions, representing the most common immunophenotype in AML, for a more consistent and standardized approach. These fractions are known to exhibit chemotherapy resistance and possess the highest leukemogenic ability¹, making them crucial targets for understanding disease progression and developing effective treatments.

Moreover, we investigated the metabolic parameters on the other leukemic fractions identified by the combination of CD45 expression with primary markers (CD34, CD117, CD38) and myeloid (CD33, HLA-DR) and primitive LSC (CD123) markers (as gating strategy 2) as reported in the manuscript (Method and Result sections and **Supplementary 1a and e**).

Gating strategy 1

a) Gating strategy 1. Blast populations were identified by $CD45^{low/-}/SSC$ gating strategy to define and analyze mainly $CD34^+$ stem cells, immature $CD34^+ CD38^{low/-}$ stem cells and $CD34^+ CD38^+$ progenitor cells. For redox metabolic analysis, the $CD34^+$ leukemic subsets were analyzed in combination with CellROX (ROS), MitoTracker CMXRos (MITO), and Thiol Tracker (GSH) using two-by-two gating on $CD3^+$ cells as a reference marker (lympho $CD3^+$ gate).

e) Gating strategy 2. Blast populations were identified by $CD45^{low/-}/SSC$ gating strategy to define and analyze immature and progenitor leukemic cells ($CD34^+$ and/or, $CD117^+$), myeloid cells ($CD33^+$, $HLA-DR^+$), and primitive LSC ($CD123^+$). For redox metabolic analysis, the leukemic subsets were analyzed in combination with CellROX (ROS), MitoTracker CMXRos (MITO), and Thiol Tracker (GSH) using two-by-two gating based on $CD3^+$ cells as reference marker (lympho $CD3^+$ gate).

When we explored the redox metabolic parameters (ROS/GSH/MITO) in different leukemic fractions, we observed that $CD34^+CD38^-$ AML blasts showed less GSH content, mitochondrial potential and low ROS levels compared to other fractions. We have now included this data and related information in our manuscript.

We added the following text in the manuscript (Result section) and the data (**Supplementary Fig.1e-h**).

Page 4, lines 127

Then, we further investigated more specific leukemic subpopulations capturing progenitors ($CD34^+CD38^+$, $CD34^+$ and/or $CD117^+$), myeloid cells ($CD33^+$, $HLA-DR^+$), and primitive LSC ($CD123^+$) (Supplementary Fig. 1e-h).

Within the leukemic blasts, we observed a higher frequency of $CD34^+CD38^-$ with low GSH content and low mitochondrial potential (GSH^{lo} $MITO^{lo}$) compared to $CD34^+ \pm CD38^{low/-} \pm CD117^+$ subsets ($P < 0.05$). Also, the percentage of $CD34^+CD38^-$ cells with low ROS levels and low mitochondrial potential were increased compared to $CD34^+CD117^+$ blasts ($P < 0.05$). No other differences were reported in the redox metabolic fraction for the leukemic subsets explored (Supplementary Fig. 1f-h).

Legend to Supplementary Fig. 1f-h

Column graphs for profiling the following leukemic subsets: CD34⁺CD38^{low/-}, CD34⁺CD38⁺, CD34⁻CD38⁺, CD34⁻CD38⁻, CD34⁻CD117⁺, CD34⁺CD117⁺, CD34⁺CD117⁺HLA-DR⁺CD33⁺CD123⁺ according to ROS/MITO (f), ROS/GSH (g), GSH/MITO (h) (n = 10). Two-way ANOVA was performed to detect significant differences between subsets.

For SCENITH analysis, we found that the selected subsets do not show significant differences in the metabolic parameters tested, supporting our bulk staining for CD34⁺ and CD38^{low/-} to identify leukemic fractions in the circulation of AML patients.

We added the following text in the manuscript (Result section, page 6, lines 195-198) and the Figure in Supplementary **Supplementary 2c**:

Of note, our analysis revealed that more specific leukemic subsets such as progenitors, myeloid cells, and primitive LSCs, did not exhibit significant differences in the metabolic parameters evaluated. This observation reinforces the validity of using bulk staining for CD34⁺ and CD38^{low/-} markers to identify leukemic subsets in the circulation of AML patients for metabolic studies.

c-f) Comparison between leukemic cell subsets derived from PB for glucose dependence (Gluko dep; c), fatty acid and aminoacid oxidation (FAAO cap; d), mitochondria dependence (Mito dep; e) and glycolytic capacity (Glyco cap; f) expressed in percentage for CD34⁺CD38^{low/-}, CD34⁺CD38⁺, CD34⁻CD38⁺, CD34⁻CD38⁻, CD34⁻CD117⁺, CD34⁺CD117⁺ and CD34⁺CD117⁺HLA-DR⁺CD33⁺CD123⁺ subsets in fresh PB from 13 AML patients at the diagnosis. One-way ANOVA followed by Šidák's multiple comparisons test.

According to Reviewer #2's suggestion, we included a sentence to highlight some limitations of the study:

Discussion: page 13 lines 560-564

By focusing primarily on CD34⁺CD38^{low/-} stem and progenitor subsets, we provided evidence in support of a consistent and standardized approach to evaluating metabolic status in liquid biopsy for AML, minimizing the impact of interindividual variations in different marker expressions. Achieving a better understanding of the metabolic profiles of different LSC phenotypes is crucial to identify new therapeutic targets.

Discussion: page 15, lines 658-660

Although our findings need to be confirmed in future studies powered by adequate sample sizes and across AML subtypes with different molecular statuses, our study highlights a previously unexplored connection between leukemic redox metabolism and EV signatures. This link may have a potential prognostic relevance for AML.

2. To establish the specificity of their results for LSC-like cells, the authors need to compare LSC-like cells with the rest of the bulk AML cells (CD34⁺CD38⁺ or non CD34⁺CD38^{low/-}) in Figure 1 and 2; also demonstrating the difference, if any, between LSC-like cells and the remaining bulk AML cells after EVs treatment.

We have included this analysis in the manuscript, as mentioned in response 1.

Regarding the impact of EV^{AML} on the remaining bulk AML cells, it is important to consider that we isolated and selected CD34⁺ cells using immunomagnetic separation, and after isolation, the remaining bulk of AML cells would primarily consist of CD34-negative cells. Then, we cultured them for 24 hours with or without EVs.

In these conditions, we evaluated the redox metabolic parameters (ROS; GSH; MITO) discriminating CD34⁺ AML cells with the remaining bulk AML cells (CD34 negative fraction) after treatment with vehicle (PBS) or EV^{AML}.

In the figure only for the reviewer, we showed a similar trend when comparing the difference between CD34⁺ AML cells and bulk AML treated with PBS (left side) or with EV^{AML} (right side). However, we only observed a significant difference in the AML cells treated with EV^{AML}, suggesting the potential effects of EV^{AML} in other subsets, as expected.

However, we have concerns about the reliability of analysing the remaining bulk AML after 24 hour in culture and under these conditions. It may not accurately represent the entire AML cell population as we can do with the fresh whole blood as reported in Figure 1 and Figure 2. Therefore, we would not including this figure in the manuscript.

Additionally, we investigated the HSPC differentiation by analysing the frequency distribution of CD34⁺ cells isolated from umbilical cord blood and co-cultured with vehicle (PBS) or EV^{AML} for 24 hours.

We measured the frequency of hematopoietic stem/progenitor cell types using flow-cytometry including hematopoietic stem cell (HSC; CD34⁺ CD38⁻ CD90⁺ CD45RA⁻ CD10⁻), common myeloid progenitor (CMP; CD34⁺ CD38⁺ CD123⁺ CD45RA⁻ CD10⁻), multipotent progenitor (MPP; CD34⁺ CD38⁻ CD45RA⁻ CD90⁻ CD10⁻), lymphoid-primed multipotent progenitor (MLP; CD34⁺ CD38⁻ CD45RA⁺ CD90⁻ CD10⁻), granulocyte-macrophage progenitor (GMP; CD34⁺ CD38⁺ CD123⁺ CD45RA⁺ CD10⁻), and megakaryocyte-erythroid progenitor (MEP; CD34⁺ CD38⁺ CD123⁻ CD45RA⁻ CD10⁻). These populations were

distinguished using a gating strategy similar to Trino et al. (Front Oncol. 2022) and we included the relative gating strategy in Supplementary Figure.

After 24 hours of EV^{AML} treatment, we did not observe a significant reduction in the HSPC subsets in treated CD34⁺ cells.

Accordingly, we added the following sentence in the manuscript:

Page 11 lines 481-482

Firstly, we tested whether EV^{AML} can interfere with HSPC differentiation, and we did not find any significant difference in the HSPC subset proportion (Supplementary 7a-b).

Gating strategy 3. Gating strategy for identification of hematopoietic stem and progenitor cells on CB CD34⁺ based on Trino et al. 2022. Common myeloid progenitor (CMP; CD123+

CD45RA⁻), megakaryocyte-erythroid progenitor (MEP; CD123⁻ CD45RA⁻) and granulocyte-macrophage progenitor (GMP; CD123⁺ CD45RA⁺) were gated from CD34⁺ CD38⁺/CD10⁻ population. Hematopoietic stem cell (HSC; CD90⁺ CD45RA⁻) multipotent progenitor (MPP; CD45RA⁻ CD90⁻), lymphoid-primed multipotent progenitor (MLP; CD45RA⁺ CD90⁻) were gated from CD34⁺ CD38^{low/-} /CD10⁻ population.

b) Frequency of hematopoietic stem/progenitor cell types including hematopoietic stem cell (HSC), common myeloid progenitor (CMP), multipotent progenitor (MPP), lymphoid-primed multipotent progenitor (MLP), granulocyte-macrophage progenitor (GMP), and megakaryocyte-erythroid progenitor (MEP) after treatments with vehicle (PBS) or EV^{AML} detected by FACS analysis (n=8). No significant differences were reported using two-ANOVA.

We also included the method in Supplementary Method section:

Flow cytometric analysis of CB CD34⁺ after EV treatment for HSPC subsets

After treatment with EVAM for 24 hours, CB CD34⁺ cells were washed and incubated with CD90 FITC (clone: 5E10; BD Pharmingen™), CD45RA-BV510 (clone: HI100; Biolegend), CD34 Pe/Cy7 (clone: 4H11; eBioscience™), CD38 APC (clone: HIT2; Biolegend), CD10 APC-H7 (clone HI10a; BD™), CD123 PE (clone: 9F5; BD™) monoclonal antibodies, at room temperature for 15 minutes in the dark. After incubation, cells were washed and resuspended in staining buffer. Then, 10,000 events were acquired on CytoFLEX and analyzed using Kaluza software version 2.1 (Beckman Coulter, Milan, Italy).

3. Providing the functional data demonstrating the significance of separating CD34⁺ cells based on the expression of ROS/GSH and ROS/MITO in Figure 1, and using SCENITH in Figure 2, would enhance the significance of their method.

We agree with the Reviewer that it would be beneficial to separate the fraction for ROS/GSH and MITO and use SCENITH. However, several limitations need to be considered.

We primarily used fresh whole blood for metabolic studies on cells without isolating or manipulating the cells to obtain a more reliable metabolic characterization. This is reported in Figure 1 (by redox metabolic staining) and Figure 2 (by SCENITH approach), respectively. Both methods require fresh whole blood to be processed in less than 1 hour. The protocol takes 1 hour for redox metabolism (see Figure 1) and less than 3 hours for SCENITH. We believe using another isolation method, such as FACS sorting, might further impact cell metabolism due to the pressure exerted by the device and the time required for sorting.

We have developed a relatively faster method for metabolic evaluation, which involves three different dyes to stain our subpopulation. Our method, as detailed in the method section, requires an incubation with the intracellular probe/dye that can be used for the detection of mitochondrial potential, ROS, and glutathione levels. However, after this staining, we are unable to use the stained samples for functional studies. Additionally, we attempted to perform the SCENITH method on the samples labeled for the redox metabolic probe, but the data did not accurately reflect reality. We are concerned that combining the two methods

is creating artifacts in the overall metabolic output due to incorporation of puromycin and previously labelling with dye.

4. Given that both the intermediate group and adverse group exhibit greater mitochondrial dependence compared to the favorable group (Figure 2e), it is necessary to explain the lack of difference between the favorable group and adverse group in Figure 1 I and L. If this lack of difference is due to the methods employed, the authors should clarify which approach is more effective.

We thank the Reviewer for pointing out this question. In our manuscript, we suggested that metabolic characterization in liquid biopsy can help the risk stratification for AML patients. We included more samples for redox metabolic analysis (n = 10) to understand whether the lack of difference between the samples was due to the number of samples. We added new figure for Figure 1i, where we showed a significant difference between ROS^{lo} MITO^{hi} fraction compared to the favourable-risk AML patients versus both intermediate- and adverse-risk patients.

We have included this information in the manuscript, page 5, lines 158-162

In particular, an increase in the fraction ROS^{lo} with MITO^{hi} was observed in both intermediate- and adverse-risk patients compared to favorable-risk patients (P > 0.05, respectively) (Fig 1i). Only intermediate-risk AML patients showed an increase in ROS^{lo} GSH^{hi} fraction compared to favorable-risk AML patients (P < 0.05) (Fig. 1l).

However, we did not find a similar trend for the fraction ROS^{lo} GSH^{hi} in Figure 1l.

The two methods used for metabolic analysis are complementary but not similar. The redox metabolic method measures the frequency of subsets where mitochondrial potential is detected in combination with ROS and GSH levels. On the other hand, the SCENITH

method does not consider ROS levels and GSH content. Instead, it detects mitochondrial dependence based on the proportion of Protein Synthesis reported by puromycin dependent on OXPHOS reporting a different readout.

We also updated all the graphs in Figure 1, adding the new sample tests and the text accordingly.

Accordingly, we also updated Figure 2, adding the new samples tested using SCENITH. Now, we change the Figure and the Result sections, accordingly:

Accordingly, we observed that CD34⁺ cells relied primarily on glucose oxidation (glucose dependence; mean percentage 83,77%; mito-dependence mean percentage 21,42%), exhibiting a high glycolytic capacity (glyco-cap, mean percentage 78,58%); at the same displaying a low percentage of FAO capacity (mean percentage 16,23%), indicative of their inability to use fatty acids and amino acids as sources for ATP production (Fib 2b). *We observed that AML patients in the favorable group had a reduction in mitochondrial dependence at the expense of glycolysis. Specifically, favorable AML patients unveiled a mitochondrial dependence equal to 8.2% compared to the adverse and intermediate groups (mean percentage: 23,5% and 22,3%, respectively; $P < 0.05$ versus adverse group) (Fig. 2e). Moreover, intermediate-risk AML patients revealed a higher glucose capacity compared to adverse group ($P < 0.05$; Fig. 2e).*

5. It is interesting that EV markers are differentially expressed in adverse-risk and intermediate-risk AML patients. To further strengthen their findings, the authors should explore whether EVs from these two AML categories have differing effects on modifying AML LSC metabolism. Additionally, if there is a method to eliminate EVs and test the metabolic status and engraftment efficiency of AML cells (e.g., transplantation with or without EVs), it would be crucial in highlighting the role of EVs.

We have addressed Reviewer #2's comment by including new data and Figures that demonstrate the impact of EVs derived from AML patients with different risks. Our updated analysis involved additional samples using CD34⁺ AML cells or the human cell line MOLM-13. Furthermore, we performed new in vitro and in vivo experiments using the human cell line OCI-AML3. We added a new main Figure (6) and adjusted the results and overall manuscript accordingly (as reported below).

We introduced the work on cell lines (adding the results with OCI-AML3) in Result section:

Page 10, from lines 389 and updated the relative figure in Supplementary Figure 5a-i

Human leukemia cell lines show a different response to EV^{AML} in redox metabolism
Then, we simultaneously detected ROS and GSH levels along with mitochondrial potential as reported above for whole blood on human cell lines from AML patients: KG-1, Kasumi-1, MV-4-11, OCI-AML3 and MOLM-13 (Supplementary Fig. 5a-f).

When the MOLM-13, KG-1 and OCI-AML3 were cocultured with EV^{AML}, we noticed that MOLM-13 only showed a lower percentage of ROS^{hi} MITO^{hi} in the presence of EV^{AML} ($P < 0.01$; Supplementary 5a), whereas OCI-AML3 showed an increase in the fraction ROS^{lo} MITO^{hi} ($P < 0.05$; Supplementary Fig. 5g). Moreover, both MOLM-13 and OCI-AML3

showed a significant increase in ROS^loGSH^hi fraction in the presence of EV^{AML} ($P < 0.05$, respectively; Supplementary Fig. 5c-h). In addition, MOLM-13 and KG-1 shared a considerable reduction in ROS^hiGSH^lo ($P < 0.01$ for MOLM-13 and $P < 0.05$ for KG-1) and a concomitant decrease in the fraction GSH^loMITO^hi ($P < 0.05$, respectively). Only, KG-1 cell line presented an increase in the fraction with high GSH content and high mitochondrial potential ($P < 0.05$: Supplementary Fig. 5f).

Conversely, EV^{AML} showed a limited effect on redox metabolic markers in the MV-4-11 and Kasumi-1 cell lines (Supplementary Fig. 6a-f). Only the fraction of ROS^hiMITO^hi cells decreased in response to EV^{AML} in MV-4-11 cells ($P < 0.05$; Supplementary Fig. 6a). In summary, EV^{AML} showed a mild effect on human leukemia cell lines, mainly decreasing the ROS^hiGSH^lo fractions in the KG-1 and MOLM-13 cell lines. This suggests that EV^{AML} might have a more evident role in specific LSC metabolism.

Supplementary Figure 5

Metabolic redox modulation by EVs of human leukemic cell lines from human leukemic cell lines including MOLM-13 (a, c, e), KG-1 (b, d, f) and OCI-AML3 (g, h, i).

*a, b, g) ROS/MITO sub-fractions in leukemic cells treated for 24 hours with vehicle (PBS) or EV^{AML}: ROS^{hi} MITO^{lo}, ROS^{hi} MITO^{hi}, ROS^{lo} MITO^{hi}; c, d, h) ROS/GSH sub-fractions in leukemic cells treated for 24 hours with vehicle (PBS) or EV^{AML}: ROS^{hi} GSH^{lo}, ROS^{hi} GSH^{hi}, ROS^{lo} GSH^{hi}. e, f, i) GSH/MITO sub-fractions in leukemic cells treated for 24 hours with vehicle (PBS) or EV^{AML}: GSH^{hi} Mito^{lo}, GSH^{hi} Mito^{hi}, GSH^{lo} Mito^{hi}. Significant differences were reported using Mann–Whitney test for unpaired samples. *p* values <0.05 (*), <0.01 (**) were considered significant.*

In response to the Reviewer's question, to emphasize the impact of EV^{AML} in adverse or intermediate-risk patients, we introduced the new Figure 6 (using cell lines), in which we performed experiments in vitro and in vivo using both OCI-AML3 and MOLM-13 and discriminated the effects between EV from different ELN risks. We firstly investigated the metabolic parameters of our interest and, secondly, the resistance to Venetoclax treatments mediated by EV^{AML}, in vitro.

Figure 6

EV^{AML} from different AML risks have distinct metabolic effects on OCI-AML3 and MOLM-13, and adverse-risk EV^{AML} enhance the engraftment of MOLM-13 in vivo.

We then investigated how circulating EV^{AML} from different ELN risk groups affects the metabolism of human cell lines. Our focus was on the two cell lines, OCI-AML3 and MOLM-13, derived from PB of AML patients and which were more responsive to EV^{AML}.

For OCI-AML3, all three types of EV^{AML} were found to significantly increase the proportion of cells with low ROS levels and high GSH content ($P < 0.0001$, respectively). Additionally, only adverse EV^{AML} increased the proportion of ROS^{lo} and high mitochondria (MITO^{hi}) ($P < 0.05$), while significantly decreasing the proportion of ROS^{hi} and GSH^{hi} ($P < 0.01$) (Fig. 6a-c).

For MOLM-13, all three types of EVs were observed to reduce the frequency of cells with high ROS and low GSH (ROS^{hi} GSH^{lo}) ($P < 0.0001$ for fav EV^{AML} and $P < 0.01$ for both intermediate and adverse EVs). However, only adverse EV^{AML} significantly decreased the percentage of cells with low GSH and high mitochondria potential (GSH^{lo} MITO^{hi}) ($P < 0.05$) (Fig. 6d-f).

We therefore examined whether pretreatment with EV^{AML} might enhance resistance to Venetoclax, a selective inhibitor of BCL-2 that has advanced treatment options for AML patients⁵. We performed our experiments on MOLM-13, since OCI-AML3 have been reported as resistant to Venetoclax⁶.

Notably, pretreatment with EV^{AML} increased the survival of MOLM-13 cells treated with Venetoclax for 24 hours ($P < 0.01$). However, we did not observe a difference when we stratified EV^{AML} according to risk (Fig. 6g).

Finally, we performed in vivo experiments. Briefly, we treated luciferase-transduced MOLM-13 or OCI-AML3 cell lines with EVs from adverse—or intermediate—or favorable-risk AML patients and intravenously injected them into NOD-scid IL2R gamma null (NSG) mice (n=40, two independent experiments). The engraftment efficiency of human leukemia cell lines was assessed weekly using bioluminescence imaging.

In the figure only for the Reviewer, it is evident that the experiments conducted with OCI-AML3 showed that pre-treatments with EV^{AML} from adverse- or intermediate-risk patients significantly enhanced the engraftment of luc-mCherry OCI-AML3 compared to control cells (vehicle). However, due to signal dispersion from deep tissue or within the tumor, the exact source of the signal could not be identified and quantified.

Whole-animal bioluminescence imaging from day 18. Ventral and dorsal view images from NSG mice for each group are shown. Mice transplanted with luc-mCherry OCI-AML3 cells pre-treated for 24 hours with vehicle control (PBS, n=5) (a-b), high-risk EV^{AML} (n=5) (c-d) or intermediate-risk EV^{AML} (n=5) (e-f).

Consequently, we deemed it inappropriate to present this data without a proper quantification to the Readers.

Therefore, we propose to display the data only from the second experiment, which involved NSG mice transplanted with luc-mCherry MOLM-13, pre-treated with EVs from favorable/intermediate/adverse risk AML patients, and untreated (vehicle). This information will be presented clearly and concisely in the manuscript, and we believe it will accurately communicate our findings.

We added the relative procedure and experiments in the Figure 6 (h-i) and in the Method sections, as outlined below:

Finally, we conducted an *in vivo* study to evaluate the effect of EV^{AML} pre-treatment on MOLM-13 or OCI-AML3 cells xenografted in NSG mice. We pre-treated Luc-mCherry MOLM-13 and OCI-AML3 cells with EV^{AML} or with vehicle control (PBS) prior to tail vein infusion and engraftment. Our results showed that pre-treatment with EV^{AML} from adverse-risk AML patients was more effective in increasing AML xenograft growth compared to treatment with a vehicle control (PBS). Interestingly, after 18 days, the treatment with EV^{AML} from adverse risk significantly improved the engraftment of Luc-mCherry MOLM-13 cells compared to untreated cells ($P < 0.001$, Fig. 6h-i). However, EV^{AML} from intermediate and favorable patients did not affect the engraftment of Luc-mCherry MOLM-13 cells (Fig.6h-i). A similar pattern, although characterized by dispersion of the signal, was observed when OCI-AML3 cell line was used (data not shown).

According to *in vitro* experiments, our *in vivo* findings indicate that EV^{AML} from adverse-risk AML patients may have an increased capacity for accelerating *in vivo* leukemia cell engraftment compared to EV^{AML} collected from intermediate-risk or favorable AML patients.

Figure 6

Figure 6

Figure 6.

Redox metabolic profiling for OCI-AML3 and MOLM-13 treated for 24 hours with EV^{AML} from adverse, intermediate or favorable-risk AML patients. ROS/MITO subsets expressed as percentage were reported for OCI-AML3 (a) or MOLM-13 (b). Percentage of ROS/GSH subsets for OCI-AML3 (b) and MOLM-13 (e). Percentages of GSH/MITO subsets for OCI-AML3 (a) and MOLM-13 (f) treated with favorable EV^{AML} (n=9), intermediate EV^{AML} (n= 8) or adverse EV^{AML} (n=8) from 8 independent experiments. Significant differences were reported by two-way ANOVA with Šidák's multiple comparisons test. g) MOLM-13 pre-treated with EV^{AML} for 4 hours before adding Venetoclax (150 nM) for 24 hours. MOLM-13 treated with vehicle or favorable EV^{AML} (n=3) or intermediate EV^{AML} (n=4) or adverse EV^{AML} (n=5) from 5 independent experiments. h-i) Whole-animal bioluminescence imaging from day 18. Ventral and dorsal view images from mice NGS for each group are shown. Each animal's region of interest (ROI) was defined at every time point (inset). Mice transplanted with luc-mCherry MOLM-13 cells pre-treated for 24 hours with vehicle control (PBS, n=5), adverse-risk EV^{AML} (n=5) or intermediate-risk EV^{AML} (n=5). The graph shows the quantification of bioluminescence as the average total photon flux per second from days 6 to 18 after initiation. Error bars represent \pm SEM.

We also repeated experiments on AML CD34⁺ isolated from patients at diagnosis, and we included the data where we tested the effects of EV^{AML} from different ELN risk categories.

We updated the old Figure 6 which is now Figure 7 and relative Supplementary Figure 8 a, b,c. We removed the Volcano plot from Nanostring experiments and we transfer in Supplementary 8d.

Figure 7

e) Percentages of ROS^{hi}MITO^{hi} and ROS^{lo}MITO^{hi} for AML CD34⁺ cells treated with favorable EV^{AML} (n=9), intermediate EV^{AML} (n=7) or adverse EV^{AML} (n=8) from 10 independent experiments. Significant differences were reported by two-way ANOVA with Šidák's multiple comparisons test. P values <0.01 (**), <0.001 (***), <0.0001 (****) were considered significant.

Supplementary Figure 8a,b,c:

Redox metabolic profiling for AML CD34⁺ cells treated for 24 hours with EV^{AML} from adverse, intermediate or favorable-risk AML patients. ROS/MITO subsets expressed as percentages were reported for AML CD34⁺ cells (a). Percentage of ROS/GSH subsets for AML CD34⁺ cells (b). Percentages of GSH/MITO subsets (c) for AML CD34⁺ cells treated with

favorable EV^{AML} ($n=9$), intermediate EV^{AML} ($n=7$) or adverse EV^{AML} ($n=8$) from 10 independent experiments. Significant differences were reported by two-way ANOVA with Šidák's multiple comparisons test.

Result section on Figure 7:

EVs from AML patients alter the redox metabolism of CD34⁺ AML cells, modulating the GSH/GPX4 axis

.....

When we cocultured AML CD34⁺ with EV^{AML} , combining ROS with mitochondrial functionality, we did not find any significant differences (Fig. 7a). However, we found a significant decrease in ROS levels and an increase in MITO levels after EV^{AML} treatment, as reported by MFI values ($P < 0.05$; Fig. 7d). As a result, within the leukemic CD34⁺ progenitor compartment, coculture with EV^{AML} significantly reduced the proportion of ROS^{hi} GSH^{lo} cells (mean percentage: 53,91%) compared to that in AML CD34⁺ cells treated with vehicle (mean percentage: 76,21%) ($P < 0.01$; Fig. 6b), as previously reported in human cell lines (KG-1 and MOLM-13, Supplementary Fig. 5c-d). Moreover, we observed an increase in the frequency of ROS^{hi} GSH^{hi} fraction in the presence of EV^{AML} ($P < 0.01$; Fig. 7b). Again, similar to what we observed for the KG-1 cell line (supplementary Fig. 5f), we also confirmed in AML CD34⁺ cells an increase in GSH^{hi} MITO^{hi} fractions ($P < 0.002$) and a reduction in GSH^{lo} MITO^{hi} fractions ($P < 0.01$; Fig. 7c).

We then analyzed the metabolic effects by sorting the data based on EV^{AML} from different ELN risk groups, we observed that all three subtypes of EV^{AML} significantly increased the frequency of CD34⁺ cells with high levels of GSH and mitochondria (MITO), and significantly decreased the frequency of CD34⁺ cells with high levels of ROS and low levels of GSH, as well as the proportion of cells with GSH^{lo} and MITO^{hi} (Supplementary Fig. 8a-c). However, only intermediate EV^{AML} were able to increase the proportion of CD34⁺ cells with low ROS and high MITO, whereas only favorable EV^{AML} increased the proportion of CD34⁺ cells with high ROS and high GSH (Supplementary Fig. 8a-b).

Significantly, unlike EVs from intermediate ($P < 0.001$) and adverse-risk patients ($P < 0.01$), EV^{AML} from favorable AML patients increased the proportion of CD34⁺ cells with high levels of ROS and high mitochondrial potential (Fig. 7e) as compared to untreated cells ($P < 0.01$; Supplementary Fig. 8a) and decreased the proportion of CD34⁺ cells with low levels of ROS and high mitochondrial potential as compared to intermediate ($P < 0.01$) and adverse-risk EV^{AML} ($P < 0.05$; Fig. 6c). These data suggest a metabolic vulnerability driven by EVs from different ELN risk categories.

Collectively, these results demonstrate that circulating EVs from AML patients may trigger AML CD34⁺ stem cells toward an increase in both mitochondrial potential and GSH levels, reducing ROS levels and showing a leukemia-dependent mechanism partially reverted by GPX4 inhibition. Understanding the metabolic dependencies mediated by EVs from different ELN risk groups can allow for the development of tailored therapeutic strategies aimed at exploiting new metabolic vulnerabilities.

We included the relative materials and methods in the Supplementary method section

Human leukemia cell line transduction

To label the cells for *in vivo* transplantation, the OCI-AML3 and MOLM-13 cell lines were stably transduced with a lentiviral vector expressing stably luciferase and mCherry, MI-Luciferase-IRES-mCherry (w168-1; Addgene plasmid, #75020), carrying mCherry as a selective marker. To infect human AML cell lines, 293T were cotransfected with 10 µg of lentiviral vector, 3 µg PMD2G envelope plasmid, 2.5 µg of REV packaging plasmid, and 5 µg of RRE transfer plasmid using calcium phosphate transfection system. After 16 hours, the media was removed and replaced with 5 mL of fresh medium. Viral supernatant was then collected at 24 and 48 hours and added directly to human AML cell lines plated at 1×10^5 supplemented with 8 µg/mL of polybrene (Merck). Spin-infection was performed for 1.30 hours at 37°C twice a day for one separate day. After 48 hours from the first cycle of infection, infected cells were sorted using FACS Fusion II (BD Bioscience).

In vivo experiments

Experiments involving animals were approved by the Italian Ministry of Health and have been done in accordance with the applicable Italian laws (D.L.vo 26/14 and following amendments), the Institutional Animal Care and Use Committee, and the institutional guidelines at the European Institute of Oncology.

To determine the *in vivo* effects of human cell lines treated with EV^{AML} on leukemia progression and engraftment, tandem tagged luciferase-mCherry human cell lines (OCI-AML3 and MOLM-13) were seeded in a 24-well plate (20×10^4) and treated with EV^{AML} (15 µg) from AML patients stratified according to ELN2022 risk ($n=5$ for each risk group, respectively) or PBS/vehicle as control. After 24 hours, human cell lines were washed and counted before transplantation. Luc-mCherry AML cells (1×10^5) treated with EV^{AML} or vehicle control (PBS) were injected with a 26-gauge needle into the lateral tail vein of 6–8-week-old male NOD-scid IL2R gamma null (NSG) mice ($n=5$ /group; two independent experiments). Mice were randomly assigned to different condition cohorts. All the mice were monitored twice a week and imaged using bioluminescence imaging to document engraftment. Animals were anesthetized for imaging *in vivo* using 2% Isoflurine (flow rate 1 L/min O_2). Spectral imaging was commenced 10 min post IP injection to allow stabilization of light output. Mice were injected with 150 mg/kg of D-Luciferin and imaged twice a week to monitor disease status and treatment efficacy. As a surrogate for tumor burden, bioluminescence was quantified using Living Image Software (version 4.7.2, PerkinElmer).

Minor

comments:

1. Please provide a representative flow gating strategy for experiments using CD34+CD38low/- cells.

We thank the Reviewer for pointing this out. We have added the three gating strategies we used for our analysis in the Supplementary figure.

Accordingly, we have added our gating strategy in the Supplementary Fig. 1a-e and in Supplementary Fig. 7c

Gating strategy 1

a) Gating strategy 1. Blast populations were identified by CD45^{low/-}/SSC gating strategy to define and analyze mainly CD34⁺ stem cells, immature CD34⁺ CD38^{low/-} stem cells and CD34⁺ CD38⁺ progenitor cells. For redox metabolic analysis, the CD34⁺ leukemic subsets were analyzed in combination with CellROX (ROS), MitoTracker CMXRos (MITO), and Thiol Tracker (GSH) using two-by-two gating on CD3⁺ cells as a reference marker (lympho CD3⁺ gate).

e) Gating strategy 2. Blast populations were identified by $CD45^{low}/SSC$ gating strategy to define and analyze immature and progenitor leukemic cells ($CD34^+$ and/or, $CD117^+$), myeloid cells ($CD33^+$, $HLA-DR^+$), and primitive LSC ($CD123^+$). For redox metabolic analysis, the leukemic subsets were analyzed in combination with CellROX (ROS), MitoTracker CMXRos (MITO), and Thiol Tracker (GSH) using two-by-two gating based on $CD3^+$ cells as reference marker (lympho $CD3^+$ gate).

Gating strategy 3. Gating strategy for identification of hematopoietic stem and progenitor cells on CB $CD34^+$ based on Trino et al. 2022. Common myeloid progenitor (CMP; $CD123^+ CD45RA^-$), megakaryocyte-erythroid progenitor (MEP; $CD123^- CD45RA^-$) and granulocyte-macrophage progenitor (GMP; $CD123^+ CD45RA^+$) were gated from $CD34^+ CD38^+/CD10$

population. Hematopoietic stem cell (HSC; CD90⁺ CD45RA⁻) multipotent progenitor (MPP; CD45RA⁻ CD90⁻), lymphoid-primed multipotent progenitor (MLP; CD45RA⁺ CD90⁻) were gated from CD34⁺ CD38^{low/-} /CD10⁻ population.

2. On line 123, the authors stated that “no significant differences were found in paired samples between the two sources”. However, statistical data for the PB CD34⁺ group and BM CD34⁺ group are still necessary (Figure 1d-f). In addition, please provide statistical data for Figure 2b, 2c, and Supplementary Figure 2b (group Mito dep and Glyco cap).

We thank the Reviewer for highlighting this missing data. We have now added the relative statistics to the data and changed the graphical representation to help readers understand the results.

Page 26 lines 1195-1196

Statistical significance was determined by two-way ANOVA with Šidák's multiple comparisons test.

3. On line 217, “Fig. 2e” should be “Fig. 3e”.

4. Please provide quantitative data for Figure 3e.

We thank the Reviewer for pointing it out.

We utilized WB to detect proteins in the EV-containing preparation and normalized the EV preparation based on the total protein amount (10 µg). However, our aim is to demonstrate the presence or absence of specific proteins rather than their quantification. Various variables in the EV field, such as preparation protocol, heterogeneity, and the absence of loading control as reported for the cells with actin, were taken into consideration.

As a result, we have identified at least one protein from each of the following categories in our extracellular vesicles (EVs): 1) transmembrane protein (e.g. CD81), 2) cytosolic protein (e.g. flotilin), and 3) major components of non-EV co-isolated structures (e.g. apolipoprotein and albumin). Additionally, as an optional measure, we included 4) endoplasmic reticulum

(ER) membrane protein (e.g. calnexin) as a cell-specific 'exclusion' marker to demonstrate EV purity.

We intentionally omitted quantification to avoid conveying a misleading message that could lead to a comparison between samples for the expression of the proteins detected. Our goal is to show only the presence or relative absence of the proteins in our EV preparation.

5. The figure legend of Figure 6f is missing.

We thank the Reviewer for pointing out this. We have now included the relative figure legend f.

6. The resolution of Figure 5f and Figure 6e is too low to be appreciated. We thank the Reviewer for pointing out this. We have now included figures with high resolution accordingly.

7. Please discuss why EV-driven metabolic adaptation has no effects on BM CD34+ blasts.

We appreciate the Reviewer for bringing up this point. We find it very interesting and believe it needs to be addressed. We have included a potential reason that could explain the different effects of EV^{AML} on PB or BM samples.

We attempted to assess the uptake of circulating EV^{AML} from AML patients in paired mononuclear cells (MNC) from PB and BM. We observed that the uptake of EV^{AML} by BM cells was lower compared to PB, suggesting that BM cells may require a different amount of EV^{AML} or more time to be uptaken. In line with the Reviewer's suggestion, we have added a possible explanation in the Discussion section for Supplementary Fig. 8i.

i

We added the following sentence in Discussion:

Our data showed that BM AML cells had lower EV uptake compared to PB AML cells from the same patients, which might explain the lower responsiveness of BM CD34⁺ cells (Supplementary Fig. 8h-i).

8. Many references (e.g., #8-15, 24, 25) lack complete information, such as page numbers and years. We apologize for the lack of information in our references, we have now updated the references accordingly.

Reviewer #3, expertise in mass-spec based metabolomic profiling (Remarks to the Author):

Forte et al. analyzed peripheral blood from a cohort of AML patients at diagnosis for risk stratification. Specifically, they assessed markers of metabolic activity of circulating tumor cells and the molecular composition of extracellular vesicles. They applied a battery of assays/technologies to characterize different molecular layers. A key finding were distinct metabolic and redox profiles of intermediate risk patients, which adds to the growing body of work in this field. However, I do have concerns regarding this manuscript that should be addressed prior to publication.

Major points:

1) The statistical power of this study appears relatively low. This is, even though the overall study comprises 100 patients as per Suppl. Tab. 1., several analyses were performed with much smaller sub-cohorts with no further explanation. This is a major limitation and should be discussed more transparently.

We thank the Reviewer for bringing this to our attention.

We have added more samples to our study and inserted a table reporting which samples were used for the main analyses in the Source Data File.

UPN	Age (Y)	Sex	ELN_2022	WBC (x 10 ⁹ /L)	EV profiling by MACSplex	EV isolation for in vitro studies	AML with CD34 ⁺ used for staining on whole blood or separated for in vitro studies	Metabolic profile (SCENITH) on whole blood (CD34 ⁺ cells)	Metabolic profile (REDOX profiling) on whole blood (CD34 ⁺ cells)	Metabolomic and Lipidomic on EVs
ID_1	65	M	adv	16,43	Y	Y	Y	Y	Y	Y
ID_2	62	M	adv	1,19	Y	Y	Y	Y	Y	Y
ID_3	79	F	adv	1,63	Y	Y	Y	Y	Y	
ID_4	60	M	adv	18,58	Y	Y	Y	Y	Y	
ID_5	75	M	adv	10,45	Y	Y	Y	Y	Y	Y
ID_6	53	M	adv	25,47	Y	Y	Y	Y	Y	Y
ID_7	69	F	adv	1,08	Y	Y	Y	Y	Y	Y
ID_8	58	F	adv	19,72	Y	Y	Y	Y	Y	
ID_9	49	M	adv	45,21	Y	Y	Y	Y	Y	Y
ID_10	66	M	adv	1,67	Y	Y	Y	Y	Y	
ID_11	64	F	adv	1,37	Y	Y	Y	Y	Y	
ID_12	73	F	adv	1,82	Y	Y	Y		Y	

ID_13	63	F	adv	5,60	Y	Y				Y
ID_14	52	M	int	28,80	Y	Y	Y	Y	Y	
ID_15	79	F	int	10,28	Y	Y	Y	Y	Y	Y
ID_16	84	M	int	2,07	Y	Y	Y	Y	Y	Y
ID_17	69	M	int	0,94	Y	Y	Y	Y	Y	Y
ID_18	48	F	int	2,53	Y	Y	Y	Y	Y	Y
ID_19	72	M	int	27,48	Y	Y	Y	Y	Y	Y
ID_20	69	M	int	2,90	Y	Y	Y	Y	Y	
ID_21	52	M	int	0,73	Y	Y	Y		Y	Y
ID_22	68	M	int	20,00	Y	Y	Y		Y	Y
ID_23	35	M	int	164,10	Y	Y				Y
ID_24	70	M	int	1,79	Y	Y	Y	Y		Y
ID_25	38	F	int	1,69	Y	Y	Y	Y		
ID_26	22	M	int	11,40	Y	Y				Y
ID_27	54	M	low	7,20	Y	Y	Y	Y	Y	Y
ID_28	82	F	low	5,30	Y	Y	Y	Y	Y	Y
ID_29	68	F	low	115,57	Y	Y	Y		Y	Y
ID_30	62	M	low	1,00	Y	Y				Y
ID_31	55	M	int	11,15	Y	Y	Y		Y	
ID_32	54	F	int	3,59	Y	Y	Y		Y	
ID_33	64	M	int	2,87	Y	Y	Y	Y	Y	
ID_34	70	M	adv	10,09	Y	Y				
ID_35	64	M	int	0,91	Y	Y	Y	Y		
ID_36	77	M	int	2,05	Y	Y	Y	Y		
ID_37	83	F	low	31,94	Y	Y				
ID_38	83	M	adv	7,68	Y	Y				
ID_39	46	M	adv	2,04	Y	Y	Y			
ID_40	76	M	low	111,05	Y	Y	Y			
ID_41	18	F	low	N/A	Y	Y				Y
ID_42	71	F	adv	26,57			Y		Y	
ID_43	72	M	adv	2,45			Y		Y	
ID_44	71	M	adv	2,28					Y	
ID_45	77	F	adv	3,22			Y		Y	
ID_46	72	M	adv	9,78			Y		Y	
ID_47	60	M	adv	5,89			Y		Y	
ID_48	62	F	adv	1,07			Y		Y	
ID_49	50	M	adv	1,70			Y		Y	
ID_50	75	M	adv	1,87			Y	Y	Y	
ID_51	55	F	adv	2,38			Y		Y	Y
ID_52	66	F	int	9,01			Y		Y	
ID_53	72	F	int	0,96			Y		Y	
ID_54	78	M	int	19,34			Y		Y	

ID_55	67	F	int	1,88			Y		Y	
ID_56	58	F	int	3,18			Y		Y	
ID_57	71	M	int	1,06			Y		Y	
ID_58	75	M	int	1,26			Y		Y	
ID_59	64	F	int	6,61			Y	Y	Y	
ID_60	75	M	int	0,98			Y	Y	Y	
ID_61	47	F	int	25,07			Y		Y	
ID_62	49	F	int	1,76			Y		Y	
ID_63	71	M	int	0,78			Y	Y	Y	
ID_64	67	F	int	0,97			Y		Y	
ID_65	44	M	int	2,23			Y		Y	
ID_66	37	M	low	22,46			Y		Y	
ID_67	65	M	adv	2,77		Y				
ID_68	72	M	int	3,64		Y				Y
ID_69	67	F	adv	22,18		Y				
ID_70	79	F	adv	2,68		Y				
ID_71	75	F	low	4,95		Y				Y
ID_72	73	F	low	145,30		Y				Y
ID_73	54	M	int	68,20		Y				Y
ID_74	55	M	low	2,00		Y				Y
ID_75	27	M	adv	235,00		Y				Y
ID_76	52	F	low	3,60		Y				Y
ID_77	51	M	adv	88,63		Y				Y
ID_78	84	M	int	204,43		Y				Y
ID_79	50	M	adv	12,30		Y				Y
ID_80	63	M	int	178,00		Y				Y
ID_81	30	F	low	262,49		Y				Y
ID_82	58	M	adv	340,00		Y				
ID_83	69	F	adv	37,04			Y		Y	
ID_84	85	F	adv	7,63			Y		Y	
ID_85	74	M	adv	38,00			Y		Y	
ID_86	71	M	adv	1,25			Y		Y	
ID_87	55	M	adv	1,82			Y		Y	
ID_88	71	M	adv	2,66			Y		Y	
ID_89	71	M	int	3,64			Y		Y	
ID_90	74	M	int	280,00			Y			
ID_91	62	M	int	3,50			Y		Y	
ID_92	58	M	adv	1,14			Y		Y	
ID_93	61	M	adv	20,51						
ID_94	77	M	int	9,12			Y		Y	
ID_95	73	M	adv	2,90						

ID_96	85	F	low	0,96			Y		Y	
ID_97	46	M	adv	2,90			Y	Y		
ID_98	79	M	low	1,90			Y	Y		
ID_99	84	F	low	31,94			Y	Y		
ID_10 0	83	M	adv	19,29			Y	Y		
ID_10 1	35	F	adv	2,21			Y	Y		
ID_10 2	71	F	adv	3,42			Y	Y		
ID_10 3	74	M	adv	3,74			Y	Y		
ID_10 4	70	M	adv	10,00			Y	Y		
ID_10 5	75	M	adv	26,85			Y	Y		
ID_10 6	74	F	adv	324,7 8			Y	Y		
ID_10 7	74	M	adv	1,58			Y	Y		
ID_10 8	76	M	adv	35,03			Y	Y		
ID_10 9	61	M	low	1,27			Y	Y		
ID_11 0	81	M	low	53,15					Y	
ID_11 1	77	F	int	1,23			Y			
ID_11 2	73	M	int	67,68			Y			
ID_11 3	68	M	int	2,61			Y			
ID_11 4	76	M	int	21,02			Y			

We need to highlight that most of the experiments in this study were carried out using fresh samples to accurately reflect, represent and capture the cellular metabolic status. For all the data from Figure 1 and Figure 2 we used only fresh blood that we processed within a few hours, taking into consideration all variables that can affect the cell metabolism (e.g., fasting) or EV composition (e.g. temperature, timing). We performed multiple experiments simultaneously in a subgroup of patients to correlate cellular and extracellular readouts, as reported by table above. All these aspects limited our sample size to a sub-cohort of patients. However, in the manuscript, we highlighted the limitations of our study

Discussion (conclusion)

Although our findings need to be confirmed in future studies powered by adequate sample sizes and across AML subtypes with different molecular statuses, our study highlights a previously unexplored connection between leukemic redox metabolism and EV signatures.

2) In view of the above, some claims appear overstated, in particular the 'prognostic relevance' of EV profiles.

We thank the Reviewer for pointing this out, and as a result, we have revised our manuscript adding the adjective 'potential' to limit our claim.

Page 1, abstract *Of note, the EVAML profiling demonstrates potential prognostic relevance.*

Page 7 line 271-271 *Circulating EVs from newly diagnosed AML patients are mainly enriched in CD44, and multiplex protein analysis reveals potential prognostic value*

Page 8, lines 326-328 *Interestingly, the expression of EV markers is closely linked to cell metabolism, and the depletion of selected markers on EVs defines intermediate- and adverse-risk patients, supporting an EV signature with potential prognostic power.*

Page 15, line 685 *This link may have a potential prognostic relevance for AML.*

We have included more data and in vivo experiments to support our claim further as reported below.

3) The authors claim that their approach is 'real-time'. However, there is no mention about the timing of the assay and no data are provided that would support an application in a real-world clinical setting.

We primarily used fresh whole blood for metabolic studies on cells without isolating or manipulating the cells to obtain a more realistic metabolic characterization. This is reported in Figure 1 (by redox metabolic staining) and Figure 2 (SCENITH approach), respectively. Both methods require fresh whole blood to be processed in less than 1 hour. The protocol takes 1 hour for redox metabolism (see Figure 1) and less than 3 hours for SCENITH.

However, we removed the term 'real-time' to avoid confusion. According to the Reviewer's suggestion, we added the following sentence to the Discussion

Page 13, lines 590-592

In contrast to other approaches, SCENITH technology applied to hematological malignancies may provide a live snapshot of the central energy metabolism of AML cell subsets and LSCs without creating metabolic artifacts. This innovative method enables the utilization of fresh whole blood and facilitates processing within just a few hours.

4) I was unable to review the quality of the lipidomics data as the Supplementary File mentioned in the reporting summary is missing (Suppl. Tab. 3 is insufficient for that purpose). Guidance may be found in PMID: 35934691. I would also strongly encourage the authors to make the MS raw files publicly available.

According to the Reviewer #3's request, we also included the MS raw data using the guidelines suggested.

Please see Source Data File.

5) The authors state 'the most highly expressed molecular species were tested as putative biomarkers detected in EVs...'. It is not clear on which cohort this test was performed. If done on the same dataset as the PLS-DA analysis, this is a redundant result and no

evidence for a potential biomarker panel; particularly given the pertinent risk of overfitting in PLS-DA.

We agree with Reviewer #3, and we corrected the sentence to avoid redundancy and only demonstrate its application.

Page 9, lines 349-351

~~The most highly expressed molecular species were used for the multivariate ROC analysis and unveiled a promising area under the curve (AUC = 0.98) (Fig. 5e), showing their potential as biomarkers.~~

The most highly expressed molecular species were used for the multivariate ROC analysis, which revealed a promising area under the curve (AUC = 0.98) (Fig. 5e) for further investigation as potential biomarkers.

6) Jayavelu et al. recently described a 'Mito-AML' subtype (PMID: 35245447). The authors should discuss their work in the context of this study.

We thank the reviewer for the suggestion and have now included a citation of this work, which is highly relevant to our focus.

Discussion section, page 14, lines 603-608

According to our research, A.K. Jayavelu et al. ⁷ have identified a new high-risk subtype of AML, defined as 'Mito-AML', characterized by high expression of mitochondrial proteins and associated with poor outcomes. Our data support that increased mitochondrial potential or dependence is associated with intermediate and adverse-risk AML. In addition, we reported a high glucose dependence exclusively in intermediate-risk AML patients, revealing that metabolic profiling can potentially predict clinical features.

Minor

points:

7) L280: please specify 'network analysis' and associated significance levels. What is the meaning of the numbers in Fig. 5b?

We thank the reviewer and we have included the additional information as requested.

We added the following sentence to the Legend Figure 5, page 28 line 1285-1286

The lines connecting the nodes indicate the interaction's direction and status. The numbers (z-score) indicate the intensity of the reaction on an arbitrary scale.

8) The resolution in some panels of Figs. 5 and 6 is not sufficient.
9) Several data are 'not shown'. Why?

We chose to exclude specific details to avoid overwhelming the essential results and to prevent confusion with relevant data. However, we removed the data that was not shown and added new figures that were necessary to better understand our messages.

As suggested by Reviewer #3, we have removed the data not shown (page 6, Lines 235-238) (page 9, line 357) and included the figure depicting the uptake of EVs labeled with the green membrane dye PKH67 in AML CD34⁺ cells within 24 hours.

We have integrated the following text into the manuscript:

page 9, line 357

EV isolated from AML patients were stained with the green membrane dye PKH67 and then co-cultured with AML CD34⁺ for 4 to 24 hours at 37°C. The uptake of EV^{AML} was measured by flow cytometry (Supplementary Fig.). To rule out passive uptake, a negative control was conducted using PKH67 dye without EVs, or by keeping the AML CD34⁺ cells at 4°C to inhibit uptake. The data confirmed the EV uptake within 24 hours as an active process.

Figure legend Supplementary Figure 7

EV^{AML} can be taken up by CD34⁺ cells. Primary human AML CD34⁺ cells were treated with PKH67-labeled EVs or with green dye PKH67 (without EVs) for 4 hours (a) or 24 hours (b) at 4°C or 37°C. Representative flow cytometry histograms showing the PKH67-labeled EV^{AML} uptake by CD34⁺ cells in the FITC channel (green histograms) in comparison to CD34⁺ cells treated with PKH67 without EV^{AML} (blue histograms) or with unstained EV^{AML} (light blue histograms) or control cells (gray histograms). AML cell lines (e.g. MOLM-13) provided similar results (data not shown). Data are presented as the mean values of 4 independent experiments.

We also added positive controls for the Western blot and the relative Figure for contaminants that was as 'data not shown'. We used unprocessed serum from AML patients and compared it to plasma-derived EVs for the detection of albumin and apolipoprotein-A1. Our results show a reduction of both albumin and apolipoprotein-A1 in EVs in comparison to serum. Additionally, we have included a cell-specific marker that is expected to be absent in EVs, calnexin, for EV purity.

We added the following test in the manuscript, and we updated the Supplementary Figure 3 in Supplementary Fig.3,

Page 6; line 235:

Western blot analyses revealed a substantial decrease in contaminants such as albumin and apolipoprotein A1 (apo-A1) in our EV fractions in comparison with the serum counterparts. Calnexin was only detected in cell lysates (Supplementary Fig. 3b).

We updated our supplementary Figure 3b.

b) Western blot analyses for contaminants. The absence of cell-specific marker calnexin (90kDa) in EV from HD and AML patients in comparison to the positive control with cell lysate. Human serum albumin (HSA; 66 kDa) and apo-A1 (28kDa) reduction in EV^{HD} and EV^{AML} compared to positive control with AML serum (representative data for two subjects for each group from 4 independent experiments).

10) For better readability, the authors should consider dropping some of the acronyms, e.g. 'LB'.

We followed the suggestion, and we removed LB as an acronym.

11) Statements such as 'for the first time' are superfluous.

We removed the statements from the entire manuscript accordingly.

Page 14, line 636

Page 15, line 625

Page 15, line 657

Reviewer #4, expertise in glutathione metabolism and AML (Remarks to the Author):

In this study, Dorian Forte and colleagues sought to simultaneously characterize the energy-related metabolism, redox potential, and mitochondrial function in CD34+/CD38Low cells isolated from the peripheral blood of a cohort of AML patients. Furthermore, in order to provide a comprehensive profile of these patients, a significant part of the study was dedicated to the characterization of the nature, size, and content of extracellular vesicles obtained from the same patients through size exclusion chromatography. This was done in comparison to extracellular vesicles isolated from healthy donors, with the ultimate objective of establishing associations between the observed metabolic disturbances in CD34+/CD38Low AML blasts and molecular cues potentially conveyed by extracellular vesicle cargos.

Through this investigation, they established that: i) the less mature CD34+/CD38Low fraction of AML blasts exhibited a decrease in glycolytic capacity and an increased reliance on mitochondria, which was associated with higher levels of the antioxidant glutathione, ii) extracellular vesicles from AML patients were more abundant than those from healthy donors, iii) these vesicles displayed various protein and lipid markers that could assist in better stratifying AML patients, iv) the extracellular vesicles had the capability to support mitochondrial metabolism in CD34+ cells.

This study is well-conducted overall, technically sound, and relies heavily on the use of primary patient materials. While most of the findings are correlative in nature, the study holds the technical merit in its exploration of the influence of extracellular vesicles on disease onset, a topic that is not trivial to investigate and remains incompletely understood in the current literature. To try to establish a more causal relationship with their observations, the authors also isolated extracellular vesicles from patients for further investigation of their metabolic impact through co-culture with AML cell lines and primary cells.

Additional major points:

1- Although the authors consistently correlate cell phenotypes and extracellular vesicle biomarkers with well-established patient stratification methods like the ELN 2022 categories, there is an overall absence of dedicated analyses intended to link metabolic phenotypes and vesicle size distribution with patients' mutational status. It would be more informative for the reader to understand if specific mutations, combinations of mutations, or classes of mutations are more closely associated with particular metabolic phenotypes or the composition of extracellular vesicles. Could the authors address this critical point using, for instance, more-in-depth associative statistical analyses?

We agree with the Reviewer #4, and we think it would be more informative for the reader to provide results of nonparametric correlation tests.

However, despite our analysis, the p-value was not statistically significant when considering specific mutations in the metabolic parameters or EV phenotypes from our cohort. We can

not confirm whether this result is due to an actual lack of association or the high number of variables and categories. As a result, we have highlighted this in the Discussion section as a possible limitation of the study.

Page 15, lines 686-689

Although our findings need to be confirmed in future studies powered by adequate sample sizes and across AML subtypes with different molecular statuses, our study highlights a previously unexplored connection between leukemic redox metabolism and EV signatures. This link may have a potential prognostic relevance for AML.

2- Does the CD34+ cell fraction exhibiting low ROS levels and high mitochondrial potential have the capacity to resist apoptosis more effectively? Does this observation correspond to a reduced mitochondrial apoptotic priming, as measured by BH3 profiling?

According to Reviewer's suggestion, we checked whether ROS^{lo} MITO^{hi} CD34+ cells were more prone to resist apoptosis.

We have developed this method for redox metabolic evaluation by combining ROS, Mitotracker, and Thioltracker dyes. However, as detailed in the method section, our method requires an incubation with the intracellular probe/dye. We cannot use the stained samples for specific functional studies in vitro or in vivo. However, we tried to combine this staining (for ROS; MITO e GSH) by adding Annexin V. As you can see in the Figure only for Reviewer #4, we observed the percentages of Annexin V+ cells (apoptotic cells) in the gate for ROS^{lo} MITO^{hi} CD34+ with vehicle or EV from AML patients after 24 hours in cultures. We did not observe any significant difference in these fractions, even taking into consideration the AML risk, suggesting that maybe EV in our culture condition does not affect the survival rate in such conditions. Therefore, we are considering not including these data in the manuscript.

However, considering the Reviewer's suggestion, we performed an in vitro set of experiments using Venetoclax (BCL2-specific BH3 mimetic class), a drug proven to be highly effective in newly diagnosed AML patients.

To test the resistance to apoptosis, we pre-treated human AML cell lines MOLM-13 with vehicle (PBS) or EV from AML patients before adding Venetoclax. After 24 hours, we collected the cells and performed an apoptosis assay. As reported, pretreatment with EV^{AML}

increased the survival of MOLM-13 cells treated with Venetoclax for 24 hours ($P < 0.01$). However, we did not observe a difference when we stratified EV^{AML} according to risk (Fig. g).

We have included these data in the Results section of the main manuscript. Page 11, lines 447-453 and relative Figure 6g.

We therefore examined whether pretreatment with EV^{AML} might enhance resistance to Venetoclax, a selective inhibitor of BCL-2 that has advanced treatment options for AML patients⁵. We performed our experiments on MOLM-13, since OCI-AML3 have been reported as resistant to Venetoclax⁶.

Notably, pretreatment with EV^{AML} increased the survival of MOLM-13 cells treated with Venetoclax for 24 hours ($P < 0.01$). However, we did not observe a difference when we stratified EV^{AML} according to risk (Fig. 6g).

3- Is the increased extracellular vesicle size distribution observed in AML patients, compared to healthy donors, associated with hyperleukocytosis or high blast counts in patients? In other words, is there a quantitative relationship between leukemia burden and vesicle size?

We thank Reviewer #4 for pointing this out.

We have included a table with specific information on morphology examination (BM and PB blasts) for all the samples used for EV isolations so that we can perform the analysis requested.

We included the following table in the manuscript (Additional file 2 Table 2) for all the AML patients on whom we performed multiplex analysis of isolated EVs:

UPN	Morphology examination (%)		Age (Y)	Sex	diagnosis	ELN_2022	WBC (x 10 ⁹ /L)	EV profiling by MACSplex
	BM blasts	PB blasts						
ID_1	60	20	65	M	novo	adv	16,43	Y
ID_2	24	13	62	M	novo	adv	1,19	Y
ID_3	30	20	79	F	novo	adv	1,63	Y
ID_4	70	88	60	M	novo	adv	18,58	Y
ID_5	N/A	34	75	M	novo	adv	10,45	Y
ID_6	90	90	53	M	novo	adv	25,47	Y
ID_7	80	8,6	69	F	novo	adv	1,08	Y
ID_8	30	7	58	F	sec	adv	19,72	Y
ID_9	50	27	49	M	ter	adv	45,21	Y
ID_10	60	3	66	M	novo	adv	1,67	Y
ID_11	50	30	64	F	novo	adv	1,37	Y
ID_12	80	35	73	F	sec	adv	1,82	Y
ID_13	25	15	63	F	sec	adv	5,60	Y
ID_14	80	51	52	M	novo	int	28,80	Y
ID_15	80	12	79	F	novo	int	10,28	Y
ID_16	70	15	84	M	sec	int	2,07	Y
ID_17	100	66	69	M	novo	int	0,94	Y
ID_18	80	8	48	F	novo	int	2,53	Y
ID_19	80	67	72	M	sec	int	27,48	Y
ID_20	N/A	68	69	M	novo	int	2,90	Y
ID_21	20	2	52	M	novo	int	0,73	Y
ID_22	70	46	68	M	sec	int	20,00	Y
ID_23	100	95	35	M	novo	int	164,10	Y
ID_24	50	0	70	M	novo	int	1,79	Y
ID_25	40	0	38	F	novo	int	1,69	Y
ID_26	80	89	22	M	novo	int	11,40	Y
ID_27	N/A	N/A	54	M	novo	fav	7,20	Y
ID_28	80	20	82	F	novo	fav	5,30	Y
ID_29	100	66	68	F	novo	fav	115,57	Y
ID_30	30	4	62	M	novo	fav	1,00	Y
ID_31	90	87	55	M	novo	int	11,15	Y
ID_32	*4	2	54	F	novo	int	3,59	Y
ID_33	60	2	64	M	novo	int	2,87	Y
ID_34	90	30	70	M	novo	adv	10,09	Y
ID_35	40	10	64	M	novo	int	0,91	Y
ID_36	*15	N/A	77	M	novo	int	2,05	Y

ID_37	90	43	83	F	novo	fav	31,94	Y
ID_38	70	44	83	M	novo	adv	7,68	Y
ID_39	*8	0	46	M	novo	adv	2,04	Y
ID_40	10	1,5	76	M	sec	fav	111,05	Y
ID_41	50	18	18	F	novo	fav	N/A	Y

* hemodiluted bone marrow aspirates or myeloid sarcoma (MS) or with specific AML-defining recurrent genetic abnormality; Abbreviations: N/A not available; sec: secondary AML; ter: Therapy-related AML; novo: de novo AML; Y: yes; F: female or M: male; adv/int/fav: adverse, intermediate and favorable ELN risk.

The characterization of markers expressed on EVs can provide biological information about the cell or tissue of origin and their functional states. We performed a specific analysis comparing markers expressed on EVs using multiplex analysis (MACSplex with 37 markers) and BM blast counts to explore whether EV profiling can reflect the BM cellular composition and associate with blast counts.

We observed that EV^{AML} were highly enriched in the leukemic marker CD44. Importantly, we found that CD44 expression on PB EV^{AML} positively correlated with BM blasts detected by morphology examination. This finding might suggest the potential of CD44 as a reliable marker for monitoring disease status in AML patients. Moreover, using the expression of specific markers on EVs as a liquid biopsy to surrogate BM composition offers significant advantages in terms of non-invasiveness, accessibility, and potential for regular monitoring.

Supplementary Fig. 4a

a) Spearman correlation between CD44 expression on PB EV expressed as MFI and BM blast percentages detected by morphology examination (n=38, AML patients, R = 0.37, P = 0.02).

Page 8, lines 292-295

Interestingly, we found a positive correlation between the frequency of BM blasts and the expression of CD44 on circulating EV^{AML} (Supplementary Fig. 4) suggesting CD44 as a putative biomarker for assessing BM blast levels in circulation.

However, we did not find any associations between vesicle size distribution, high blast counts in patients, and leukemia burden.

We found a positive trend between WBC and the size of EV^{AML} at the diagnosis, without reaching a significant difference as reported in the Figure only for the Reviewer. We believe that this information should not be included in the main manuscript.

Spearman r

r	0,4120
95% confidence interval	-0,05132 to 0,7294
P value	
P (two-tailed)	0,0710
P value summary	ns

4- On the one hand, the authors demonstrated that specific extracellular vesicle markers are more prevalent in AML patients and are linked to the metabolic state of CD34+ cells. On the other hand, they showed that when isolated extracellular vesicles were co-cultured with AML CD34+ cells, they influenced their mitochondrial metabolism. However, it would provide valuable insights to determine whether extracellular vesicles with specific surface markers are more likely to induce these metabolic changes in CD34+ AML cell metabolism. For example, based on their vesicle profiling data, could the authors selectively isolate (by immunoprecipitation, for instance) extracellular vesicles that express distinct classes of molecules associated with specific metabolic phenotypes and conduct co-culture-based metabolic experiments with CD34+ cells using these marker-enriched immunoprecipitated vesicles?

We agree with the Reviewer #4, and it would be very interesting to sort specific EVs from our EV preparation. Unfortunately, we tried to perform this type of test using Mitenyi beads specific for CD44 PE but the amount of EVs was very low. This experiment is technically challenging due to their small size and low abundance in biological samples of EVs.

To address this challenge, we should employ strategies to enrich EV populations before sorting, such as differential ultracentrifugation, which will completely change our EV composition, requiring new characterization beyond the scope of the current work.

5- In line with the previous question, the results section does not clearly indicate the impact of extracellular vesicles isolated from AML patients on cord-blood-derived CD34+ cells. Although the authors reported that they do not influence their redox metabolism, it remains uncertain whether they affect other metabolic characteristics or the growth of healthy cord-blood-derived CD34+ cells. Does the observed metabolic reprogramming induced by extracellular vesicles from AML patients selectively affect cells with a (pre-)malignant phenotype?

According to Reviewer #4's suggestion, we investigated the HSPC differentiation by analysing the frequency distribution of CD34+ cells isolated from umbilical cord blood and co-cultured with vehicle (PBS) or EV^{AML} for 24 hours.

We measured the frequency of hematopoietic stem/progenitor cell types using flow-cytometry including hematopoietic stem cell (HSC; CD34+ CD38- CD90+ CD45RA- CD10-), common myeloid progenitor (CMP; CD34+ CD38+ CD123+ CD45RA- CD10-), multipotent progenitor (MPP; CD34+ CD38- CD45RA- CD90- CD10-), lymphoid-primed multipotent progenitor (MLP; CD34+ CD38- CD45RA+ CD90- CD10-), granulocyte-macrophage progenitor (GMP; CD34+ CD38+ CD123+ CD45RA+ CD10-), and megakaryocyte-erythroid progenitor (MEP; CD34+ CD38+ CD123- CD45RA- CD10-). These populations were distinguished using a gating strategy similar to Trino et al. (Front Oncol. 2022) and we included the relative gating strategy in Supplementary Figure.

After 24 hours of EV^{AML} treatment, we did not observe a significant reduction in the HSPC subsets in treated CD34+ cells.

Accordingly, we added the following sentence in the manuscript:

Page 11 lines 481-482

Firstly, we tested whether EV^{AML} can interfere with HSPC differentiation, and we did not find any significant difference in the HSPC subset proportion (Supplementary 7a-b).

Gating strategy 3. Gating strategy for identification of hematopoietic stem and progenitor cells on CB CD34⁺ based on Trino et al. 2022. Common myeloid progenitor (CMP; CD123⁺ CD45RA⁻), megakaryocyte-erythroid progenitor (MEP; CD123⁻ CD45RA⁻) and granulocyte-macrophage progenitor (GMP; CD123⁺ CD45RA⁺) were gated from CD34⁺ CD38⁺/CD10⁻ population. Hematopoietic stem cell (HSC; CD90⁺ CD45RA⁻) multipotent progenitor (MPP; CD45RA⁻ CD90⁻), lymphoid-primed multipotent progenitor (MLP; CD45RA⁺ CD90⁻) were gated from CD34⁺ CD38^{low/-} /CD10⁻ population.

b) Frequency of hematopoietic stem/progenitor cell types including hematopoietic stem cell (HSC), common myeloid progenitor (CMP), multipotent progenitor (MPP), lymphoid-primed multipotent progenitor (MLP), granulocyte-macrophage progenitor (GMP), and megakaryocyte-erythroid progenitor (MEP) after treatments with vehicle (PBS) or EV^{AML} detected by FACS analysis (n=8). No significant differences were reported using two-ANOVA.

We also included the method in Supplementary Method section:

Flow cytometric analysis of CB CD34⁺ after EV treatment for HSPC subsets

After treatment with EVAM for 24 hours, CB CD34⁺ cells were washed and incubated with CD90 FITC (clone: 5E10; BD Pharmingen™), CD45RA-BV510 (clone: HI100; Biolegend), CD34 Pe/Cy7 (clone: 4H11; eBioscience™), CD38 APC (clone: HIT2; Biolegend), CD10 APC-H7 (clone HI10a; BD™), CD123 PE (clone: 9F5; BD™) monoclonal antibodies, at room temperature for 15 minutes in the dark. After incubation, cells were washed and resuspended in staining buffer. Then, 10,000 events were acquired on CytoFLEX and analyzed using Kaluza software version 2.1 (Beckman Coulter, Milan, Italy).

Regarding the question on the possible EV role on a (pre-)malignant phenotype, we think that it is a really interesting question and a very important topic that needs to be addressed in a separate work.

We are currently working on MDS samples considered to be in “pre-leukemia” status and collecting data to understand the effect of EV in this context. However, we believe that this is outside the scope of the current manuscript.

6- Throughout the study, the authors frequently asserted that extracellular vesicles from AML patients might play a crucial role as a "mediator of metabolic reprogramming," associated with "a more aggressive and chemoresistant phenotype," without providing substantial evidence to support this claim. In addition to the changes in mitochondrial metabolism outlined by the authors, it would be valuable to see additional functional evidence that these extracellular vesicles convey in vitro a signal of enhanced proliferation and chemoresistance. For instance, are primary AML cells, healthy CD34+ cells, or AML cell lines more proliferative or more resistant to cytarabine and daunorubicin (or any other front-line therapies) when exposed to extracellular vesicles isolated from patients with a poor prognosis?

Considering the Reviewer #4's question on chemoresistance phenotype, we performed an in vitro experiment using Venetoclax (BCL2-specific BH3 mimetic class), a drug proven to be highly effective in newly diagnosed AML patients.

However, considering the Reviewer's suggestion, we performed an in vitro set of experiments using Venetoclax (BCL2-specific BH3 mimetic class), a drug proven to be highly effective in newly diagnosed AML patients.

To test the resistance to apoptosis, we pre-treated human AML cell lines MOLM-13 with vehicle (PBS) or EV from AML patients before adding Venetoclax. After 24 hours, we collected the cells and performed an apoptosis assay. As reported, pretreatment with EV^{AML} increased the survival of MOLM-13 cells treated with Venetoclax for 24 hours ($P < 0.01$). However, we did not observe a difference when we stratified EV^{AML} according to risk (Fig. 6g).

We have included these data in the Results section of the main manuscript. Page 11, lines 447-453 and relative Figure 6g.

We therefore examined whether pretreatment with EV^{AML} might enhance resistance to Venetoclax, a selective inhibitor of BCL-2 that has advanced treatment options for AML patients⁵. We performed our experiments on MOLM-13, since OCI-AML3 have been reported as resistant to Venetoclax⁶.

Notably, pretreatment with EV^{AML} increased the survival of MOLM-13 cells treated with Venetoclax for 24 hours ($P < 0.01$). However, we did not observe a difference when we stratified EV^{AML} according to risk (Fig. 6g).

In response to the Reviewer's question, to emphasize the impact of EV^{AML} from adverse-risk AML patients, we performed more in vitro and in vivo experiments. Our updated analysis involved additional samples using CD34⁺ AML cells or the human cell line MOLM-13. Furthermore, we performed new in vitro and in vivo experiments using the human cell line OCI-AML3. We added a new main Figure (6) and adjusted the results and overall manuscript accordingly (as reported below).

We introduced the work on cell lines (adding the results with OCI-AML3) in Result section:

Page 10, from lines 389 and updated the relative figure in Supplementary Figure 5a-i

Human leukemia cell lines show a different response to EV^{AML} in redox metabolism

Then, we simultaneously detected ROS and GSH levels along with mitochondrial potential as reported above for whole blood on human cell lines from AML patients: KG-1, Kasumi-1, MV-4-11, OCI-AML3 and MOLM-13 (Supplementary Fig. 5a-f).

When the MOLM-13, KG-1 and OCI-AML3 were cocultured with EV^{AML}, we noticed that MOLM-13 only showed a lower percentage of ROS^{hi} MITO^{hi} in the presence of EV^{AML} ($P < 0.01$; Supplementary 5a), whereas OCI-AML3 showed an increase in the fraction ROS^{lo} MITO^{hi} ($P < 0.05$; Supplementary Fig. 5g). Moreover, both MOLM-13 and OCI-AML3 showed a significant increase in ROS^{lo}GSH^{hi}

fraction in the presence of EV^{AML} ($P < 0.05$, respectively; Supplementary Fig. 5c-h). In addition, MOLM-13 and KG-1 shared a considerable reduction in ROS^{hi} GSH^{lo} ($P < 0.01$ for MOLM-13 and $P < 0.05$ for KG-1) and a concomitant decrease in the fraction GSH^{lo}MITO^{hi} ($P < 0.05$, respectively). Only, KG-1 cell line presented an increase in the fraction with high GSH content and high mitochondrial potential ($P < 0.05$: Supplementary Fig. 5f).

Conversely, EV^{AML} showed a limited effect on redox metabolic markers in the MV-4-11 and Kasumi-1 cell lines (Supplementary Fig. 6a-f). Only the fraction of ROS^{hi} MITO^{hi} cells decreased in response to EV^{AML} in MV-4-11 cells ($P < 0.05$; Supplementary Fig. 6a). In summary, EV^{AML} showed a mild effect on human leukemia cell lines, mainly decreasing the ROS^{hi} GSH^{lo} fractions in the KG-1 and MOLM-13 cell lines. This suggests that EV^{AML} might have a more evident role in specific LSC metabolism.

Supplementary Figure 5

Metabolic redox modulation by EVs of human leukemic cell lines from human leukemic cell lines including MOLM-13 (a, c, e), KG-1 (b, d, f) and OCI-AML3 (g, h, i).

*a, b, g) ROS/MITO sub-fractions in leukemic cells treated for 24 hours with vehicle (PBS) or EV^{AML}: ROS^{hi} MITO^{lo}, ROS^{hi} MITO^{hi}, ROS^{lo} MITO^{hi}; c, d, h) ROS/GSH sub-fractions in leukemic cells treated for 24 hours with vehicle (PBS) or EV^{AML}: ROS^{hi} GSH^{lo}, ROS^{hi} GSH^{hi}, ROS^{lo} GSH^{hi}. e, f, i) GSH/MITO sub-fractions in leukemic cells treated for 24 hours with vehicle (PBS) or EV^{AML}: GSH^{hi} Mito^{lo}, GSH^{hi} Mito^{hi}, GSH^{lo} Mito^{hi}. Significant differences were reported using Mann–Whitney test for unpaired samples. *p* values <0.05 (*), <0.01 (**) were considered significant.*

Then, to emphasize the impact of EV^{AML} in adverse or intermediate-risk patients, we introduced the new Figure 6 (using cell lines), in which we performed experiments in vitro and in vivo using both OCI-AML3 and MOLM-13 and discriminated the effects between EV from different ELN risks. We firstly investigated the metabolic parameters of our interest and, secondly, the resistance to Venetoclax treatments mediated by EV^{AML}, in vitro.

Figure 6

EVAML from different AML risks have distinct metabolic effects on OCI-AML3 and MOLM-13, and adverse-risk EV^{AML} enhance the engraftment of MOLM-13 in vivo.

We then investigated how circulating EV^{AML} from different ELN risk groups affects the metabolism of human cell lines. Our focus was on the two cell lines, OCI-AML3 and MOLM-13, derived from PB of AML patients and which were more responsive to EV^{AML}.

For OCI-AML3, all three types of EV^{AML} were found to significantly increase the proportion of cells with low ROS levels and high GSH content ($P < 0.0001$, respectively). Additionally, only adverse EV^{AML} increased the proportion of ROS^{lo} and high mitochondria (MITO^{hi}) ($P < 0.05$), while significantly decreasing the proportion of ROS^{hi} and GSH^{hi} ($P < 0.01$) (Fig. 6a-c).

For MOLM-13, all three types of EVs were observed to reduce the frequency of cells with high ROS and low GSH (ROS^{hi} GSH^{lo}) ($P < 0.0001$ for fav EV^{AML} and $P < 0.01$ for both intermediate and adverse EVs). However, only adverse EV^{AML} significantly decreased the percentage of cells with low GSH and high mitochondria potential (GSH^{lo} MITO^{hi}) ($P < 0.05$) (Fig. 6d-f).

We therefore examined whether pretreatment with EV^{AML} might enhance resistance to Venetoclax, a selective inhibitor of BCL-2 that has advanced treatment options for AML patients⁵. We performed our experiments on MOLM-13, since OCI-AML3 have been reported as resistant to Venetoclax⁶.

Notably, pretreatment with EV^{AML} increased the survival of MOLM-13 cells treated with Venetoclax for 24 hours ($P < 0.01$). However, we did not observe a difference when we stratified EV^{AML} according to risk (Fig. 6g).

Finally, we performed *in vivo* experiments. Briefly, we treated luciferase-transduced MOLM-13 or OCI-AML3 cell lines with EVs from adverse—or intermediate—or favorable-risk AML patients and intravenously injected them into NOD-scid IL2R gamma null (NSG) mice (n=40, two independent experiments). The engraftment efficiency of human leukemia cell lines was assessed weekly using bioluminescence imaging.

In the figure only for the Reviewer, it is evident that the experiments conducted with OCI-AML3 showed that pre-treatments with EV^{AML} from adverse- or intermediate-risk patients significantly enhanced the engraftment of luc-mCherry OCI-AML3 compared to control cells (vehicle). However, due to signal dispersion from deep tissue or within the tumor, the exact source of the signal could not be identified and quantified.

Whole-animal bioluminescence imaging from day 18. Ventral and dorsal view images from NSG mice for each group are shown. Mice transplanted with luc-mCherry OCI-AML3 cells pre-treated for 24 hours with vehicle control (PBS, n=5) (a-b), high-risk EV^{AML} (n=5) (c-d) or intermediate-risk EV^{AML} (n=5) (e-f).

Consequently, we deemed it inappropriate to present this data without a proper quantification to the Readers.

Therefore, we propose to display the data only from the second experiment, which involved NSG mice transplanted with luc-mCherry MOLM-13, pre-treated with EVs from favorable/intermediate/adverse risk AML patients, and untreated (vehicle). This information will be presented clearly and concisely in the manuscript, and we believe it will accurately communicate our findings.

We added the relative procedure and experiments in the Figure 6 (h-i) and in the Method sections, as outlined below:

Finally, we conducted an *in vivo* study to evaluate the effect of EV^{AML} pre-treatment on MOLM-13 or OCI-AML3 cells xenografted in NSG mice. We pre-treated Luc-mCherry MOLM-13 and OCI-AML3 cells with EV^{AML} or with vehicle control (PBS) prior to tail vein infusion and engraftment. Our results showed that pre-treatment with EV^{AML} from adverse-risk AML patients was more effective in increasing AML xenograft growth compared to treatment with a vehicle control (PBS). Interestingly, after 18 days, the treatment with EV^{AML} from adverse risk significantly improved the engraftment of Luc-mCherry MOLM-13 cells compared to untreated cells ($P < 0.001$, Fig. 6h-i). However, EV^{AML} from intermediate and favorable patients did not affect the engraftment of Luc-mCherry MOLM-13 cells (Fig.6h-i). A similar pattern, although characterized by dispersion of the signal, was observed when OCI-AML3 cell line was used (data not shown).

According to *in vitro* experiments, our *in vivo* findings indicate that EV^{AML} from adverse-risk AML patients may have an increased capacity for accelerating *in vivo* leukemia cell engraftment compared to EV^{AML} collected from intermediate-risk or favorable AML patients.

Figure 6

Figure 6

Figure 6.

Redox metabolic profiling for OCI-AML3 and MOLM-13 treated for 24 hours with EV^{AML} from adverse, intermediate or favorable-risk AML patients. ROS/MITO subsets expressed as percentage were reported for OCI-AML3 (a) or MOLM-13 (b). Percentage of ROS/GSH subsets for OCI-AML3 (b) and MOLM-13 (e). Percentages of GSH/MITO subsets for OCI-AML3 (a) and MOLM-13 (f) treated with favorable EV^{AML} (n=9), intermediate EV^{AML} (n=8) or adverse EV^{AML} (n=8) from 8 independent experiments. Significant differences were reported by two-way ANOVA with Šidák's multiple comparisons test. g) MOLM-13 pre-treated with EV^{AML} for 4 hours before adding Venetoclax (150 nM) for 24 hours. MOLM-13 treated with vehicle or favorable EV^{AML} (n=3) or intermediate EV^{AML} (n=4) or adverse EV^{AML} (n=5) from 5 independent experiments. h-i) Whole-animal bioluminescence imaging from day 18. Ventral and dorsal view images from mice NGS for each group are shown. Each animal's region of interest (ROI) was defined at every time point (inset). Mice transplanted with luc-mCherry MOLM-13 cells pre-treated for 24 hours with vehicle control (PBS, n=5), adverse-risk EV^{AML} (n=5) or intermediate-risk EV^{AML} (n=5). The graph shows the quantification of bioluminescence as the average total photon flux per second from days 6 to 18 after initiation. Error bars represent \pm SEM.

We also repeated experiments on AML CD34⁺ isolated from patients at diagnosis, and we included the data where we tested the effects of EV^{AML} from different ELN risk categories.

We updated the old Figure 6 which is now Figure 7 and relative Supplementary Figure 8 a, b, c. We removed the Volcano plot from Nanostring experiments in previous Fig. 6 and we transfer in Supplementary Fig. 8d.

Figure 7

e) Percentages of ROS^{hi}MITO^{hi} and ROS^{lo}MITO^{hi} for AML CD34⁺ cells treated with favorable EV^{AML} (n=9), intermediate EV^{AML} (n=7) or adverse EV^{AML} (n=8) from 10 independent experiments. Significant differences were reported by two-way ANOVA with Šidák's multiple comparisons test. P values <0.01 (**), <0.001 (***) , <0.0001 (****) were considered significant.

Supplementary Figure 8a,b,c:

Redox metabolic profiling for AML CD34⁺ cells treated for 24 hours with EV^{AML} from adverse, intermediate or favourable-risk AML patients. ROS/MITO subsets expressed as percentages were reported for AML CD34⁺ cells (a). Percentage of ROS/GSH subsets for

AML CD34⁺ cells (b). Percentages of GSH/MITO subsets (c) for AML CD34⁺ cells treated with favorable EV^{AML} (n=9), intermediate EV^{AML} (n= 7) or adverse EV^{AML} (n=8) from 10 independent experiments. Significant differences were reported by two-way ANOVA with Šidák's multiple comparisons test.

Result section on Figure 7:

EVs from AML patients alter the redox metabolism of CD34⁺ AML cells, modulating the GSH/GPX4 axis

.....

When we cocultured AML CD34⁺ with EV^{AML}, combining ROS with mitochondrial functionality, we did not find any significant differences (Fig. 7a). However, we found a significant decrease in ROS levels and an increase in MITO levels after EV^{AML} treatment, as reported by MFI values ($P < 0.05$; Fig. 7d). As a result, within the leukemic CD34⁺ progenitor compartment, coculture with EV^{AML} significantly reduced the proportion of ROS^{hi} GSH^{lo} cells (mean percentage: 53,91%) compared to that in AML CD34⁺ cells treated with vehicle (mean percentage: 76,21%) ($P < 0.01$; Fig. 6b), as previously reported in human cell lines (KG-1 and MOLM-13, Supplementary Fig. 5c-d). Moreover, we observed an increase in the frequency of ROS^{hi} GSH^{hi} fraction in the presence of EV^{AML} ($P < 0.01$; Fig. 7b). Again, similar to what we observed for the KG-1 cell line (supplementary Fig. 5f), we also confirmed in AML CD34⁺ cells an increase in GSH^{hi} MITO^{hi} fractions ($P < 0.002$) and a reduction in GSH^{lo} MITO^{hi} fractions ($P < 0.01$; Fig. 7c).

We then analyzed the metabolic effects by sorting the data based on EV^{AML} from different ELN risk groups, we observed that all three subtypes of EV^{AML} significantly increased the frequency of CD34⁺ cells with high levels of GSH and mitochondria (MITO), and significantly decreased the frequency of CD34⁺ cells with high levels of ROS and low levels of GSH, as well as the proportion of cells with GSH^{lo} and MITO^{hi} (Supplementary Fig. 8a-c). However, only intermediate EV^{AML} were able to increase the proportion of CD34⁺ cells with low ROS and high MITO, whereas only favorable EV^{AML} increased the proportion of CD34⁺ cells with high ROS and high GSH (Supplementary Fig. 8a-b).

Significantly, unlike EVs from intermediate ($P < 0.001$) and adverse-risk patients ($P < 0.01$), EV^{AML} from favorable AML patients increased the proportion of CD34⁺ cells with high levels of ROS and high mitochondrial potential (Fig. 7e) as compared to untreated cells ($P < 0.01$; Supplementary Fig. 8a) and decreased the proportion of CD34⁺ cells with low levels of ROS and high mitochondrial potential as compared to intermediate ($P < 0.01$) and adverse-risk EV^{AML} ($P < 0.05$; Fig. 6c). These data suggest a metabolic vulnerability driven by EVs from different ELN risk categories.

Collectively, these results demonstrate that circulating EVs from AML patients may trigger AML CD34⁺ stem cells toward an increase in both mitochondrial potential and GSH levels, reducing ROS levels and showing a leukemia-dependent mechanism partially reverted by GPX4 inhibition. Understanding the metabolic dependencies mediated by EVs from different ELN risk groups can allow for the development of tailored therapeutic strategies aimed at exploiting new metabolic vulnerabilities.

We included the relative materials and methods in the Supplementary method section

Human leukemia cell line transduction

To label the cells for *in vivo* transplantation, the OCI-AML3 and MOLM-13 cell lines were stably transduced with a lentiviral vector expressing stably luciferase and mCherry, MI-Luciferase-IRES-mCherry (w168-1; Addgene plasmid, #75020), carrying mCherry as a selective marker. To infect human AML cell lines, 293T were cotransfected with 10 µg of lentiviral vector, 3 µg PMD2G envelope plasmid, 2.5 µg of REV packaging plasmid, and 5 µg of RRE transfer plasmid using calcium phosphate transfection system. After 16 hours, the media was removed and replaced with 5 mL of fresh medium. Viral supernatant was then collected at 24 and 48 hours and added directly to human AML cell lines plated at 1×10^5 supplemented with 8 µg/mL of polybrene (Merck). Spin-infection was performed for 1.30 hours at 37°C twice a day for one separate day. After 48 hours from the first cycle of infection, infected cells were sorted using FACS Fusion II (BD Bioscience).

In vivo experiments

Experiments involving animals were approved by the Italian Ministry of Health and have been done in accordance with the applicable Italian laws (D.L.vo 26/14 and following amendments), the Institutional Animal Care and Use Committee, and the institutional guidelines at the European Institute of Oncology.

To determine the *in vivo* effects of human cell lines treated with EV^{AML} on leukemia progression and engraftment, tandem tagged luciferase-mCherry human cell lines (OCI-AML3 and MOLM-13) were seeded in a 24-well plate (20×10^4) and treated with EV^{AML} (15 µg) from AML patients stratified according to ELN2022 risk ($n=5$ for each risk group, respectively) or PBS/vehicle as control. After 24 hours, human cell lines were washed and counted before transplantation. Luc-mCherry AML cells (1×10^5) treated with EV^{AML} or vehicle control (PBS) were injected with a 26-gauge needle into the lateral tail vein of 6–8-week-old male NOD-scid IL2R gamma null (NSG) mice ($n=5$ /group; two independent experiments). Mice were randomly assigned to different condition cohorts. All the mice were monitored twice a week and imaged using bioluminescence imaging to document engraftment. Animals were anesthetized for imaging *in vivo* using 2% Isoflurine (flow rate 1 L/min O_2). Spectral imaging was commenced 10 min post IP injection to allow stabilization of light output. Mice were injected with 150 mg/kg of D-Luciferin and imaged twice a week to monitor disease status and treatment efficacy. As a surrogate for tumor burden, bioluminescence was quantified using Living Image Software (version 4.7.2, PerkinElmer).

References for Rebuttal file

1. Zeijlemaker, W. *et al.* CD34(+)CD38(-) leukemic stem cell frequency to predict outcome in acute myeloid leukemia.
2. Forte, D. *et al.* Bone Marrow Mesenchymal Stem Cells Support Acute Myeloid Leukemia Bioenergetics and Enhance Antioxidant Defense and Escape from Chemotherapy. *Cell Metab* **32**, 829-843.e829 (2020).
3. Su, Y. *et al.* The Imipridone ONC213 Targets alpha-Ketoglutarate Dehydrogenase to Induce Mitochondrial Stress and Suppress Oxidative Phosphorylation in Acute Myeloid Leukemia. *Cancer Res* **84**, 1084-1100 (2024).
4. Gullberg, J., Jonsson, P., Nordstrom, A., Sjostrom, M. & Moritz, T. Design of experiments: an efficient strategy to identify factors influencing extraction and derivatization of *Arabidopsis thaliana* samples in metabolomic studies with gas chromatography/mass spectrometry. *Analytical Biochemistry* **331**, 283 (2004).
5. Konopleva, M. *et al.* Efficacy and Biological Correlates of Response in a Phase II Study of Venetoclax Monotherapy in Patients with Acute Myelogenous Leukemia. *Cancer Discov* **6**, 1106-1117 (2016).
6. Lima, K. *et al.* Obatoclax reduces cell viability of acute myeloid leukemia cell lines independently of their sensitivity to venetoclax. *Hematol Transfus Cell Ther* **44**, 124-127 (2022).
7. Jayavelu, A.K. *et al.* The proteogenomic subtypes of acute myeloid leukemia. *Cancer Cell* **40**, 301-317 e312 (2022).

Dear Reviewers,

Please find below a second point-by-point detailed rebuttal to your comments.

Reviewer #1 (Remarks to the Author):

In general, the authors have improved the manuscript in terms of EV characterization and clarified their point on using a broader non-AML specific EV population, due to the aim of establishing a new diagnostic/prognostic liquid biopsy tool.

Further, the authors have responded to all my major concerns.

1. The WB EV characterization has been improved significantly. WB analyses included for calnexin (cell marker), Apo-A1, and HSA (non-EV marker). I would suggest including the cell lysate and EV samples on the same gel/WB. Also, given the four independent experiments, a densitometric analysis could be shown.

We appreciate Reviewer #1 for acknowledging the improvements of the manuscript and the efforts put into the manuscript revision. We repeated the WB to include cell lysate and EV samples on the same gels. Accordingly, we have now reported the densitometric analyses for the tested markers. However, we aim to demonstrate the presence or absence of specific proteins rather than their quantification. Several factors in the extracellular vesicle (EV) field, such as preparation methods, heterogeneity, and the lack of a proper loading control as seen in cells with actin, need to be considered. Therefore, we intentionally did not include statistical analysis on the densitometric data to avoid conveying a misleading message that could lead to a comparison between samples for the expression of the proteins detected.

SUPPLEMENTARY FIGURE 3b

b) Western blot analyses for contaminants. Human serum albumin (HSA; 66 kDa) and apo-A1 (28kDa) reduction in EV^{HD} and EV^{AML} compared to positive control with unprocessed AML serum. The absence of cell-specific marker calnexin (90kDa) in EV from HD and AML patients in comparison to the positive control with cell lysate. Densitometric analysis of western blot results presented was performed using image processing software. Data are presented as mean values \pm SEM of three to four independent experiments.

2. The authors added important information about PB and BM blast percentage and correlations of EV-marker expression between both sources which might be used for disease monitoring. No further concerns.

3-5. The authors fully addressed my comments.

Reviewer #2 (Remarks to the Author):

The authors have performed extensive work to address the reviewers' critiques, which is highly appreciated. However, the main concern remains.

We thank Reviewer #2 for acknowledging the extensive work we put into the review process for our manuscript.

The study indicates that EVs isolated from favorable, intermediate, and adverse AMLs induce different and inconsistent metabolic changes. For example, while “all three types of EVs were observed to reduce the frequency of cells with high ROS and low GSH (ROShiGSHlo)”, “only adverse EVAML increased the proportion of ROSlo and high mitochondria (MITOhi) ($p < 0.05$), while significantly decreasing the proportion of ROShi and GSHhi ($p < 0.01$) (Fig. 6a-c)”. These inconsistencies raise questions about the significance of these changes. I still believe it is essential to provide functional data demonstrating the importance of separating CD34+ blasts based on the expression of ROS/GSH and ROS/MITO is essential. Some changes seem meaningless without this context. Craig Jordan's group has demonstrated the feasibility of utilizing Redox-stained AML cells for functional studies, and combining this with GSH or MITO should be feasible.

We recognize Reviewer #2's concern that our approach might have limitations in formally demonstrating the correlation between the metabolic profile and the functional state of leukemic cells. However, increasing evidence suggests cellular specialization is closely linked to underlying metabolic activity (*Ghosh-Choudhary S et al., Trends Cell Biol. 2020*). This relationship has also been observed in leukemic cell subsets in AML (*Patel SB et al., Front Oncol. 2022; Mishra SK et al., Blood. 2023*). In this context, although our experiments do not provide a detailed functional characterization of these cellular subsets, the use of redox metabolic states of well-established leukemic cell subsets—already known for their different functional roles within the hematopoietic hierarchy—strongly supports a link between their metabolic profiles and functional status.

Said that we agree with the Reviewer that a further set of experiments aiming at demonstrating the functional state of CD34+ cells according to the different redox metabolic profiles might strengthen the reliability and consistency of our data. We then attempted to perform the experiments suggested by Reviewer #2 to provide functional data to better explain our results, despite the technical limitations that we have resolved.

Using our established protocol, we used frozen CD34+ AML cells and stained them for ROS, GSH, and MITO. Following staining, we sorted the viable CD34+ cell populations according to the gating strategy described below, to functionally validate the selected subsets using a clonogenic assay. Our results, even using frozen AML samples, confirmed a more aggressive phenotype for the subset with low ROS levels combined with high GSH content or high mitochondrial functionality. Notably, the highest clonogenic output was observed in the same fraction that showed a significant increase in AML patients with intermediate or adverse risk compared to the favorable group, as shown in Figure 1i-l of the main manuscript. These findings suggest that these subsets (ROS_{low} MITO_{high} and ROS_{low} GSH_{high}) may play an important role in leukemogenesis and disease aggressiveness, highlighting the relevance of redox metabolism, even from a clinical perspective. We have now included the relevant data and results in the main manuscript.

RESULTS (page 5, lines 158-170)

To investigate the functional capabilities of these redox metabolic subsets, we repeated the experiments using isolated CD34⁺ AML cells. The cells were labeled for ROS, MITO, and GSH and then sorted accordingly. Specifically, we focused on the ROS-low and ROS-high subsets in relation to their GSH content or mitochondrial functionality. We also compared the fractions of cells with low and high GSH within the CD34⁺ AML cells that exhibited high mitochondrial potential. We then evaluated the survival and proliferative capacity of each subset using a colony-forming cell assay. Intriguingly, the ROS^{lo} population revealed a higher and significant clonogenic potential in both scenarios with high GSH content ($p < 0.05$) and high mitochondrial potential ($p < 0.05$) (Supplementary: Fig. 1m-n). While a high level of GSH and high mitochondrial potential led to an increase in clonogenic output, the change was not statistically significant when compared to the fraction with low GSH and high mitochondrial potential (GSH^{lo} MITO^{hi}) (Supplementary: Fig. 1o). Here, we observed that the leukemic hematopoietic compartment in AML patients shows a transition to lower ROS levels and higher mitochondrial potential and GSH levels, suggesting a more aggressive disease. Our data also suggest a novel combination of metabolic markers that may help risk stratification and clinical prediction of AML patients.

Figure legend for Supplementary Figure 1m,n,o

m-o) On the top, representative dot plots illustrating the gating strategy used for cell sorting. On the bottom: m) CFU counts in CD34⁺ AML cells sorted based on high GSH content, comparing ROS low (purple column) to ROS high (cyan column) fractions. n) CFU counts in CD34⁺ AML cells with high mitochondrial functionality, comparing ROS low (cerulean column) to ROS high (green column) fractions. o) CFC counts in CD34⁺ AML cells sorted for high mitochondrial functionality, comparing GSH low (pink column) to GSH high (yellow column) fractions ($n = 4$ independent experiments). Significant differences were reported using the Mann–Whitney test for unpaired samples with () $p < 0.05$ considered significant. Data are presented as mean values \pm SEM.*

In the Method section, we have also included the relevant methods.

Page 17, lines 755-767

Flow cytometry analysis and fluorescence-activated cell sorting (FACS) after redox metabolic staining

[....]

In a set of experiments, PB CD34⁺ AML cells that had been previously purified were thawed and prepared for FACS sorting. The CD34⁺ AML cells were stained with DAPI (Sigma–Aldrich, Milan, Italy) to exclude dead cells and were stained with a two-by-two combination of ROS, MITO, and GSH dyes. Sorting was performed using a 100 µm nozzle with a pressure of 20 PSI. A BD FACSAria™ Fusion Special Order (SORP) cell sorter cytometer (BD Biosciences) equipped with three lasers was used (405 nm, 488 nm and 640 nm). Sorted fractions are shown in Supplementary Figure 1m-o. After completing the sorter, cells were centrifuged, resuspended in RPMI medium, counted, and plated in clonogenic assays.

Clonogenic assays

CD34⁺ AML cells (1000 cells/plate) were cultured at 37°C and 5% CO₂ in 35-mm dishes in methylcellulose-based medium (human StemMACS HSC-CFU lite w/ Epo, Miltenyi Biotech). After 10 days of incubation, colony-forming unit (CFU-C) growth was evaluated by standard morphologic criteria using an inverted microscope (Axiovert 40, Zeiss).

Furthermore, the majority of experiments were performed on CD34⁺ AML blasts. Referring to these cells as AML stem cells or LSC-like is misleading, as they are merely bulk blasts. Using the term AML cells is more appropriate for the experiments described in the manuscript. In supplemental Figure 5, the authors investigated human cell lines reacting to EV from AMLs, and concluded, “This suggests that EVAML might have a more evident role in specific LSC metabolism” (line 423-424). This conclusion is inaccurate, as the experiments were conducted on bulk cell line cells.

We have reviewed the suggestions from Reviewer #2 and removed the term "LSC-like" from the entire manuscript and the Abstract. Additionally, as recommended, we have also eliminated the word "stem" from the abstract and the entire manuscript.

Page 3, lines 46, 53

Page 4, line 128

Page 5, lines 175, 186

Page 9, line 375

Page 11, line 441

Page 12, lines 499, 509, 529

Page 15, line 646

We appreciate Reviewer #2 for bringing this to our attention. Following the reviewer's suggestion, we have replaced the term 'LSC' with 'AML cell subsets' and corrected the sentence.

Page 10, lines 402-404

This suggests that EV^{AML} may play a more significant role in the metabolism of specific AML cell subsets. This suggests that EV^{AML} might have a more evident role in specific LSC metabolism.

In Figure 2d, the authors observed a significant increase in mitochondrial dependence and a reduction in glycolytic capacity in the more immature CD34+CD38low/- fraction. However, in Supplemental Figure 2e, no significant difference was observed, and the authors claimed, "our analysis revealed that more specific leukemic subsets, such as progenitors, myeloid cells, and primitive LSCs, did not exhibit significant differences in the metabolic parameters evaluated (Supplementary Fig. 2c-f). This observation reinforces the validity of using bulk staining for CD34+ and CD38low/- markers to identify leukemic subsets in the circulation of AML patients for metabolic studies". This discrepancy is confusing and further supports my above point – using the term AML blasts.

The study indicates that EVs isolated from favorable, intermediate, and adverse AMLs induce different and inconsistent metabolic changes. For example, while "all three types of EVs were observed to reduce the frequency of cells with high ROS and low GSH (ROShiGSHlo)", "only adverse EVAML increased the proportion of ROSlo and high mitochondria (MITOhi) (p<0.05), while significantly decreasing the proportion of ROShi and GSHhi (p< 0.01) (Fig. 6a-c)". These inconsistencies raise questions about the significance of these changes.

We thank the Reviewer #2 for highlighting this specific point. It is important to consider the impact of intra- and inter-patient heterogeneity in a disease like AML, as AML cell lines and primary samples from patients may exhibit different behaviours. Cell lines may differ from the in vivo setting in important aspects, especially considering the metabolic response. To address this, we have performed several independent experiments to limit variabilities due to in vitro cell cultures (timing, passages...) and the materials isolated (EV profiles and cargo) from AML patients that arise from genetic and non-genetic sources.

Our research focuses on metabolic studies, which involve experiments on fresh primary AML specimens and cell lines (as different recipient cells) to provide a broad spectrum of EVs' metabolic effects. We have made every effort to address all questions raised and have added additional data, discriminating EVs from different risk groups as requested by other Reviewers. This has added more complexity but provided a broader perspective on our results, reflecting the complexity of the AML scenario. Therefore, some apparent inconsistencies and complications should be expected, which has naturally generated further inquiries as reported by Reviewer #2.

We agree with Reviewer #2 that this point needs to be clarified in the discussion section, as requested. To address this transparently, we propose adding a paragraph in the discussion section that outlines potential explanations for these complexities that we have not previously included.

DISCUSSION

Page 14, lines 600-604

It has also been reported that EVs can induce phenotypic changes in recipient cells and modify cellular metabolism (45). Specifically, we observed that EVs isolated from AML patients with favorable, intermediate, and adverse prognoses induce metabolic changes that depend on the recipient cells, whether primary AML CD34+ cells or cell lines. This suggests that the complexity of the leukemia scenario is influenced not only by the origin of the EVs but also by the intrinsic characteristics of the recipient cells. However, we demonstrated that EVAML might be a key mediator of metabolic reprogramming, enhancing the mitochondrial dependence of circulating CD34+ cells and suggesting a more aggressive and chemoresistant phenotype. Indeed, AML cells pre-treated with EVAML displayed increased resistance to venetoclax treatments. This indicates that EVAML may directly affect the response to drugs. Additionally, we found that EVAML from high-risk AML patients enhanced the engraftment of AML cell lines such as MOLM-13, supporting our interests in EV cargo and the EV roles in intracellular metabolism. Thus, the understanding of the complex metabolic cross-talk between leukemic cells and EVs is certainly a key point for a comprehensive characterization of the AML metabolic landscape.

We followed the suggestion of Reviewer #2 and removed the misleading terms as mentioned. Furthermore, we revised the manuscript and removed any misleading information.

Page 6, lines 201-203

Our analysis revealed that more specific leukemic subsets, such as progenitors, myeloid cells, and primitive LSCs, did not exhibit significant differences in the metabolic parameters evaluated (Supplementary Fig. 2c-f). ~~This observation reinforces the validity of using bulk staining for CD34+ and CD38low/- markers to identify leukemic subsets in the circulation of AML patients for metabolic studies~~

Reviewer #3 (Remarks to the Author):

The authors have thoroughly revised their manuscript. In particular, the sample size is now described more transparently, and the authors have toned down their claims regarding the prognostic value.

Two concerns remain:

1) The authors should more explicitly describe their acceptance criteria for lipid identifications, as some species with odd-numbered fatty acyl chains might be unlikely in the samples (e.g. one third of the reported DG species), see also <https://doi.org/10.1038/s41580-024-00758-4>

In terms of the acceptance criteria, we used the following two factors for our assessments:

a) The variance between theoretical MS1 and experimental MS1 should be < 25 ppm. In our dataset, the average variance is 4.30 ppm.

b) The total score determined by the MS-DIAL annotation software should be > 80%. In our dataset, the average score is 96.7%.

The presence of molecular species with odd-numbered chains should cause concern if their abundance is like that of species with even-numbered chains. In our case, the odd/even ratio is 3.28% for AML and 3.94% for HD, which is typical for a mammalian biological sample [Jenkins, B., Seyssel, K., Chiu, S. et al. *Odd Chain Fatty Acids; New Insights of the Relationship Between the Gut Microbiota, Dietary Intake, Biosynthesis and Glucose Intolerance. Sci Rep 7, 44845 (2017). <https://doi.org/10.1038/srep44845>].*

We understand the difficulties in identifying lipids using spectrometric data. To address this, we followed the guidelines provided by the Lipid Standard Initiative (refer to line 865 of our manuscript). The main contributors to this initiative are the authors of the article "*Pitfalls in lipid mass spectrometry of mammalian samples – a brief guide for biologists*" that you referred us to.

According to Reviewer #3, we added the following information to the Result and Method section:

RESULTS (page 8, lines 321-324)

In our dataset, based on the acceptance criteria outlined in the methods section, the variance between theoretical and experimental MS1 was 4.30 ppm. The average score assigned by the MS-DIAL annotation software was 96.7%. Additionally, we observed an odd/even ratio of 3.28% for EV^{AML} and 3.94% for EV^{HD}.

METHODS (page 19, lines 849-852; 856-862)

Lipid extraction and liquid chromatography quadrupole time-of-flight mass spectrometry (LC/MS QTOF) analysis

[.....]

The acceptance criteria for identifying lipids were as follows: 1) The variance between theoretical MS1 and experimental MS1 must be less than 25 ppm. 2) The total score assigned by the MS-DIAL annotation software must exceed 80%.

[.....]

For the network analysis, the diagrams were generated using the Graph Editor tool (https://csacademy.com/app/graph_editor/, accessed on 10 January 2023) and included a statistical reinterpretation of data from LipidOne. Network transformations were analyzed by assessing the ratios of product to reactant masses for each sample. To determine which biochemical reactions were enhanced or inhibited, we compared the mean mass ratios between EV^{AML} and EV^{HD}, following the methodology outlined by A. Nguyen et al. (65).

2) I would still appreciate more detail on the network analysis presented in Fig. 5b.

We thank Reviewer #3 for pointing out this lack of clarity. The network diagram depicts potential transformations between lipid classes in the EV^{AML} group relative to the EV^{HD} group. These transformations can be explained by referring to the established pathways of lipid classes, as reported in previous studies:

Vance, J.E. Phospholipid Synthesis and Transport in Mammalian Cells. Traffic 2015, 16, 1–18.

Lewis, A.C.; Wallington-Beddoe, C.T.; Powell, J.A.; Pitson, S.M. Targeting Sphingolipid Metabolism as an Approach for Combination Therapies in Haematological Malignancies. Cell. Death Discov. 2018, 4, 72.

Kim, O.; Lee, S.; An, W. Impact of Blood or Erythrocyte Membrane Fatty Acids for Disease Risk Prediction: Focusing on Cardiovascular Disease and Chronic Kidney Disease. Nutrients 2018, 10, 1454.

The analysis of network transformations was performed by examining the ratios of product to reactant masses for each sample. We compared the mean mass ratios between EV^{AML} and EV^{HD} to determine which biochemical reactions were either enhanced or inhibited, following the methodology described by A. Nguyen et al. (ref 65 in our manuscript): [*Nguyen, A.; Rudge, S.A.; Zhang, Q.; Wakelam, M.J. Using Lipidomics Analysis to Determine Signalling and Metabolic Changes in Cells. Curr. Opin. Biotechnol. 2017, 43, 96–103*]. The diagram was generated using the Graph Editor tool (https://csacademy.com/app/graph_editor/, accessed on 10 January 2023) and involved a statistical reinterpretation of data obtained from LipidOne.

Following Reviewer #3's suggestions, we included the following information in the Method section.

METHODS (page 19, lines 856-861)

[.....]

For the network analysis, the diagrams were generated using the Graph Editor tool (https://csacademy.com/app/graph_editor/, accessed on 10 January 2023) and included a statistical reinterpretation of data from LipidOne. Network transformations were analyzed by assessing the ratios of product to reactant masses for each sample. To determine which

biochemical reactions were enhanced or inhibited, we compared the mean mass ratios between EVAML and EVHD, following the methodology outlined by A. Nguyen et al. (65).

Reviewer #4 (Remarks to the Author):

The authors satisfactorily addressed the points raised by the reviewer and substantially revised their manuscript, resulting in substantial improvements that justify its publication.